# GaLLoP: Gradient-based Sparse Learning on Low-Magnitude Parameters

## Abstract

Sparse fine-tuning techniques adapt LLMs to downstream tasks by only tuning a sparse subset of model parameters. However, the effectiveness of sparse adaptation depends on optimally selecting the model parameters to be fine-tuned. In this work, we introduce a novel sparse fine-tuning technique named **GaLLoP**: **G**radient-based Sp**a**rse **L**earning on **Lo**w-Magnitude **P**arameters, which fine-tunes only those model parameters which have the largest gradient magnitudes on downstream tasks and the smallest pre-trained magnitudes, intuitively prioritizing parameters that are highly task-relevant, but minimally disruptive to pre-trained knowledge. Our experimentation with LLaMA3 8B and Gemma 2B as base models shows that GaLLoP consistently improves or matches the in-distribution as well as out-of-distribution performance obtained via the usage of other leading parameter-efficient fine-tuning techniques, including LoRA, DoRA, and SAFT. Our analysis demonstrates that GaLLoP mitigates catastrophic forgetting and memorization of task data, as important pre-trained parameters remain unchanged, and stabilizes performance relative to other fine-tuning techniques, robustly generalizing across most random seeds.

## 1 Introduction

Noting that pre-trained LLMs have a low intrinsic dimension (Aghajanyan et al., 2021a), efficient reparametrization of fine-tuning via a low-rank decomposition made techniques such as LoRA, DoRA, and their variants (Hu et al., 2022; yang Liu et al., 2024; Kopiczko et al., 2024) gain widespread popularity. However, such techniques are still susceptible to overfitting (Zhang et al., 2023; Wang et al., 2024) and hence, fine-tuning models using them can also result in the *memorization* of patterns in the training dataset(s) and the loss of pre-trained knowledge, i.e., *catastrophic forgetting* (McCloskey & Cohen, 1989). This can further impair their *generalizability*, i.e., their ability to perform well on tasks related to yet unseen during fine-tuning, not only for cases wherein the task data used for testing possesses the same distribution as the fine-tuning data and differs only in content (*In-Distribution* (ID)) but also for cases wherein the test data and the fine-tuning data possess different distributions altogether (*Out-of-Distribution* (OOD)) (Miller et al., 2021). Sparse Fine-Tuning (SpFT) techniques (Ansell et al., 2024; Nguyen et al., 2024) have recently emerged as promising alternatives to overcome these issues since they leverage the low intrinsic dimensionality of LLMs by directly fine-tuning only a small fraction of the original model parameters without introducing any additional parameters. However, the effectiveness of sparse adaptation crucially depends upon the selection criterion used to decide which parameters to update.

In this work, we introduce a novel SpFT technique named **GaLLoP**: **G**radient-based Sp**a**rse **L**earning on **Lo**w-Magnitude **P**arameters. To enhance both ID as well as OOD generalizability, GaLLoP fine-tunes only a selected fraction of the model parameters which have the largest gradients on the downstream task (indicating strong ID task relevance (Nguyen et al., 2024)) and the smallest pre-trained magnitudes (preserving the pre-trained knowledge (Zhou et al., 2025; Ramesh et al., 2024) for OOD tasks).

Through experiments on eight datasets with LLaMA3 8B (Grattafiori et al., 2024) and Gemma 2B (Team et al., 2024) as base models, we show that fine-tuning models with GaLLoP consistently enhances both their ID and OOD generalizability by preventing catastrophic forgetting and memorization as compared to state-of-the-art (SOTA) PEFT and post-training model editing techniques.

Furthermore, we show that GaLLoP demonstrates robustness to overtraining and stabilizes performance.

## 2 RELATED WORK

### 2.1 PARAMETER-EFFICIENT FINE-TUNING (PEFT)

Given the large number of trainable parameters in LLMs, PEFT techniques facilitate the compute- and memory-efficient adaptation of LLMs to downstream tasks by only updating a small number of model parameters while keeping the rest frozen (Houlsby et al., 2019). They can be broadly classified into the following three categories:

**Additive Fine-Tuning (AFT):** Additional modules (*adapter* layers) are connected to or introduced into the original *frozen* LLM and these new modules (with a lower number of parameters than the original model) are then fine-tuned (Houlsby et al., 2019; Lin et al., 2020). However, their sequential processing introduces unwanted latency during training as well as inference (Hu et al., 2022).

**Reparametrized Fine-Tuning (RFT):** A low-rank decomposition-based reparametrization of the update matrix can be performed with almost the same effectiveness as the original full-rank representation (Hu et al., 2022). The fine-tuned parameters can then be merged with the pre-trained parameters prior to inference, doing away with any additional latency as in AFT. This category includes **LoRA** (Hu et al., 2022), the original RFT technique; **DoRA**, which fine-tunes magnitude and directional components separately to close LoRA's performance gap with Full Fine-Tuning (FFT) (yang Liu et al., 2024); as well as other variants which either increase LoRA's efficiency (Kopiczko et al., 2024; Valipour et al., 2023) or its expressivity (Huang et al., 2025; Hayou et al., 2024; Nikdan et al., 2024). However, in contrast to GaLLoP, these techniques are still susceptible to overfitting (Zhang et al., 2023; Wang et al., 2024; 2025) since all the newly introduced parameters are fine-tuned on the downstream task.

**Sparse Fine-Tuning (SpFT):** Only a selected fraction of the model parameters is fine-tuned on the downstream task to ensure high memory and compute efficiency. Diff pruning (Guo et al., 2021) and LT-SFT (Ansell et al., 2022) incorporate FFT phases, defeating the very purpose of SpFT. PaFi (Liao et al., 2023) selects the parameters with the smallest pre-trained magnitudes while FISH Mask (Sung et al., 2021), SIFT (Song et al., 2024), SAFT (Nguyen et al., 2024), SMT (He et al., 2025), and SpIEL (Ansell et al., 2024) select the parameters with the highest gradient magnitudes on a downstream task for fine-tuning. While FISH Mask, SIFT, SAFT, and SMT generate a *static* (fixed before fine-tuning) mask of the parameters to be fine-tuned via a single pass on the downstream task dataset, SpIEL iteratively generates *dynamic* masks by alternating between the updation, deletion, and growth of the candidate parameter set. To the best of our knowledge, GaLLoP is the only SpFT technique which incorporates dual parameter selection criteria: high task gradient magnitudes and low pre-trained magnitudes, which consistently improves generalizability and ensures stability.

### 2.2 POST-TRAINING MODEL EDITING

Most fine-tuning techniques (except SAFT (Nguyen et al., 2024)) focus solely on improving ID accuracy on a target downstream dataset at the risk of OOD performance degradation (Miller et al., 2021; Taori et al., 2020). Consequently, weight-space modifications to a single fine-tuned model can serve as a robust and efficient alternative to complex modifications in fine-tuning. **WiSE-FT** (**We**ight-**S**pace **E**nsembles for **F**ine-**T**uning) (Wortsman et al., 2022) performs a linear ensembling of zero-shot and end-to-end fine-tuned model weights to combine robustness with target-specific accuracy. **LiNeS** (**L**ayer-**i**ncreasing **Ne**twork **S**caling) performs layer-wise parameter scaling by maintaining shallower layers close to their pre-trained values for preserving generality while allowing deeper layers to retain task-specific characteristics (Wang et al., 2025). However, the dependence of editing techniques on the fine-tuned model limits their own performance improvements (if any).

## 3 MOTIVATION

Fine-tuning a model with our SpFT algorithm must not only minimize the model's loss on the downstream task (for enhancing ID generalizability) but at the same time, also preserve the pre-

trained knowledge stored in the model parameters (for enhancing OOD generalizability). The first criterion is straightforward to realize: since parameters with the largest gradients generally lead to fast convergence to the optimum on the downstream task, they must be selected for fine-tuning so as to minimize the loss on the downstream task (Nguyen et al., 2024). However, the second criterion does not directly follow from the first since parameters with high gradients on the downstream task can still be highly relevant for the pre-trained model and fine-tuning them can potentially lead to catastrophic forgetting (Aghajanyan et al., 2021b). While the model pruning literature regards parameters with the smallest pre-trained magnitudes as the least important (Han et al., 2015), fine-tuning them does not necessarily lead to the least impact on the pre-trained loss (Lee et al., 2019).

Therefore, we analyze the differences in ID and OOD performance upon fine-tuning only the top-$\rho\%$ of the model parameters based on whether they have a) the smallest or b) the largest pre-trained magnitudes. We consider eight reasoning datasets (Hu et al., 2023) and fine-tune Gemma 2B (Team et al., 2024) models, considering four different density levels. Models are fine-tuned on the training set of each of these datasets (*individually*) and their performance is evaluated on the corresponding dataset's test set (unseen during fine-tuning yet possesses the training set's distribution), which measures their ID accuracy, and the test sets of the remaining seven datasets (unseen during fine-tuning and possess distributions which differ from the training set's distribution), which measures their OOD accuracy (averaged over the seven datasets). The averaged (over all possible ID and OOD task combinations) ID and OOD accuracies of these models are shown in Figure 1 (see Appendix H for further details).

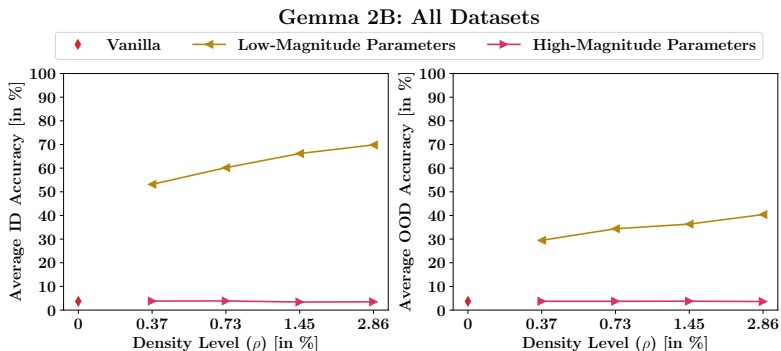

Figure 1: Fine-tuning the top-$\rho\%$ of model parameters with the largest pre-trained magnitudes leads to no performance improvements over the non fine-tuned, vanilla model while fine-tuning the top-$\rho\%$ of model parameters with the smallest pre-trained magnitudes enhances the pre-trained knowledge, leading to significantly increased ID (left) and OOD (right) generalizability.

While fine-tuning the parameters with the largest pre-trained magnitudes leads to no performance improvements over the non fine-tuned, vanilla model (only 3-4% ID and OOD accuracies), fine-tuning the parameters with the smallest pre-trained magnitudes leads to increasingly high ID ($\approx$ 50 - 70% accuracy) and OOD ($\approx$ 30 - 40% accuracy) generalizability. These findings are also in agreement with recent studies (Zhou et al., 2025; Ramesh et al., 2024) and support our hypothesis that fine-tuning low-magnitude parameters enhances pre-trained knowledge.

## 4 METHODOLOGY

Motivated by the results discussed in Section 3, we now present our SpFT algorithm, **GaLLoP**: **G**radient-based **Sp**arse **L**earning on **Lo**w-Magnitude **P**arameters. Given a density level $\rho$, GaLLoP fine-tunes the top-$\rho\%$ of the model parameters with the largest gradient magnitudes (to enhance ID generalizability) and the smallest pre-trained magnitudes (to enhance OOD generalizability). We now give a formal description of how GaLLoP works.

GaLLoP operates in two phases (see Figure 2). Consider a model with a parameter vector $\boldsymbol{\theta} \in \mathbb{R}^D$ which needs to be fine-tuned on a dataset $\mathcal{D} = \{(\boldsymbol{x}_n, \boldsymbol{y}_n)\}_{n=1}^N$, where, $\boldsymbol{x}_n \in \mathbb{R}^P$ and $\boldsymbol{y}_n \in \mathbb{R}^Q$. In the first phase, given a density level $\rho$, GaLLoP selects the learnable parameters using a

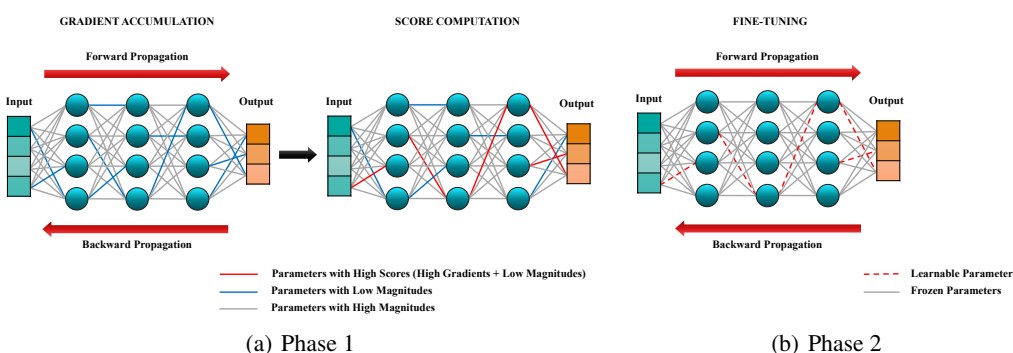

(a) Phase 1          (b) Phase 2

Figure 2: A visual overview of GaLLoP: (a) Phase 1 involves the selection of learnable parameters and (b) Phase 2 involves their fine-tuning. Note that the example of a fully connected neural network has only been considered for ensuring simplicity in visualization; GaLLoP is model-agnostic and can be applied to other models such as transformers and CNNs as well.

dataset sample proportion $d_s$ (corresponding dataset sample size $= d_s N$). To do so, it proceeds by computing the gradient vector $\boldsymbol{g}$ of the net fine-tuning loss $\mathcal{L}(\boldsymbol{x}_n, \boldsymbol{y}_n; \boldsymbol{\theta})$ which is given by:

$$\boldsymbol{g} = \frac{1}{d_s N} \sum_{n=1}^{d_s N} \nabla_{\boldsymbol{\theta}} \mathcal{L}(\boldsymbol{x}_n, \boldsymbol{y}_n; \boldsymbol{\theta}). \tag{1}$$

This allows us to enforce our first parameter selection criterion (high-magnitude gradients). In order to simultaneously enforce our second parameter selection criterion (low- (pre-trained) magnitude parameters), we compute a score vector $\boldsymbol{s}$ such that:

$$\boldsymbol{s} = \left( \frac{\operatorname{abs}(\boldsymbol{g})}{\operatorname{abs}(\boldsymbol{\theta}) + \epsilon} \right), \tag{2}$$

where, abs(.) computes the element-wise absolute value of a vector and a small value $\epsilon \approx 10^{-8}$ has been added to the denominator so as to prevent numerical overflows.

Subsequently, in order to select only the top-$\rho\%$ of all model parameters $\boldsymbol{\theta}$ for fine-tuning based on their corresponding scores $\boldsymbol{s}$, we compute a binary mask vector $\boldsymbol{m}$ such that, for a given parameter $\boldsymbol{\theta}_i$ with a score $\boldsymbol{s}_i$, the mask value $\boldsymbol{m}_i$ is given by:

$$\boldsymbol{m}_i = \begin{cases} 1 & \text{if } \boldsymbol{s}_i \geq s_t, \\ 0 & \text{otherwise.} \end{cases} \tag{3}$$

where, the score threshold $s_t$ is computed as follows:

$$s_t = \texttt{sorted}_d(\boldsymbol{s}_0, \boldsymbol{s}_1, \ldots, \boldsymbol{s}_{\text{end}})[k] \tag{4}$$

where, $k = \lfloor \rho D \rceil$ and $\texttt{sorted}_d(.)$ is the function used for sorting an array in the descending order.

Finally, in the second phase, GaLLoP only updates the selected, *unmasked*, parameters using mini-batch gradient descent while rendering the values of the remaining (unselected), *masked*, parameters unchanged by masking out their gradients during the update using $\boldsymbol{m}$. We give a practical implementation of GaLLoP in Appendix A.

## 5 EXPERIMENTAL SETTINGS

### 5.1 DATASETS

To examine the effectiveness of GaLLoP, we perform fine-tuning experiments on eight common-sense reasoning datasets with predefined training and test sets (Hu et al., 2023): ARC-c (Clark et al., 2018), ARC-e (Clark et al., 2018), BoolQ (Clark et al., 2019), HellaSwag (Zellers et al.,

2019), OBQA (Mihaylov et al., 2018), PIQA (Bisk et al., 2020), SIQA (Sap et al., 2019), and Wino-Grande (Sakaguchi et al., 2021). Further details are provided in Appendix B.

For the purpose of experimentation, we consider all possible ID and OOD combinations involving these datasets by following a round-robin approach. Thus, for each experimental run, a model is fine-tuned on the training set of one of these eight datasets and then evaluated on its test set as well as the test sets of all the remaining seven datasets. Accordingly, eight different experimental runs are performed as part of an experiment for a given density level $\rho$ that is enforced during the fine-tuning of a model with a given algorithm. Performance evaluations on the test set of the dataset used for fine-tuning serve as a measure of ID generalizability while those on the test sets of the remaining datasets (not used for fine-tuning) serve as a measure of the OOD generalizability.

## 5.2 MODEL ARCHITECTURES

We perform our experiments with Gemma 2B (Team et al., 2024) (*relatively small-sized*) and LLaMA3 8B (Grattafiori et al., 2024) (*relatively large-sized*) as base models.

## 5.3 BASELINES

For a rigorous evaluation of performance, we compare GaLLoP with several SOTA fine-tuning algorithms for LLMs. We employ Full Fine-Tuning (FFT) as our fine-tuning baseline and the Zero-Shot (Vanilla) model performance as an overall, non fine-tuning (pre-trained) baseline. We also include LoRA (Hu et al., 2022) and DoRA (yang Liu et al., 2024) from the RFT category, SAFT (Nguyen et al., 2024) and SpIEL (Ansell et al., 2024) from the SpFT category, and WiSE-FT (Wortsman et al., 2022) and LiNeS (Wang et al., 2025) from the post-training model editing category.

## 5.4 EVALUATION METRICS

Since we consider all possible ID and OOD task combinations to ensure a robust evaluation, a fine-tuning experiment with an algorithm $\mathcal{A}$ and a density level $\rho$ on $N^D$ datasets involves $N^D$ experimental runs, such that, in each run $r$, fine-tuning is performed on a dataset $\mathcal{D}^{f_r}$. For a given experimental run $r$, we evaluate the ID and OOD performance of all the fine-tuned models using ID accuracy and OOD accuracy which are defined as follows:

$$\text{Accuracy}_r^{\text{ID}} = \text{accuracy}(\mathcal{D}_{test}^{f_r}) \quad ; \quad \text{Accuracy}_r^{\text{OOD}} = \frac{1}{N^D - 1} \sum_{\substack{n=1 \\ n \neq f_r}}^{N^D} \text{accuracy}(\mathcal{D}_{test}^n), \quad (5)$$

where, $\text{accuracy}(\mathcal{D}_{test})$ computes the percentage of correct responses generated by a fine-tuned model on the test set $\mathcal{D}_{test}$ of a dataset $\mathcal{D}$.

In addition to the standard accuracy metrics, we introduce two new metrics, the Forget Ratio and the Collapse Rate, which aim to quantify the extent of catastrophic forgetting and memorization incurred upon fine-tuning (respectively). For a given experimental run $r$, while fine-tuning leads to performance improvements on the (ID) downstream task, it may result in the degradation of zero-shot (vanilla) performance on the remaining OOD tasks due to the loss of pre-trained knowledge, i.e., catastrophic forgetting. Accordingly, we define the forget ratio as a measure of this relative drop in OOD performance such that:

$$\text{Forget Ratio}_r = \max\left(0, \frac{\text{Accuracy}_{\text{Vanilla}, r}^{\text{OOD}} - \text{Accuracy}_r^{\text{OOD}}}{\text{Accuracy}_{\text{Vanilla}, r}^{\text{OOD}}}\right), \quad (6)$$

where, $\text{Accuracy}_{\text{Vanilla}, r}^{\text{OOD}}$ refers to the accuracy of the zero-shot (vanilla) model on the OOD test sets for the $r^{\text{th}}$ experimental run. From the above equation, it follows that gains in OOD performance, relative to the performance of the vanilla model, lead to a 0% forget ratio as desired.

The collapse rate for a given experimental run $r$ measures the extent to which fine-tuning results in severe memorization of patterns present in $\mathcal{D}^{f_r}$. It is computed by determining the total number of OOD datasets, on the test sets of which, the accuracy drops to $\approx 0\%$ (indicative of a complete

collapse) and is hence, defined as follows:

$$\text{Collapse Rate}_r = \sum_{\substack{n=1 \\ n \neq f_r}}^{N^D} \mathbb{1}[\lfloor \text{accuracy}(\mathcal{D}_{test}^n) \rfloor = 0\%], \tag{7}$$

where, $\mathbb{1}[.]$ denotes the indicator function which evaluates to 1 when its argument is True and 0 otherwise.

For each fine-tuning algorithm $\mathcal{A}$ and density level $\rho$, the obtained performance and the extent of catastrophic forgetting and memorization incurred across the entire experiment consisting of $N^D$ experimental runs is given by the average of the aforementioned four metrics across all the runs.

### 5.5 IMPLEMENTATION DETAILS

We perform all our experiments using Torchtune (torchtune maintainers & contributors, 2024) and Huggingface's PEFT library (Mangrulkar et al., 2022). We use NVIDIA RTX A6000 and/or NVIDIA RTX 6000 Ada GPUs with 48 GB of internal memory and perform mixed-precision (using the BF16 datatype), distributed fine-tuning of all our models (except SpIEL, which can be only run on a single GPU (Ansell et al., 2024)) using Fully Sharded Data Parallel (FSDP) (Zhao et al., 2023), activation checkpointing, and gradient accumulation. To ensure a fair comparison, we apply each fine-tuning algorithm across all the same transformer layers and maintain the same density level ($\rho$) for a given experiment. We explore four density levels: {0.24%, 0.47%, 0.93%, 1.85%} for LLaMA3 8B, and {0.37%, 0.73%, 1.45%, 2.86%} for Gemma 2B which correspond to ranks {8, 16, 32, 64} used for reparametrized fine-tuning. Further details on the hyperparameters are given in Appendix C.

## 6 RESULTS AND DISCUSSION

### 6.1 ID AND OOD ACCURACY

Figure 3 shows the averaged ID and OOD accuracies of models fine-tuned with GaLLoP against those of the vanilla models and models fine-tuned and/or edited with the competing algorithms (per-run ID and OOD accuracies are discussed in detail in Section E.1 of Appendix E).

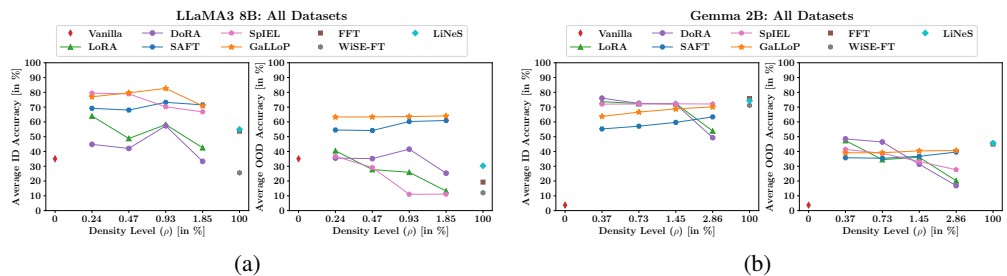

(a)  (b)

Figure 3: a) LLaMA3 8B models fine-tuned with GaLLoP form a dominant Pareto front for both ID and OOD performance (on average) over all the models fine-tuned and/or edited with the competing algorithms across all density levels. b) Gemma 2B models fine-tuned with GaLLoP attain consistently high and balanced ID and OOD performance (on average) across all density levels.

LLaMA3 8B and Gemma 2B models fine-tuned with GaLLoP attain high ID and OOD accuracies (on average) while maintaining relatively balanced ID and OOD performance across all the density levels. In fact, LLaMA3 8B models fine-tuned with GaLLoP form a dominant Pareto front for both ID and OOD performance over all the other fine-tuned and/or edited models. Models fine-tuned with GaLLoP consistently surpass those fine-tuned with SAFT with a high average margin of roughly 10% (for both ID and OOD performance), which only narrows down for the highest density level due to *high gradient dilution* and *low-magnitude dilution*. With the increase in the density

level, parameters with a relatively higher pre-trained magnitude and lower gradient magnitude also fall into the 'high gradient and low-magnitude' category, resulting in an overlap between the set of parameters selected by SAFT and GaLLoP. Nevertheless, as is observed, GaLLoP is more efficient in its selection of parameters and would still form a performance upper bound over SAFT even as $\rho \rightarrow 100\%$ (asymptotic limit), particularly because it would still select a larger/at least as large a concentration of low-magnitude high gradient parameters than/as the latter.

While models fine-tuned with SpIEL show high ID accuracies, they perform poorly on OOD tasks with a huge average gap ($\gtrsim 30\%$) in their own ID and OOD accuracies (on average), which widens with the increase in the density level. Contrary to the other SpFT methods, SpIEL iteratively selects parameters, potentially updating a much larger number of parameters overall. Hence, it leads to increasingly high overfitting on the in-domain distribution with the increase in the density level. Models fine-tuned with RFT techniques also exhibit high levels of overfitting since all the newly introduced trainable parameters are fine-tuned on the training set. Further, while the ID and OOD accuracies obtained via RFT are greater than those obtained via GaLLoP for Gemma 2B for the first two density levels, these gains are only confined to certain datasets and not to others due to catastrophic forgetting and memorization, as we show subsequently in Section 6.2 and Section 6.3.

Moreover, the performance of FFTed models is highly dependent on their zero-shot performance. While FFT works quite well for the small Gemma 2B model by yielding high performance levels comparable to those yielded by GaLLoP, it completely fails for the 4X larger LLaMA3 8B model. Overtraining, i.e., considerably higher pre-training (6T pre-training tokens for Gemma 2B (Team et al., 2024) versus 15T pre-training tokens for LLaMA3 8B (Grattafiori et al., 2024)), makes models much more sensitive to parameter updates, leading to severe catastrophic forgetting and losses in OOD as well as ID generalizability (Kumar et al., 2022; Springer et al., 2025). In fact, this is also the reason why the performance gap between GaLLoP and the competing algorithms decreases for Gemma 2B as compared to LLaMA3 8B. However, GaLLoP exhibits superior performance in both pre-training regimes which implies that it is robust to overtraining, a pre-training regime wherein the competing algorithms simply fail. Finally, the fine-tuned performance dependence and meagre performance improvements (*if any*) yielded by WiSE-FT and LiNeS defeats the very purpose of these post-training model editing techniques which were developed with an aim to provide higher robustness to generalization while avoiding the complexities of fine-tuning.

## 6.2 CATASTROPHIC FORGETTING

Figure 4 shows the averaged forget ratios of models fine-tuned with GaLLoP against those fine-tuned and/or edited with the competing algorithms (per-run forget ratios are discussed in Section E.2 of Appendix E). In an ideal scenario, the forget ratio must be zero since fine-tuning a model must not lead to any loss in the pre-trained knowledge stored in the model.

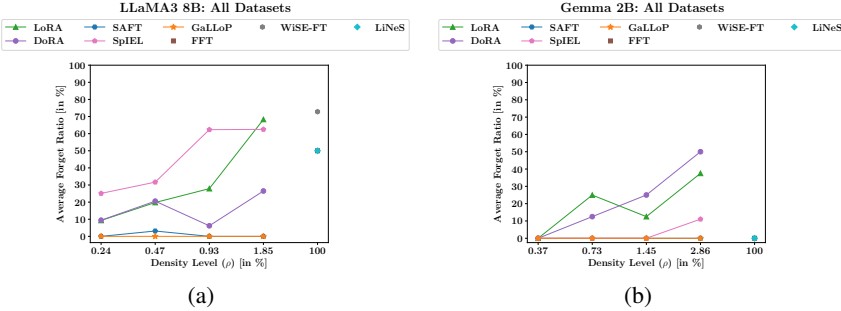

Figure 4: Models fine-tuned with GaLLoP show 0% forget ratios across all density levels.

LLaMA3 8B and Gemma 2B models fine-tuned with GaLLoP do not undergo catastrophic forgetting since they consistently show 0% forget ratios. In contrast, all the competing algorithms show high forget ratios which may even take on increasingly high values with the increase in the density level due to overfitting. In general, for all competing algorithms, fine-tuned Gemma 2B models show significantly decreased forget ratios as compared to their LLaMA3 8B counterparts. While this

does happen because of overtraining, there are, in actuality, *two sides* of the same coin. While lesser pre-training of Gemma 2B (as compared to LLaMA3 8B) lowers the sensitivity of updates to its parameters and allows fine-tuning it to be more stable and performant (which decreases the risk of catastrophic forgetting) (Springer et al., 2025), it also leads to the decreased and (here) near-zero, vanilla (zero-shot) performance of the former as compared to the latter, in the first place itself (see Figure 15 in Appendix E). Since forget ratios are defined relative to vanilla performance (Equation 6), even though the OOD performance of certain fine-tuned Gemma 2B models is quite low, it never goes below that of their vanilla counterpart, which leads to all of them attaining a 0% forget ratio. Nevertheless, these fine-tuning algorithms may lead to a more severe phenomenon which impairs generalizability: *memorization*. This forms the subject of our subsequent discussion.

## 6.3 MEMORIZATION

Figure 5 shows the total collapse rates of models fine-tuned with GaLLoP against those fine-tuned and/or edited with the competing algorithms. In an ideal scenario, the collapse rate must be zero since fine-tuning a model must not lead to the memorization of any kind of patterns present in the dataset used for fine-tuning it.

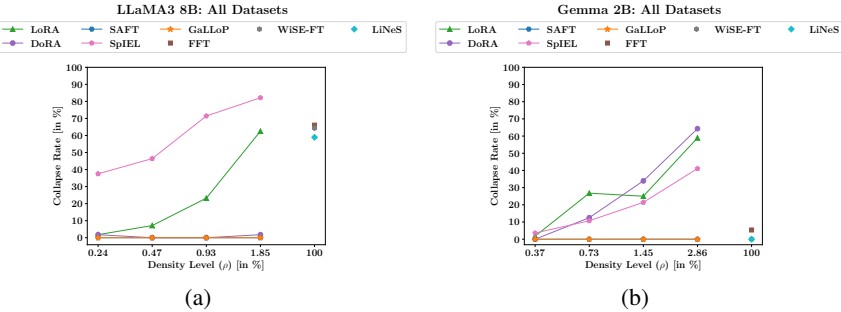

(a)  (b)

Figure 5: Models fine-tuned with GaLLoP show 0% collapse rates across all density levels.

LLaMA3 8B and Gemma 2B models fine-tuned with GaLLoP and SAFT exhibit 0% collapse rates across all density levels. Combined with the dominant ID as well as OOD performance and 0% forget ratios exhibited by models fine-tuned with GaLLoP over those fine-tuned with SAFT (see Figure 3 and Figure 4), it is clear that fine-tuning models with GaLLoP allows for the attainment of high yet balanced generalizability.

For the other algorithms, fine-tuned LLaMA3 8B models show higher collapse rates than their Gemma 2B counterparts due to overtraining. Furthermore, while memorization inevitably occurs due to overfitting, it can manifest in different ways. Throughout our experiments, the most dominant form of memorization observed by us is that of the memorization of the response format of the dataset used for fine-tuning. With the increase in the density level, this kind of memorization leads to models fine-tuned with these algorithms increasingly failing on an OOD task which does not share the same response format as the ID task. Another, more severe form of memorization seems to be more pervasive since it affects the performance of fine-tuned LLaMA3 8B models across all the datasets (ID as well as OOD) irrespective of whether they share response formats or not: memorization of the most frequently occurring words/phrases in the fine-tuning dataset. This leads to the generation of repetitive sequences as answers and consequently, degrades their intelligibility. Since we find that repetition is only restricted to RFT and does not occur on performing SpFT, we attribute its occurrence to the fact that the former class of algorithms restrict fine-tuning to be performed on newly introduced parameters, tied to specific positions in the model, which leads to *concerted overfitting* while the latter class of algorithms allow for much more flexibility in fine-tuning via a position-independent selection of parameters to be fine-tuned and lead to *scattered (unstructured) learning*. Finally, LLaMA3 8B models fine-tuned with SpIEL and FFT (and hence, even those edited with WiSE-FT and LiNeS) occasionally undergo the worst form of memorization: generation of the EOS token. In fact, not only FFT but also SpIEL (as explained earlier in Section 6.1) fine-tunes a much larger number of parameters than the other algorithms and hence, both of them are more

prone to lead to instability in the learning dynamics in the overtrained regime which can lead to such a detrimental form of memorization. We discuss all these interesting cases in more detail with representative samples of LLM-generated responses in Appendix F.

## 6.4 STABILITY

Figure 6 shows the boxplots of ID and OOD performance for LLaMA3 8B models fine-tuned with GaLLoP against models fine-tuned with the competing algorithms across 20 different random seeds. We deliberately choose the PIQA dataset here because the FFTed LLaMA3 8B model attains the lowest ID and OOD performance on this dataset (see Figure 8(e) in Appendix E). Moreover, we only consider the highest $\rho$ ($= 1.85\%$) so as to rigorously analyze performance stability in the overtrained regime with the largest examined number of update-sensitive parameters. Note that we do not consider editing techniques here since their stability is inherently dependent upon the stability of the fine-tuned model (see Section 6.1). More details are given in Appendix G.

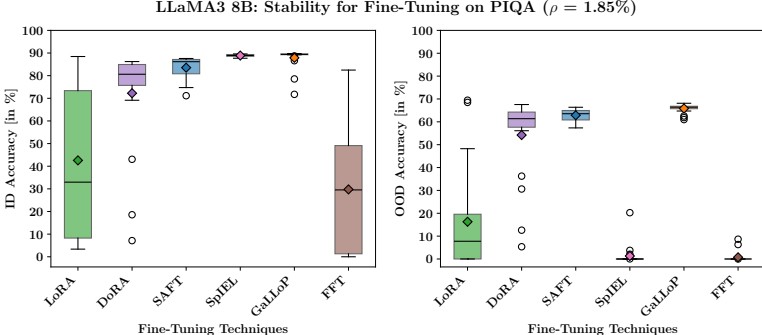

Figure 6: LLaMA3 8B models fine-tuned with GaLLoP are the most stable and consistently attain the highest ID and OOD accuracies across 20 different random seeds upon fine-tuning on the PIQA dataset with the highest density level ($\rho = 1.85\%$).

Models fine-tuned with GaLLoP consistently attain the highest median ID and OOD accuracies with the least interquartile ranges. Furthermore, the difference between the farthest outlier and median performance is amongst the lowest for GaLLoP and never falls below the lowest performance level of SAFT. Hence, fine-tuning with GaLLoP leads to the highest performance stability. In contrast, models fine-tuned with competing algorithms show high performance instability and lower median accuracies, with instability being the highest on performing RFT or FFT (Dodge et al., 2020; Mosbach et al., 2021). Directly leveraging low intrinsic dimensionality by infusing sparsity (as in SpFT) effectively regularizes fine-tuning as compared to reparametrization (as in RFT) which does not constrain updates to the newly introduced low-rank matrices and hence, leads to instability.

## 7 CONCLUSION

In this work, we have thus developed a novel SpFT technique named **GaLLoP**: **G**radient-based **Sp**arse **L**earning on **Lo**w-Magnitude **P**arameters which enhances the ID as well as the OOD generalizability of models, prevents catastrophic forgetting and memorization, ensures robustness to overtraining, and stabilizes performance (see Table 1 for a summary of comparisons with competing algorithms). Nevertheless, as is the case for other SpFT techniques, GaLLoP leads to unstructured sparsity. Accelerating unstructured fine-tuning poses a challenge for current hardware that is optimized for performing dense and/or structured computations (Hooker, 2021; Nguyen et al., 2024). An interesting direction for future work could therefore be to perform densification of this unstructured sparsity by following an aggregation scheme (He et al., 2025) so as to make it more structured (only) for fine-tuning and further increase the memory and compute efficiency of GaLLoP.

| Performance Metric | GaLLoP | Competing Algorithms |
|---|---|---|
| **ID and OOD Accuracy** | Fine-tuned models form a **dominant Pareto front** and/or **show high performance** (LLaMA3 8B) or **consistently show high and balanced performance** (Gemma 2B). | Fine-tuned models either breakdown in the overtrained regime (LLaMA3 8B) or increasingly overfit the distribution of the fine-tuning dataset with the increase in the density level and/or fine-tuning dataset size (Gemma 2B). |
| **Forget Ratio** (Catastrophic Forgetting) | Fine-tuned models show **0% forget ratios** across all density levels and experimental runs. | Fine-tuned models show increasingly high forget ratios with the increase in the density level because they overfit the distribution of the fine-tuning dataset. |
| **Collapse Rate** (Severe Memorization) | Fine-tuned models show **0% collapse rates** across all density levels and experimental runs. | Fine-tuned models (except SAFT-based) show increasingly high collapse rates with the increase in the density level because they overfit the distribution of the fine-tuning dataset. |
| **Stability** | Fine-tuned models show the **highest performance stability** by attaining the highest median ID and OOD accuracies with the least interquartile ranges and showing 0% variability in their forget ratios and collapse rates. | Fine-tuned models exhibit unstable performance by attaining lower median ID and OOD accuracies with high interquartile ranges and showing high values of and/or high variability in their forget ratios and collapse rates. |

Table 1: Comparative summary of our experimental results showing the benefits of fine-tuning models with GaLLoP.

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

# Appendix

## Table of Contents

## A  PRACTICAL IMPLEMENTATION OF GaLLoP

Following the method outlined in Section 4, in practice, since we fine-tune LLMs with billions of parameters, sorting the corresponding set of billions of scores in order to compute the score threshold $s_t$ (see Equation 4) is computationally inefficient. Therefore, we perform layer-wise uniform random sampling of the scores and utilize only a small proportion $s$ of the total number of scores in order to compute $s_t$.

Accordingly, we now use a more specific, layer-wise notation for the model to be fine-tuned.

Consider a deep neural network model with $L$ layers in total (1 input layer of width $p$, $L-2$ hidden layers of width $k$, and 1 output layer of width $q$), represented by $\boldsymbol{y} = F_{\{\boldsymbol{\Theta}^l\}_{l=1}^L}(\boldsymbol{x})$, such that the parameter matrices $\boldsymbol{\Theta}^l$ possess the following set of dimensions:

$$\boldsymbol{\Theta}^1 \in \mathbb{R}^{p \times k}$$
$$\boldsymbol{\Theta}^l \in \mathbb{R}^{k \times k} \quad \forall \quad l \in \{2, \dots, L-1\}$$
$$\boldsymbol{\Theta}^L \in \mathbb{R}^{k \times q}$$

This model is to be fine-tuned on a dataset $\mathcal{D} = \{(\boldsymbol{x}_n, \boldsymbol{y}_n)\}_{n=1}^N$, where, $\boldsymbol{x}_n \in \mathbb{R}^P$ and $\boldsymbol{y}_n \in \mathbb{R}^Q$.

In line with this notation, Equation 1, Equation 2, and Equation 3 can be re-written as follows:

$$\boldsymbol{G}^l = \frac{1}{d_s N} \sum_{n=1}^{d_s N} \nabla_{\boldsymbol{\Theta}^l} \mathcal{L}(\boldsymbol{x}_n, \boldsymbol{y}_n; \{\boldsymbol{\Theta}^l\}_{l=1}^L), \tag{8}$$

$$\boldsymbol{S}^l = \left( \frac{\text{abs}(\boldsymbol{G}^l)}{\text{abs}(\boldsymbol{\Theta}^l) + \epsilon} \right), \tag{9}$$

$$\boldsymbol{M}_{ij}^l = \begin{cases} 1 & \text{if } \boldsymbol{S}_{ij}^l \geq s_t, \\ 0 & \text{otherwise,} \end{cases} \tag{10}$$

where, abs(.) computes the element-wise absolute value of a matrix.

Now, on performing the layer-wise uniform random sampling of scores (with a small proportion $s$ of the total number of scores), since the full set of scores is still required for the computation of the binary mask (note that an element-wise comparison: $\mathcal{O}(D)$, is much less compute intensive than sorting: $\mathcal{O}(D \log D)$, for large $D$), we modify Equation 8, Equation 9, and Equation 10 such that for each layer $l$, $\boldsymbol{G}^l$, $\boldsymbol{S}^l$, and $\boldsymbol{M}^l$ respectively become:

$$\boldsymbol{G}^l = \left( \frac{\frac{1}{d_s N} \sum\limits_{n=1}^{d_s N} \text{abs}(\nabla_{\boldsymbol{\Theta}^l} \mathcal{L}(\boldsymbol{x}_n, \boldsymbol{y}_n; \{\boldsymbol{\Theta}^l\}_{l=1}^L))}{\text{abs}(\boldsymbol{\Theta}^l) + \epsilon} \right), \tag{11}$$

$$\boldsymbol{S}^l \overset{s}{\sim} \mathcal{U}\left(\boldsymbol{G}^l\right), \tag{12}$$

$$\boldsymbol{M}^l = \begin{cases} 1 & \text{if } \boldsymbol{G}^l \geq s_t, \\ 0 & \text{otherwise.} \end{cases} \tag{13}$$

Moreover, since we now modify the gradients *in place* using Equation 11 and instead of storing all the $D$ scores (as in Equation 2), we now store only $s\%$ of them (as in Equation 12), we now save on $(100 - s)\%$ of the memory as well.

Throughout our experiments (see Figure 7 for representative cases), we observe that with a dataset sample proportion ($d_s$) of 0.5, sampling only $s = 10\%$ of the scores to compute $s_t$ allows us to obtain the required $\rho\%$ of the parameters above the threshold which are to be selected for fine-tuning, i.e., the effective density level $\rho_{eff}$ matches the required density level $\rho$. This drastically reduces the compute and memory requirements of GaLLoP and hence, makes it highly efficient.

An algorithmic implementation of GaLLoP is given in Algorithm 1. Note that we set $\epsilon$ to $10^{-8}$ in Equation 11 as it is just an order of magnitude smaller than the smallest non-zero pre-trained parameter magnitude in both the LLMs used in our experiments and hence, does not artificially inflate any pre-trained parameter's magnitude.

---

**Algorithm 1:** GaLLoP

---

**Input:** Fine-Tuning Dataset $\{(\boldsymbol{x}_n, \boldsymbol{y}_n)\}_{i=1}^N$, Number of Layers $L$, Density Level $\rho$,
    Pre-Trained Model $F_{\{\boldsymbol{\Theta}^l\}_{l=1}^L}(\boldsymbol{x})$, Learning Rate $\eta$, Number of Epochs $T$,
    Dataset Sample Proportion $d_s$, Score Sample Proportion $s$,
    Loss Function $\mathcal{L}(\boldsymbol{x}_n, \boldsymbol{y}_n; \{\boldsymbol{\Theta}^l\}_{l=1}^L)$

**global** $D$, $\epsilon$
$\epsilon \leftarrow 10^{-8}$
Initialize total parameter count $D \leftarrow 0$
**for** $l \leftarrow 1$ **to** $L$ **do**
  $D \leftarrow D + \texttt{numel}(\boldsymbol{\Theta}^l)$
**end**
**# Phase 1: Selection of Learnable Parameters**
Initialize layer-wise gradient matrices $\boldsymbol{G}^l \leftarrow 0 \quad \forall \quad l \in \{1, \ldots, L\}$
**for** $n \leftarrow 1$ **to** $d_s N$ **do**
  **for** $l \leftarrow 1$ **to** $L$ **do**
    $\boldsymbol{G}^l \leftarrow \boldsymbol{G}^l + \dfrac{1}{d_s N} \nabla_{\boldsymbol{\Theta}^l} \mathcal{L}(\boldsymbol{x}_n, \boldsymbol{y}_n; \{\boldsymbol{\Theta}^l\}_{l=1}^L)$
  **end**
**end**
Initialize layer-wise score matrices $\boldsymbol{S}^l \leftarrow 0 \quad \forall \quad l \in \{1, \ldots, L\}$
**for** $l \leftarrow 1$ **to** $L$ **do**
  $\boldsymbol{G}^l \leftarrow \left( \dfrac{\text{abs}(\boldsymbol{G}^l)}{\text{abs}(\boldsymbol{\Theta}^l) + \epsilon} \right)$
  $\boldsymbol{S}^l \overset{s}{\sim} \mathcal{U}\left(\boldsymbol{G}^l\right)$
**end**
Compute threshold $s_t \leftarrow \texttt{sorted}_d(\boldsymbol{S}_{00}^0, \boldsymbol{S}_{01}^0, \ldots, \boldsymbol{S}_{\text{end}}^L)[\lfloor \rho D \rfloor]$
Compute layer-wise mask matrices $\boldsymbol{M}^l \leftarrow \begin{cases} 1 & \text{if } \boldsymbol{G}^l \geq s_t, \\ 0 & \text{otherwise.} \end{cases} \quad \forall \quad l \in \{1, \ldots, L\}$
**# Phase 2: Fine-Tuning of Learnable Parameters**
Initialize new layer-wise parameter matrices $\tilde{\boldsymbol{\Theta}}^l \leftarrow \boldsymbol{\Theta}^l \quad \forall \quad l \in \{1, \ldots, L\}$
**for** $t \leftarrow 1$ **to** $T$ **do**
  **for** $l \leftarrow 1$ **to** $L$ **do**
    $\tilde{\boldsymbol{\Theta}}^l \leftarrow \tilde{\boldsymbol{\Theta}}^l - \dfrac{1}{\eta} \left( \boldsymbol{M}^l \odot \dfrac{1}{|\mathcal{B}_t|} \sum_{(\boldsymbol{x}_n, \boldsymbol{y}_n) \in \mathcal{B}_t} \nabla_{\boldsymbol{\Theta}^l} \mathcal{L}(\boldsymbol{x}_n, \boldsymbol{y}_n; \{\boldsymbol{\Theta}^l\}_{l=1}^L) \right) \Bigg|_{\boldsymbol{\Theta}^l = \tilde{\boldsymbol{\Theta}}^l}$
  **end**
**end**
**Output:** Fine-Tuned Model $\tilde{F}_{\{\tilde{\boldsymbol{\Theta}}^l\}_{l=1}^L}(\boldsymbol{x})$

---

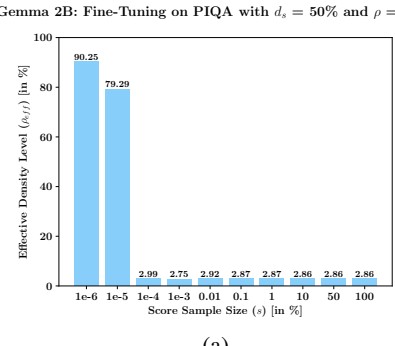 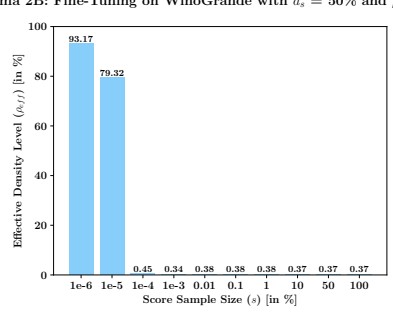

(a)  (b)

Figure 7: Representative experiments show that sampling only $s = 10\%$ of the scores allows us to attain a stable effective density level $\rho_{eff}$ which matches the required density level $\rho$ while drastically reducing the memory and compute requirements of GaLLoP.

## B   DATASETS

A detailed overview of the eight commonsense reasoning datasets which have been used throughout our experiments is as follows:

- **ARC-c**: The **A**I2 **R**easoning **C**hallenge (ARC) dataset consists of natural, grade-school level science questions, authored for standardized tests taken by humans. ARC-c is the Challenge Set of this question bank, containing only those questions which have been incorrectly answered by both a retrieval-based and a word co-occurrence algorithm. Hence, high performance on this dataset requires AI models to possess advanced reasoning capabilities (Clark et al., 2018).

- **ARC-e**: The Easy Set of the ARC dataset which consists of the questions remaining in the ARC dataset post the formation of the Challenge Set (Clark et al., 2018).

- **BoolQ**: A dataset of naturally occurring, True/False (boolean) questions which have been formed from queries directed to the Google search engine. It requires AI models to utilize complex, non-factoid information for solving them (Clark et al., 2019). Note that the version of the BoolQ dataset used by us – which is created by Hu et al. (2023) – does not include the (context) passages alongside the questions at all.

- **HellaSwag**: The **H**arder **E**ndings, **L**onger contexts, and **L**ow-shot **A**ctivities for **S**ituations **W**ith **A**dversarial **G**enerations dataset tests the robustness of AI models towards commonsense Natural Language Inference (NLI). It comprises questions formed from the ActivityNet captions dataset (Krishna et al., 2017) and the online WikiHow manuals with challenging, incorrect answer options obtained via Adversarial Filtering (Zellers et al., 2019).

- **OBQA**: The **O**pen**B**ook**QA** dataset tests the multi-hop reasoning ability of AI models in answering questions based on elementary-level science facts and commonsense knowledge. Note that the version of the OBQA dataset used by us – which is created by Hu et al. (2023) – does not include the (open book) scientific facts alongside the questions at all (Mihaylov et al., 2018).

- **PIQA**: The **P**hysical **I**nteraction: **Q**uestion **A**nswering dataset tests the physical commonsense of AI models by requiring knowledge of the physical properties of objects used in day-to-day life by humans to answer the 'how-to' questions contained in it. Syntactically and topically similar semantic perturbations or alternative solutions have been introduced by annotators to counteract the possibility of spurious biases assisting AI models in achieving high performance (Bisk et al., 2020).

- **SIQA**: The **S**ocial **I**ntelligence **Q**uestion **A**nswering dataset tests the social and emotional intelligence of AI models with regards to everyday situations. Some incorrect answer options are obtained via question-switching around the same context so as to minimize the occurrence of stylistic artifacts arising from the cognitive biases of human annotators which

could otherwise be exploited by an AI model for obtaining high performance (Sap et al., 2019).

- **WinoGrande**: This large-scale fill-in-the-blank dataset tests pronoun resolution-based commonsense reasoning capabilities of AI models and is unsolvable upon their complete reliance on embedding associations. Systematic algorithmic bias reduction is performed via AfLite, a lightweight and improved Adversarial Filtering algorithm (Sakaguchi et al., 2021).

The sizes of the training and test splits of each of these datasets as well as their response formats are provided in Table 2.

Table 2: Details of all the datasets (Hu et al., 2023) used across all our experiments.

| Dataset | Training Set Size | Test Set Size | Response Format |
|---------|-------------------|---------------|-----------------|
| ARC-c | 1119 | 1172 | the correct answer is answer$\langle$ID$\rangle$ ID = {1,2,3,4} |
| ARC-e | 2251 | 2376 | the correct answer is answer$\langle$ID$\rangle$ ID = {1,2,3,4} |
| BoolQ | 9427 | 3270 | the correct answer is $\langle$BOOL$\rangle$ BOOL = {true/false} |
| HellaSwag | 39905 | 10042 | the correct answer is ending$\langle$ID$\rangle$ ID = {1,2,3,4} |
| OBQA | 4957 | 500 | the correct answer is answer$\langle$ID$\rangle$ ID = {1,2,3,4} |
| PIQA | 16113 | 1837 | the correct answer is solution$\langle$ID$\rangle$ ID = {1,2} |
| SIQA | 33410 | 1954 | the correct answer is answer$\langle$ID$\rangle$ ID = {1,2,3} |
| WinoGrande | 63238 | 1267 | the correct answer is option$\langle$ID$\rangle$ ID = {1,2} |

# C  HYPERPARAMETERS FOR FINE-TUNING ALGORITHMS

We fix the maximum input sequence length to 512 in all our experiments and utilize the chunked cross-entropy loss for fine-tuning, so as to save memory by upcasting only a single chunk (token) at a time from BF16 to FP32 while computing the loss. We perform early stopping in order to select the best fine-tuned model checkpoint across the three epochs, based on the ID accuracy on the test-dev set (here, the ID test set serves as the test-dev set for a given experimental run). We set $\alpha = 0.5$ for WiSE-FT and $\alpha = \beta = 0.5$ for LiNeS following the overall recommendations of their authors (Wortsman et al., 2022; Wang et al., 2025). We perform activation checkpointing in the same manner for all fine-tuning algorithms and apply it to only the attention layers of our models since these layers account for a majority of the activation memory and utilizing activation checkpointing can help in reducing the memory footprint of fine-tuning. Application of activation checkpointing to only the attention layers is a default in Torchtune recipes (torchtune maintainers & contributors, 2024). Furthermore, for our experiments, the number of gradient accumulation steps is always chosen such that the effective batch size (= Per-GPU batch size $\times$ gradient accumulation steps $\times$ no. of GPUs) remains 16 for each fine-tuning experiment for a fair comparison. The effective batch size is limited to 16 because unfortunately, SpIEL does not support distributed fine-tuning and its authors did not use gradient accumulation in their own experiments (Ansell et al., 2024). Consequently, for our experiments with SpIEL, 16 is the largest batch size which can fit in our NVIDIA RTX A6000/RTX 6000 Ada GPU's memory. The specific hyperparameters are as follows:

## C.1  GALLOP AND SAFT

Table 3 shows the hyperparameter configurations employed upon performing fine-tuning with GaL-LoP and SAFT. We use a low learning rate of 2e-5 in order to prevent divergences during fine-tuning.

Table 3: Hyperparameter configurations of GaLLoP and SAFT used while fine-tuning LLaMA3 8B and Gemma 2B models. * indicates that Effective Batch Size = (Per-GPU Batch Size $\times$ Gradient Accumulation Steps $\times$ No. of GPUs).

| Hyperparameters | LLaMA3 8B | Gemma 2B |
|---|---|---|
| Dataset Sample Proportion $d_s$ | 0.5 | |
| Score Sample Proportion $s$ | 0.1 | |
| Optimizer | Fused AdamW ($\beta_1 = 0.9$; $\beta_1 = 0.999$) | |
| Weight Decay | 0 | |
| Learning Rate | 2e-5 | |
| Learning Rate Scheduler | Cosine | |
| Warmup Steps | 100 | |
| Effective Batch Size* | 16 | |
| Per-GPU Batch Size | 4 | |
| Gradient Accumulation Steps | 1 | 2 |
| No. of GPUs | 4 | 2 |
| Epochs | 3 | |
| Target Modules | q_proj, k_proj, v_proj, up_proj, down_proj | |

## C.2  SPIEL

Table 4 shows the hyperparameter configurations employed upon performing fine-tuning with SpIEL. Most of these are in line with those mentioned in (Ansell et al., 2024). However, unlike Ansell et al. (2024) and just like with GaLLoP and SAFT, we find that a lower learning rate works better since it prevents divergences during fine-tuning. Moreover, during the course of our experiments, we find that adding weight decay upon fine-tuning with SpIEL does not help. Upon exploring the addition of weight decay from amongst the following four values: $\{0, 3, 10, 30\}$ (same as those examined by Ansell et al. (2024)) while fine-tuning LLaMA3 8B, we find that adding any amount of weight decay leads to a comparable/slightly-lowered performance with/than a zero weight decay. Hence, we do not use weight decay while fine-tuning models using SpIEL.

Table 4: Hyperparameter configurations of SpIEL used while fine-tuning LLaMA3 8B and Gemma 2B models. * indicates that Effective Batch Size = (Per-GPU Batch Size × Gradient Accumulation Steps × No. of GPUs).

| Hyperparameters | LLaMA3 8B | Gemma 2B |
|---|---|---|
| Optimizer | AdamW ($\beta_1 = 0.9$; $\beta_1 = 0.999$) | |
| Weight Decay | 0 | |
| Learning Rate | 2e-5 | |
| Learning Rate Scheduler | Linear | |
| Warmup Ratio | 0.03 | |
| Effective Batch Size | 16 | |
| Per-GPU Batch Size | 16 | |
| Gradient Accumulation Steps | 1 | |
| No. of GPUs | 1 | |
| Epochs | 3 | |
| Target Modules | q_proj, k_proj, v_proj, up_proj, down_proj | |

### C.3 FFT

Table 5 shows the hyperparameter configurations employed upon performing fine-tuning with FFT. Just like with GaLLoP and SAFT, we use a low learning rate of 2e-5 in order to prevent divergences during fine-tuning.

Table 5: Hyperparameter configurations of FFT used while fine-tuning LLaMA3 8B and Gemma 2B models. * indicates that Effective Batch Size = (Per-GPU Batch Size × Gradient Accumulation Steps × No. of GPUs).

| Hyperparameters | LLaMA3 8B | Gemma 2B |
|---|---|---|
| Optimizer | Fused AdamW ($\beta_1 = 0.9$; $\beta_1 = 0.999$) | |
| Weight Decay | 0 | |
| Learning Rate | 2e-5 | |
| Learning Rate Scheduler | Cosine | |
| Warmup Steps | 100 | |
| Effective Batch Size | 16 | |
| Per-GPU Batch Size | 4 | |
| Gradient Accumulation Steps | 1 | |
| No. of GPUs | 4 | |
| Epochs | 3 | |

### C.4 LoRA and DoRA

Table 6 and Table 7 show the hyperparameter configurations employed upon performing fine-tuning with LoRA and DoRA (respectively). Most of these are in line with those mentioned in (yang Liu et al., 2024). Unlike GaLLoP and SAFT, we observe that a low learning rate of 2e-5 slows down learning and results in decreased ID and OOD performance for both LoRA and DoRA. Hence, we use the learning rates of 3e-4 and 1e-4 suggested by yang Liu et al. (2024) for faster (and divergence-free) learning and better performance for LoRA and DoRA (respectively). Moreover, during the course of our experiments, we find that adding dropout upon fine-tuning with LoRA and DoRA does not help. Upon exploring the addition of dropout from amongst the following four values: {0, 0.05, 0.10, 0.20} while fine-tuning Gemma 2B, we find that adding any amount of dropout leads to a comparable/slightly-lowered performance with/than the case when no dropout is used. Hence, we do not use dropout while fine-tuning models using LoRA and DoRA.

## D Hyperparameters for Decoding

The values of the hyperparameters used for decoding the LLM-generated outputs are provided in Table 8 and have been taken *as-is* from (Hu et al., 2023) so as to maintain consistency with prior work.

Table 6: Hyperparameter configurations of LoRA used while fine-tuning LLaMA3 8B and Gemma 2B models. * indicates that Effective Batch Size = (Per-GPU Batch Size $\times$ Gradient Accumulation Steps $\times$ No. of GPUs).

| Hyperparameters | LLaMA3 8B | Gemma 2B |
|---|---|---|
| Optimizer | Fused AdamW ($\beta_1 = 0.9$; $\beta_1 = 0.999$) | |
| Weight Decay | 0.01 | |
| Learning Rate | 3e-4 | |
| Learning Rate Scheduler | Cosine | |
| Warmup Steps | 100 | |
| Effective Batch Size* | 16 | |
| Per-GPU Batch Size | 4 | |
| Gradient Accumulation Steps | 1 | 2 |
| No. of GPUs | 4 | 2 |
| Epochs | 3 | |
| Target Modules | q_proj, k_proj, v_proj, up_proj, down_proj | |

Table 7: Hyperparameter configurations of DoRA used while fine-tuning LLaMA3 8B and Gemma 2B models. * indicates that Effective Batch Size = (Per-GPU Batch Size $\times$ Gradient Accumulation Steps $\times$ No. of GPUs).

| Hyperparameters | LLaMA3 8B | Gemma 2B |
|---|---|---|
| Optimizer | Fused AdamW ($\beta_1 = 0.9$; $\beta_1 = 0.999$) | |
| Weight Decay | 0.01 | |
| Learning Rate | 1e-4 | |
| Learning Rate Scheduler | Cosine | |
| Warmup Steps | 100 | |
| Effective Batch Size* | 16 | |
| Per-GPU Batch Size | 4 | |
| Gradient Accumulation Steps | 1 | 2 |
| No. of GPUs | 4 | 2 |
| Epochs | 3 | |
| Target Modules | q_proj, k_proj, v_proj, up_proj, down_proj | |

Table 8: Hyperparameters for decoding the output generated by the fine-tuned models into text. Values have been taken as-is from (Hu et al., 2023).

| Hyperparameters | Values |
|---|---|
| Temperature $T$ | 0.1 |
| $k$ (as in top-$k$) | 40 |
| $p$ (as in top-$p$) | 0.75 |
| Number of Beams (as in Beam Search) | 4 |

# E PER-RUN RESULTS OF OUR EXPERIMENTS

## E.1 ID AND OOD ACCURACY

For LLaMA3 8B, models fine-tuned on the ARC-c and ARC-e datasets using GaLLoP form a dominant Pareto front for both ID as well as OOD performance over models fine-tuned and/or edited on the same datasets using the other algorithms (respectively; see Figure 8(a) and Figure 8(b)). While models fine-tuned on the HellaSwag, OBQA, PIQA, and WinoGrande datasets using SpIEL outperform those fine-tuned on the same datasets using GaLLoP by a slight margin on the ID test sets, models fine-tuned with GaLLoP form a dominant Pareto front on the corresponding OOD test sets on which the models fine-tuned with SpIEL perform quite poorly and with a performance drop that widens as we approach the higher density levels (respectively; see Figure 8(c), Figure 8(d), Figure 8(e), and Figure 8(f)). On the SIQA dataset, the model fine-tuned with SAFT is quite competitive and has an almost similar/comparable level of performance with the model fine-tuned with GaLLoP (see Figure 8(g)).

For Gemma 2B, since pre-training has been performed on a much less number of tokens (= 6T) as compared to LLaMA3 8B (= 15T), there is a significantly lower risk of fine-tuning sensitivity due to overtraining (Springer et al., 2025). Hence, Gemma 2B models fine-tuned with some of the competing PEFT algorithms (LoRA, DoRA, SpIEL) and even FFT (and hence, WiSE-FT and LiNeS) perform much better than their LLaMA3 8B counterparts. However, these models still perform quite poorly when it comes to OOD tasks and display high performance drops as we approach the higher density levels. These drops in performance are not only present for the OOD tasks (LoRA, DoRA, and SpIEL) but are also present for the ID tasks (LoRA and DoRA). Moreover, an important trend is noticeable across all these datasets: overfitting across all the density levels with the increase in the size of the dataset used for fine-tuning. While models fine-tuned with LoRA and DoRA attain higher accuracies as compared to models fine-tuned with GaLLoP on both ID and OOD tasks upon utilizing small datasets (ARC-c and ARC-e; models fine-tuned with LoRA already start exhibiting large OOD performance drops upon increasing the density level for fine-tuning on ARC-e) for fine-tuning (see Figure 9), they tend to overfit and break down with drastic OOD performance drops along with even moderately-high/high ID performance drops for higher density levels as move towards utilizing moderately-large sized datasets (OBQA, BoolQ, PIQA, SIQA, HellaSwag; see Figure 10, Figure 11(a), and Figure 11(b)) and completely break down on both the ID and OOD tasks upon fine-tuning on WinoGrande, the largest dataset (see Figure 11(c)). In contrast, models fine-tuned with GaLLoP demonstrate high and stable performance across all density levels and upon fine-tuning on all the datasets (*individually*) with sizes spread across the spectrum, and form a dominant Pareto front over all the other fine-tuned models when fine-tuning is performed on WinoGrande, the largest dataset (see Figure 9, Figure 10, and Figure 11). This shows that fine-tuning with GaLLoP makes models the most robust to overfitting. Finally, on shifting our focus to models fine-tuned with SAFT, we can clearly see that on average (across all datasets: see Figure 3(b)) as well as individually, across seven out of the eight datasets (see Figure 9, Figure 10(a), Figure 10(c), and Figure 11), models fine-tuned with GaLLoP form a dominant Pareto front over models fine-tuned with SAFT.

Finally, for both LLaMA3 8B and Gemma 2B, an important case to note is that of fine-tuning on the BoolQ dataset. While the LLaMA3 8B (except for the highest density level) and Gemma 2B models fine-tuned with GaLLoP consistently outperform the models fine-tuned with SAFT on the ID (BoolQ) test set, the models fine-tuned with SAFT consistently outperform (except for the first two density levels for LLaMA3 8B) the models fine-tuned with GaLLoP on the corresponding OOD test sets (see Figure 8(h) and Figure 10(b)). This might potentially be due to the heavily skewed distribution of the correct responses for BoolQ in its own test set ('True': 62.171%; 'False': 37.829%) which is in consonance with the correct response distribution in its training set ('True': 62.304%; 'False': 37.686%); 'True' is roughly about twice as frequent as 'False' for BoolQ. On analyzing the generated responses, we find that models fine-tuned with GaLLoP consistently try to learn what characterizes a 'False' response whereas models fine-tuned with SAFT *undesirably* generate 'True' as the most frequent response, *albeit incorrectly* (their ID accuracies are generally lower than those of the models fine-tuned with GaLLoP), possibly due to the near-perfect memorization of 'True' as the correct response. For LLaMA3 8B, fine-tuning with GaLLoP leads to a huge drop in the prediction rate of 'True' (from 66.391% for $\rho = 0.93\%$ to 47.073% for $\rho = 1.85\%$) while fine-tuning with SAFT leads to a fairly constant and high prediction rate of 'True' (80.764% for $\rho = 0.93\%$

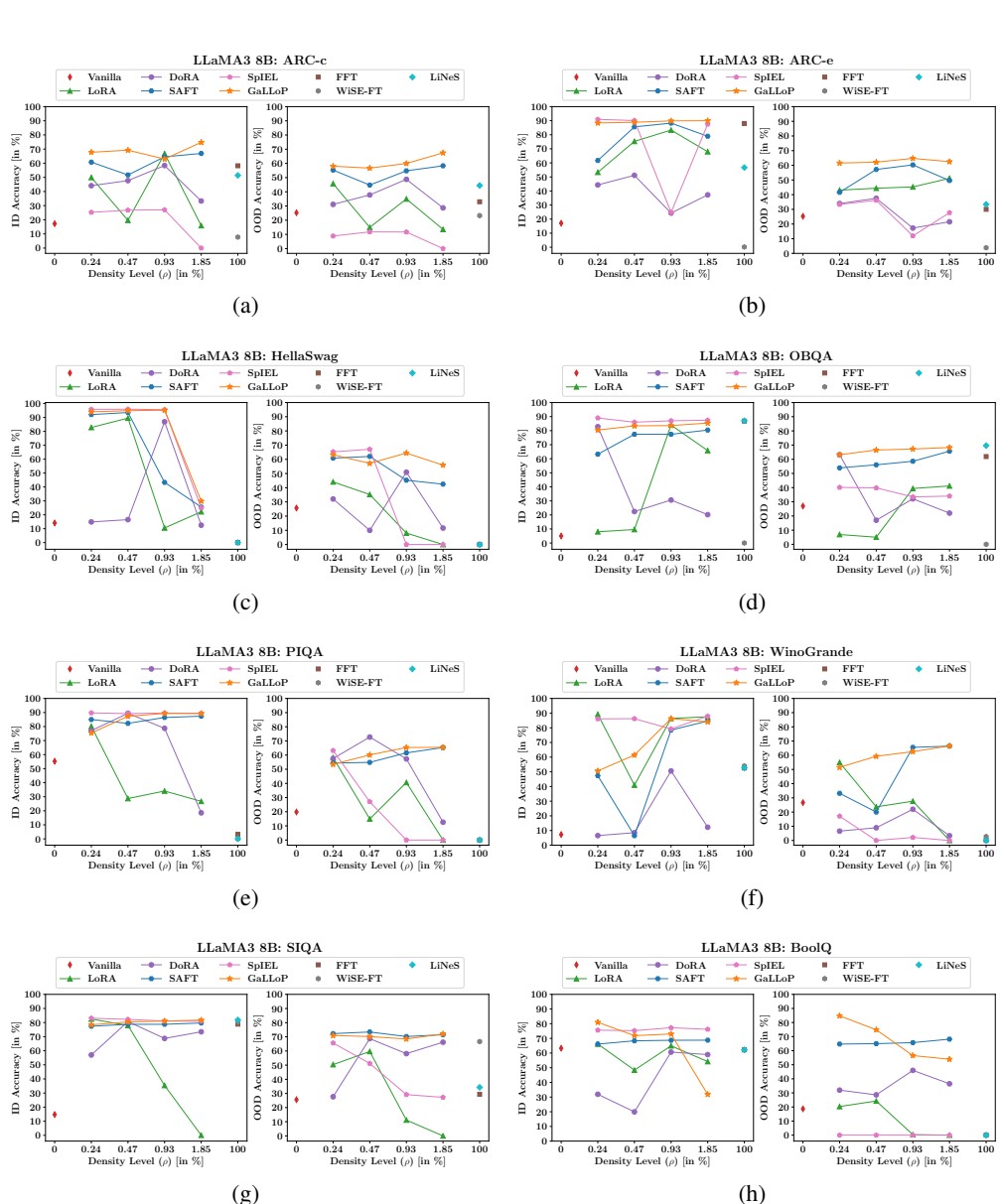

Figure 8: LLaMA3 8B models fine-tuned with GaLLoP attain the most stable and highest/high levels of ID and OOD performance across all density levels when fine-tuning is performed on a) ARC-c, b) ARC-e, c) HellaSwag, d) OBQA, e) PIQA, f) WinoGrande, g) SIQA, and h) BoolQ (except OOD).

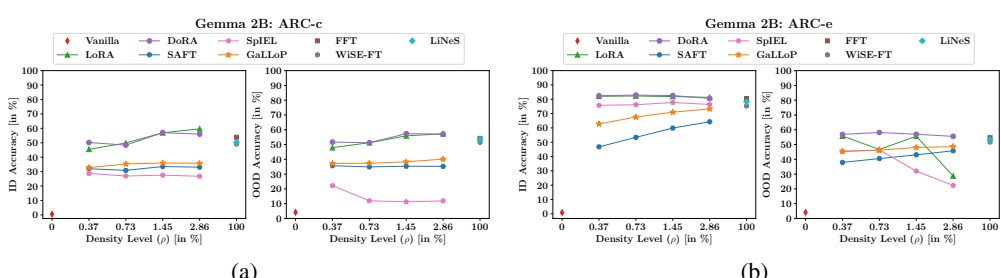

Figure 9: While Gemma 2B models fine-tuned on a) ARC-c and b) ARC-e with GaLLoP attain stable and high levels of ID and OOD performance across all density levels, Gemma 2B models fine-tuned on the same datasets with DoRA attain the highest levels of ID and OOD performance across all density levels owing to the two training sets possessing the smallest sizes.

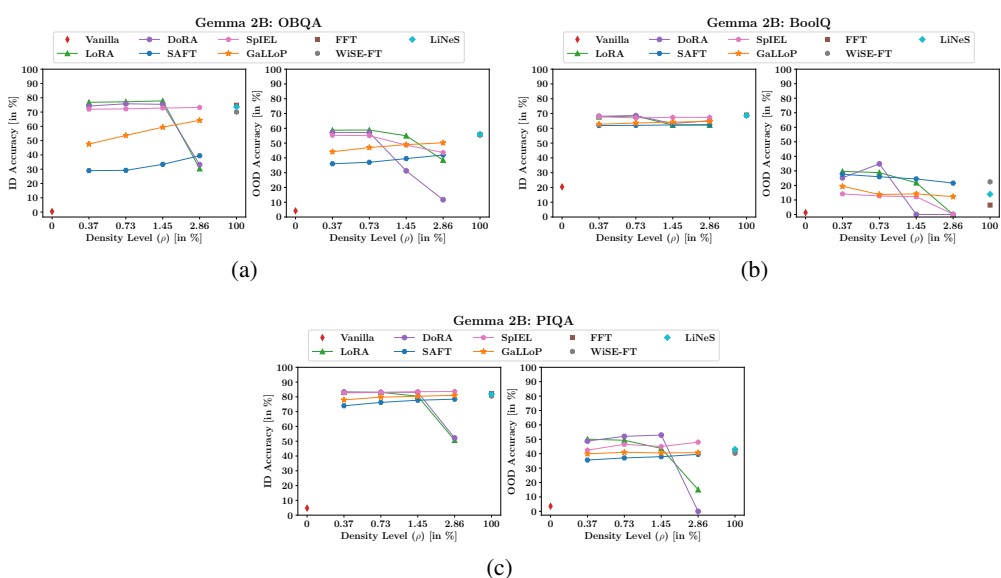

Figure 10: While Gemma 2B models fine-tuned on a) OBQA, b) BoolQ, and c) PIQA with GaLLoP attain stable and high levels of ID and OOD performance across all density levels, Gemma 2B models fine-tuned with LoRA and DoRA tend to overfit on their moderately-large sized training sets and show drastic ID and/or OOD performance drops.

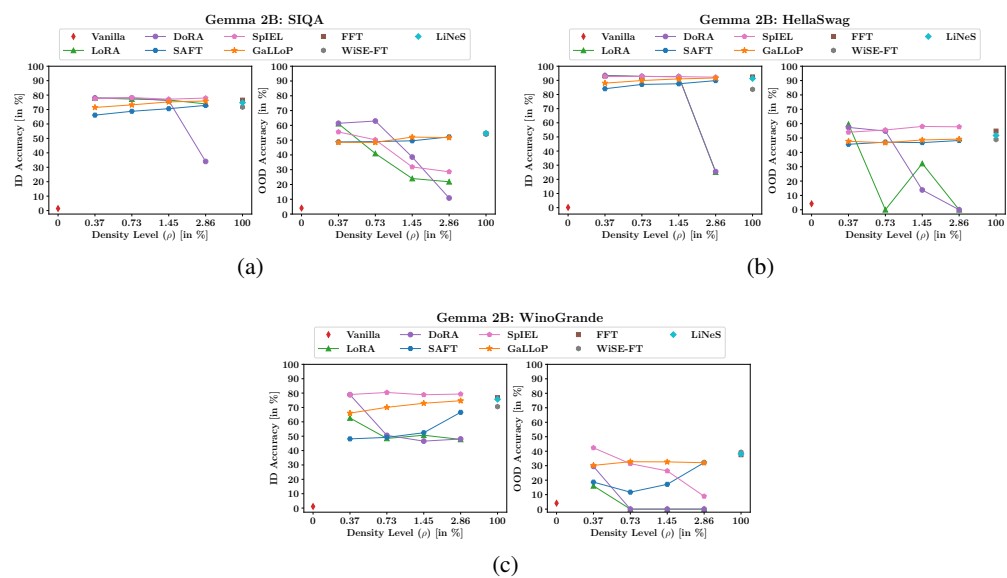

(a)  (b)

(c)

Figure 11: While Gemma 2B models fine-tuned on a) SIQA, b) HellaSwag, and c) WinoGrande with GaLLoP attain the most stable and the highest/high levels of ID and OOD performance across all density levels, Gemma 2B models fine-tuned with LoRA or DoRA tend to overfit on their moderately-large/large sized training sets and show drastic ID and/or OOD performance drops.

and 80.367% for $\rho = 1.85\%$). For Gemma 2B, such a trend can be seen across all density levels in Figure 12.

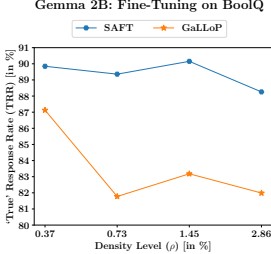

Figure 12: Gemma 2B models fine-tuned with GaLLoP on the highly skewed BoolQ dataset increasingly learn what characterizes a 'False' response while those fine-tuned with SAFT frequently generate 'True' as the response potentially due to the memorization of the dominant correct response in BoolQ.

While memorizing and frequently generating 'True' allows models fine-tuned with SAFT to easily attain a constant, moderate/moderately-low level OOD accuracy, learning what makes a response 'False' seems to require harder effort on the part of models fine-tuned with GaLLoP and they increasingly start to respond with something different than the response format for the OOD tasks: either they generate the answer in words rather than the corresponding option number, or generate an explanation to the answer, or even generate the question itself, or etc. We leave the further examination of this phenomenon for future work.

### E.2 CATASTROPHIC FORGETTING

The per-run forget ratios are shown in Figure 14 for fine-tuned and/or edited LLaMA3 8B models and in Figure 13 for fine-tuned and/or edited Gemma 2B models.

The fact that fine-tuning LLaMA3 8B models with the runners-up algorithm SAFT results in instability with sudden and high rises/falls in ID and/or OOD performance on all datasets (except BoolQ; see Figure 8) and moderately-high catastrophic forgetting for $\rho = 0.47\%$ on the largest dataset (WinoGrande) (see Figure 14(f)) while fine-tuning them with GaLLoP does not — reveals the benefits of GaLLoP's dual parameter selection criterion. Restricting the selection of high gradient parameters to those with the smallest pre-trained magnitudes leads to increased stability, relatively balanced ID as well as OOD performance, and prevents catastrophic forgetting across all datasets even in the overtrained regime.

Interestingly, upon fine-tuning Gemma 2B models individually on four datasets (ARC-c, ARC-e, OBQA, and SIQA), no catastrophic forgetting occurs (and hence those plots are not shown here). While this follows, *in part*, from the explanation provided in Section 6.2, why this specifically happens for only these four datasets is discussed in detail in the next appendix on memorization (Appendix F).

Finally, as a follow-up to the discussion on the near-zero, vanilla (zero-shot) performance of Gemma 2B as compared to the relatively higher vanilla (zero-shot) performance of LLaMA3 8B (see Section 6.2), we analyze the responses of the vanilla Gemma 2B and LLaMA3 8B models. Consequently, we observe that while the vanilla LLaMA3 8B model does respond with an answer/answers adhering to the response format (on most datasets), the vanilla Gemma 2B model, in most cases, does not respect the response format while responding or simply repeats the entire question or repeats all the answer choices verbatim from the question used for prompting it. A representative example of the responses generated by both the vanilla models is shown in Figure 15.

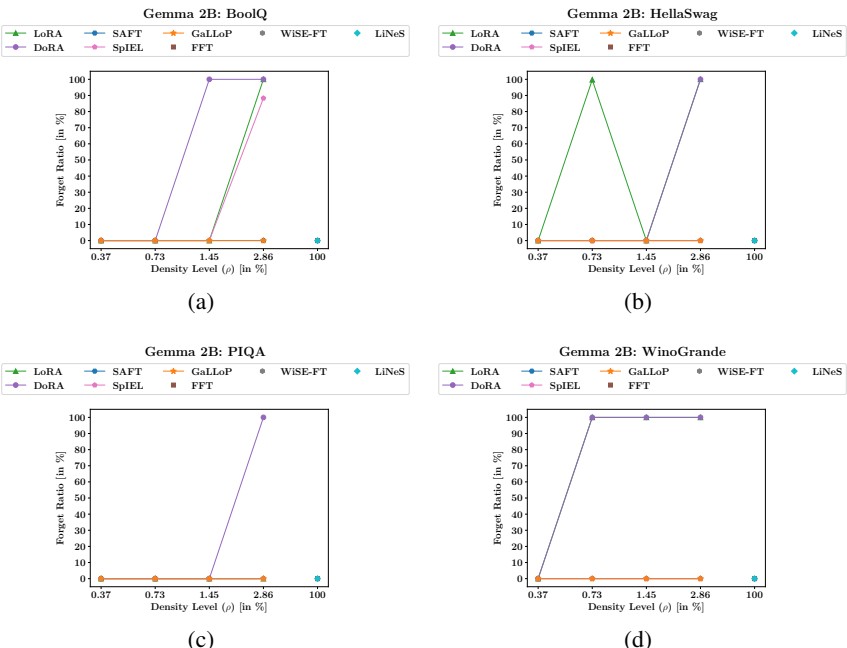

Figure 13: Gemma 2B models fine-tuned with GaLLoP show a 0% forget ratio across all density levels when fine-tuning is performed on a) BoolQ, b) HellaSwag, c) PIQA, and d) WinoGrande.

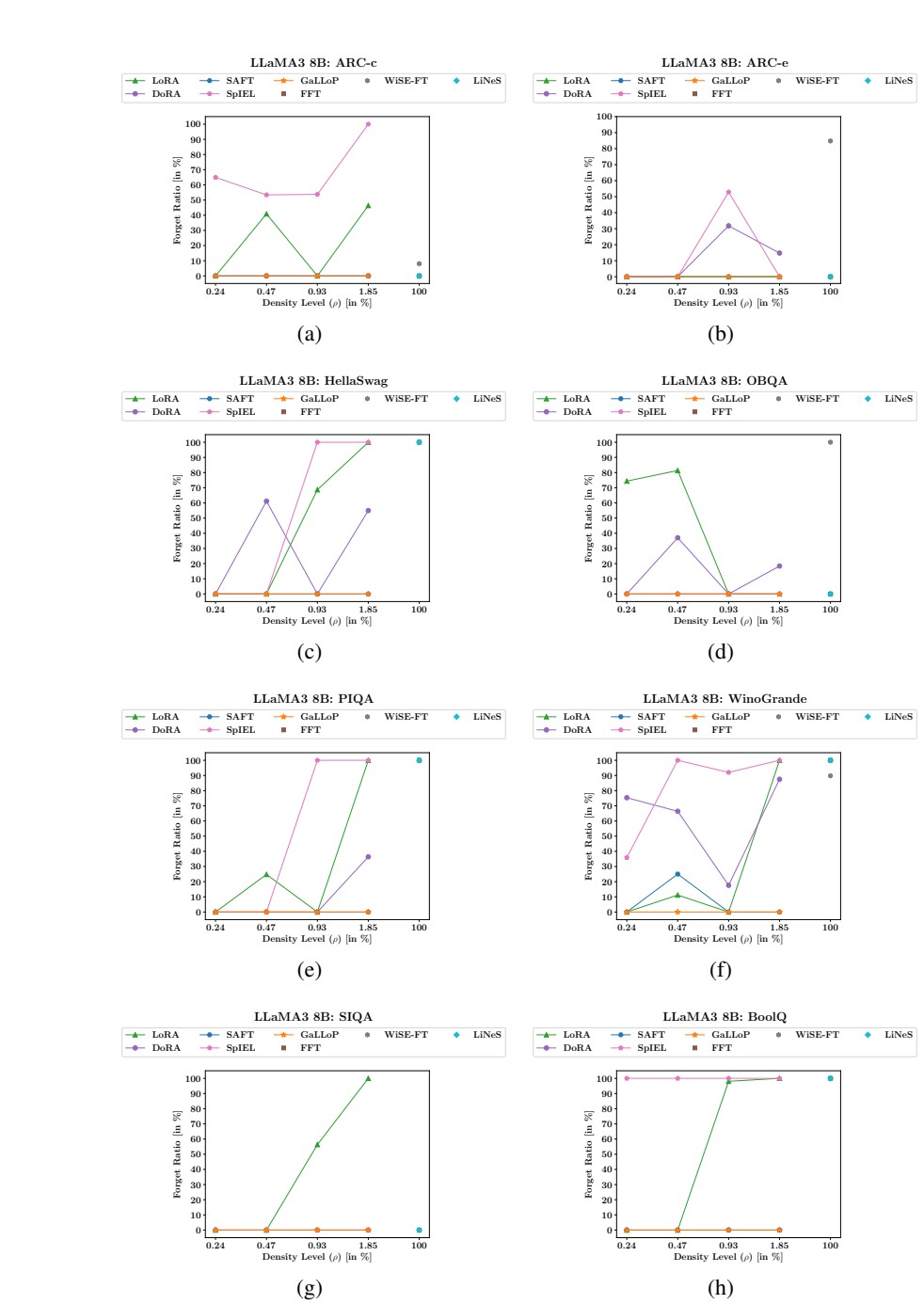

Figure 14: LLaMA3 8B models fine-tuned with GaLLoP show a 0% forget ratio across all density levels when fine-tuning is performed on a) ARC-c, b) ARC-e, c) HellaSwag, d) OBQA, e) PIQA, f) WinoGrande, g) SIQA, and h) BoolQ.

---

**Zero-Shot (Vanilla) Models**

**Prompt (Example from PIQA):**
Below is an instruction that describes a task. Write a response that appropriately completes the request.

### Instruction:
Please choose the correct solution to the question: How can I raise and lower a picavet with a camera attached so it can be flown with a kite?

Solution1: You would implement a pulley system with wheels and strong string.

Solution2: You would implement a pulley system with carabiners and strong string.

Answer format: solution1/solution2

### Response:

- - - - - - - - - - - - - - - - - - - - - - - - - - - - - - - - - - - - - - - - - -

**Reference Response:** the correct answer is solution2

**LLaMA3 8B**

**Response:** solution1

**Gemma 2B**

**Response:** I would implement a pulley system with wheels and strong string.

I would implement a pulley system with carabiners and strong string.

---

Figure 15: The vanilla LLaMA3 8B model generates an answer when prompted with a question from the PIQA dataset while the vanilla Gemma 2B model generates both answers without respecting the answer format when prompted with the same question.

# F    MEMORIZATION

Collapse rates and forget ratios do not always reveal the same information; rather, they actually reveal *complementary* information. For instance, even though the LLaMA3 8B models fine-tuned with DoRA possess high forget ratios varying between 10-20% (see Figure 4(a)), they do not undergo extremely high memorization and hence, showcase low collapse rates (see Figure 5(a)). Additionally, while Gemma 2B models fine-tuned with SpIEL (for the first three density levels) and FFT possess 0% forget ratios (see Figure 4(b)), they do undergo memorization (see Figure 5(b)). Decrease in the extent of memorization on fine-tuning LLaMA3 8B models with DoRA as compared to fine-tuning them with LoRA is likely due to the former's fine-grained (magnitude and directional) reparametrized update of the overtrained parameter matrix which imparts higher learning capabilities than the latter. Nevertheless, since DoRA is still an RFT technique, all the trainable parameters introduced by it are fine-tuned on the training set which then leads to catastrophic forgetting and moderate levels of memorization. 0% forget ratios versus high collapse rates of Gemma 2B models fine-tuned with SpIEL and FFT follows from our previous discussion on the performance of the vanilla Gemma 2B model (see Section 6.2). Near-zero performance of the vanilla Gemma 2B model along with the near-zero performance of the same model fine-tuned with SpIEL and FFT yields a 0% forget ratio in line with Equation 6. However, an analysis of the responses generated by both of them (vanilla and fine-tuned) reveal striking differences between them with the latter consistently responding with an EOS token (as discussed in Section 6.3).

When models memorize the response formats of the dataset used for fine-tuning (see Table 2 in Appendix B for response formats), it means that while a model fine-tuned on the training set of any one of the following datasets: ARC-c, ARC-e, OBQA, and SIQA (which share the same response format), is increasingly likely to perform well on the test sets of the remaining three OOD datasets (note that this is also the reason why forget ratios equal zero on these datasets for the fine-tuned Gemma 2B models: see Section E.2), models fine-tuned on any one of the following datasets: BoolQ, HellaSwag, PIQA, and WinoGrande (which possess response formats completely orthogonal to the other datasets), increasingly fail on the test sets of all the corresponding seven OOD datasets.

While repetition of the most frequently occurring words/phrases in the fine-tuning dataset is partly due to the maximum likelihood-based fine-tuning objective (Welleck et al., 2020b), and might also arise due to an unfortunate set of parameters selected for combined deterministic beam-search (Graves, 2012) and random (top-$k$ (Fan et al., 2018) and top-$p$ (Holtzman et al., 2020)) sampling (Shaham & Levy, 2022; Welleck et al., 2020a; Borec et al., 2024) which makes the repetition-encouraging nature dominant during decoding (this is solely because of the parameter values defined by Hu et al. (2023) and enlisted in Table 8 in Appendix D), its occurrence seems to also depend upon the type of fine-tuning employed since it only occurs on performing RFT and not SpFT. Following the discussion in Section 6.3, *concerted overfitting*, along with the detrimental effects of maximum likelihood-based fine-tuning, only gets exacerberated in the overtrained regime (Springer et al., 2025) which is why this is only observed for LLaMA3 8B (and not Gemma 2B) models.

While the generation of the EOS token might partly be due to the dominant effect of beam search (over random sampling) in the overtrained regime which consequently might cause high probablities to be assigned to the shortest hypotheses with just the EOS token (Stahlberg & Byrne, 2019), the fact that it only occurs upon fine-tuning LLaMA3 8B models with SpIEL and FFT must also be noted and hence, we attribute it to the tendency of these algorithms to cause instability in the learning dynamics in the overtrained regime.

Representative samples of responses generated by such models, showcasing all the aforementioned forms of memorization, against those generated by models fine-tuned with GaLLoP and SAFT can be seen in Figure 16, Figure 17, Figure 18, and Figure 19.

---

**Gemma 2B Models Fine-Tuned on WinoGrande with $\rho = 0.73\%$**

**Prompt (Example from ARC-e):**
Below is an instruction that describes a task. Write a response that appropriately completes the request.

### Instruction:
Please choose the correct answer to the question: A student throws a ball into the air. While the ball travels up, the speed of the ball decreases. What force causes the ball to slow while traveling up?

Answer1: electricity Answer2: gravity Answer3: magnetism Answer4: tension

Answer format: answer1/answer2/answer3/answer4

### Response:

- - - - - - - - - - - - - - - - - - - - - - - - - - - - - - - - - - - - - - - - - - - - - -

**Reference Response:** the correct answer is answer2

**LoRA**

**Response:** the correct answer is option1

**DoRA**

**Response:** the correct answer is option1

**SAFT**

**Response:** the correct answer is 'answer2'

**SpIEL**

**Response:** the correct answer is answer2

**GaLLoP**

**Response:** the correct answer is answer2

**FFT**

**Response:** the correct answer is answer2

**WiSE-FT**

**Response:** the correct answer is answer2

**LiNeS**

**Response:** the correct answer is answer2

Figure 16: Gemma 2B models fine-tuned on WinoGrande with GaLLoP, SAFT, SpIEL, FFT and/or edited with WiSE-FT and LiNeS respect ARC-e's response format whereas those fine-tuned with LoRA and DoRA do not.

---

**Gemma 2B Models Fine-Tuned on SIQA with $\rho = 1.45\%$**

**Prompt (Example from HellaSwag):**
Below is an instruction that describes a task. Write a response that appropriately completes the request.

### Instruction:
Please choose the correct ending to complete the given sentence: Having an ice cream: Children bring desert out for their family member. the family

Ending1: floats in a river. Ending2: member stands looking into a hut and then handing people photographs. Ending3: member cuts a piece of sunscreen. Ending4: enjoys eating the desert together.

Answer format: ending1/ending2/ending3/ending4

### Response:

- - - - - - - - - - - - - - - - - - - - - - - - - - - - - - - - - - - - - - - - - - - - - - - - - - -

**Reference Response:** the correct answer is ending4

**LoRA**

**Response:** the correct answer is answer4

**DoRA**

**Response:** the correct answer is answer4

**SAFT**

**Response:** the correct answer is ending4

**SpIEL**

**Response:** the correct answer is answer4

**GaLLoP**

**Response:** the correct answer is ending4

**FFT**

**Response:** the correct answer is answer3

**WiSE-FT**

**Response:** the correct answer is answer4

**LiNeS**

**Response:** the correct answer is answer4

---

Figure 17: Gemma 2B models fine-tuned on SIQA with GaLLoP and SAFT respect HellaSwag's response format whereas those fine-tuned and/or edited with the other competing algorithms do not.

---

**LLaMA3 8B Models Fine-Tuned on ARC-c with $\rho = 1.85\%$**

**Prompt (Example from OBQA):**
Below is an instruction that describes a task. Write a response that appropriately completes the request.

### Instruction:
Please choose the correct answer to the question: As the rain forest is deforested the atmosphere will increase with

Answer1: oxygen Answer2: nitrogen Answer3: carbon Answer4: rain

Answer format: answer1/answer2/answer3/answer4

### Response:

- - - - - - - - - - - - - - - - - - - - - - - - - - - - - - - - - - - - - - - - - - - - - - - -

**Reference Response:** the correct answer is answer3

**LoRA**

**Response:** the correct answer is answer3assistantassistant

the correct answer is answer3

assistant

the correct answer is answer

**DoRA**

**Response:** the correct answer is answer3assistant the correct answer is answer3assistant the correct answer is

**SAFT**

**Response:** the correct answer is answer3

**SpIEL**

**Response:** < EOS >

**GaLLoP**

**Response:** the correct answer is answer3

**FFT**

**Response:** the correct answer is answer3

**WiSE-FT**

**Response:** the correct answer is answer4

**LiNeS**

**Response:** the correct answer is answer3

Figure 18: LLaMA3 8B models fine-tuned on ARC-c with GaLLoP, SAFT, FFT and/or edited with WiSE-FT and LiNeS respect OBQA's response format whereas those fine-tuned with LoRA, DoRA, and SpIEL either generate repetitive text or do not respond (except with the EOS token; < EOS >).

**LLaMA3 8B Models Fine-Tuned on PIQA with $\rho = 1.85\%$**

**Prompt (Example from BoolQ):**
Below is an instruction that describes a task. Write a response that appropriately completes the request.

### Instruction:
Please answer the following question with true or false, question: is northern ireland part of the great britain?

Answer format: true/false

### Response:

---

**Reference Response:** the correct answer is false

**LoRA**

**Response:** the correct answer is solution2assistant

the correct answer is solution1assistant

the correct answer is solution2assistant

the correct

**DoRA**

**Response:** the correct answer is trueassistant

the correct answer is trueassistant

the correct answer is true

**SAFT**

**Response:** the correct answer is true, northern ireland is part of the great britain.

the correct answer is true, northern ireland is part of the great

**SpIEL**

**Response:** the correct answer is solution

**GaLLoP**

**Response:** the correct answer is false

**FFT**

**Response:** < EOS >

**WiSE-FT**

**Response:** < EOS >

**LiNeS**

**Response:** < EOS >

Figure 19: LLaMA3 8B models fine-tuned on PIQA with GaLLoP and SAFT respect BoolQ's response format whereas those fine-tuned and/or edited with the other competing algorithms either do not respect the response format and/or generate repetitive text or do not respond (except with the EOS token; < EOS >).

## G  VARIABILITY IN FORGET RATIOS AND COLLAPSE RATES

Figure 20(a) and Figure 20(b) show the boxplots of forget ratios and collapse rates (respectively) for LLaMA3 8B models fine-tuned with GaLLoP against models fine-tuned with the competing algorithms across 20 different random seeds.

Models fine-tuned with GaLLoP and SAFT consistently attain 0% forget ratios and 0% collapse rates. In contrast, models fine-tuned with SpIEL consistently attain high forget ratios (80 - 100%) and high collapse rates mostly equivalent to a complete failure on all the seven OOD datasets. The extent of variability in their forget ratios and collapse rates is also quite high and never allows them to perform on par/even close to models fine-tuned with GaLLoP on OOD tasks. These observations can possibly be owed to SpIEL's dynamic parameter selection: variability in random seeds can lead to changes in the candidate parameter set and hence, different sets of parameters being considered for updates at each reselection stage. Models fine-tuned with the RFT techniques also possess high forget ratios and high collapse rates (LoRA and FFT) and/or with high variance in their values (LoRA, DoRA, and FFT) owing to an absence of regularization during fine-tuning.

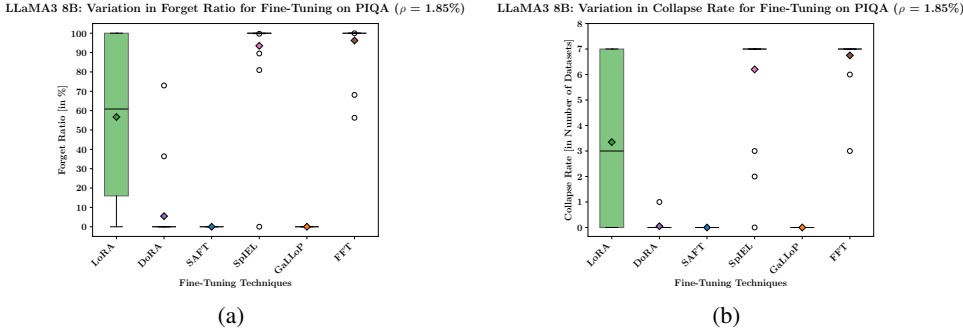

Figure 20: LLaMA3 8B models fine-tuned with GaLLoP consistently attain a) 0% forget ratios and b) 0% collapse rates across 20 different random seeds upon fine-tuning on the PIQA dataset with the highest density level ($\rho = 1.85\%$).

## H  ABLATIONS ON PARAMETER SELECTION CRITERIA

To investigate whether GaLLoP's dual parameter selection criterion (parameters with high gradients and low-magnitudes) potentially leads to performance benefits over selecting parameters satisfying only one of these two criteria, we conduct an ablation study on Gemma 2B models. Figure 21 shows the results of this study with the corresponding averaged ID and OOD accuracies which are reported for all the four density levels. The per-run performance results are shown in Figure 22.

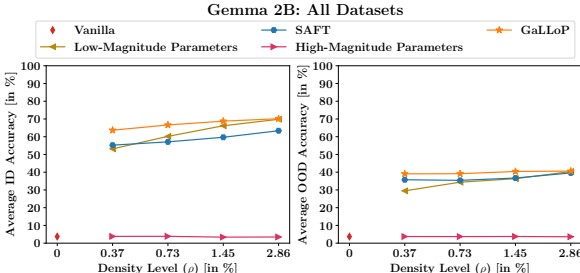

Figure 21: Gemma 2B models fine-tuned with GaLLoP form a dominant Pareto front for both ID and OOD performance (on average) over models fine-tuned by selecting parameters on the basis of either parameter or gradient magnitudes.

Fine-tuning with GaLLoP allows for the attainment of the highest ID and OOD accuracies: both on average and across all the datasets (except OOD accuracies obtained upon fine-tuning on BoolQ: see Figure 22(h) and Section E.1 in Appendix E for an explanation). Further, as can also be seen from Figure 21 and as discussed in Section 3, fine-tuning high-magnitude parameters leads to no improvements over the performance of the non fine-tuned, vanilla counterpart. In contrast to this, fine-tuning low-magnitude parameters allows models to perform quite well on both ID and OOD tasks. In fact, except for the OOD performance obtained upon fine-tuning with BoolQ (see Figure 22(h); the explanation from Section E.1 in Appendix E applies here as well, *albeit* with lesser OOD performance metrics than the model fine-tuned with GaLLoP), ID and OOD performance levels obtained on fine-tuning low-magnitude parameters interpolate between the corresponding performance levels obtained on fine-tuning parameters with high gradients (as in SAFT; generally forms the lower bound) and fine-tuning low-magnitude parameters with high gradients (as in GaLLoP; forms the upper bound): see Figure 22.

Moreover, with increasing density levels, performance obtained on fine-tuning low magnitude parameters generally overtakes that obtained on fine-tuning parameters with high gradients (as in SAFT) and finally, saturates at a performance level comparable to that obtained on fine-tuning with GaLLoP. This illustrates how GaLLoP potentially operates in different density regimes to yield the most superior performance: while the high gradient parameter selection criterion might dominate at lower density levels, the low-magnitude parameter selection criterion becomes increasingly dominant as the density level increases.

Further, as mentioned earlier (see Section 6.1), *low-magnitude dilution* kicks in as we move closer towards the highest density level because of which we observe a somewhat saturation and convergence in the performance attained by models fine-tuned with GaLLoP and those fine-tuned by selecting parameters with low-magnitudes. This trend of saturation is also reflected in models fine-tuned with SAFT which suggests that *high gradient dilution* starts to set in at that point as well. Effectively, in the asymptotic limit ($\rho \rightarrow 100\%$), as what can also be expected theoretically, the performance of models fine-tuned with static SpFT algorithms such as GaLLoP and SAFT, and models fine-tuned by selecting parameters with low-magnitudes, will converge to the performance of models fine-tuned with the dense FFT algorithm. This hypothesis is also supported by several previous studies wherein it was found that FFT predominantly updates a small fraction of low magnitude parameters for downstream task(s) and utilization of SpFT should be able to recover FFT performance (Chen et al., 2020; Prasanna et al., 2020; Jaiswal et al., 2023; Ramesh et al., 2024). Nevertheless, as can be seen from our experiments, by combining the strengths of high gradients and low-magnitudes, fine-tuning with GaLLoP is expected to continue forming a performance upper bound over other static SpFT algorithms.

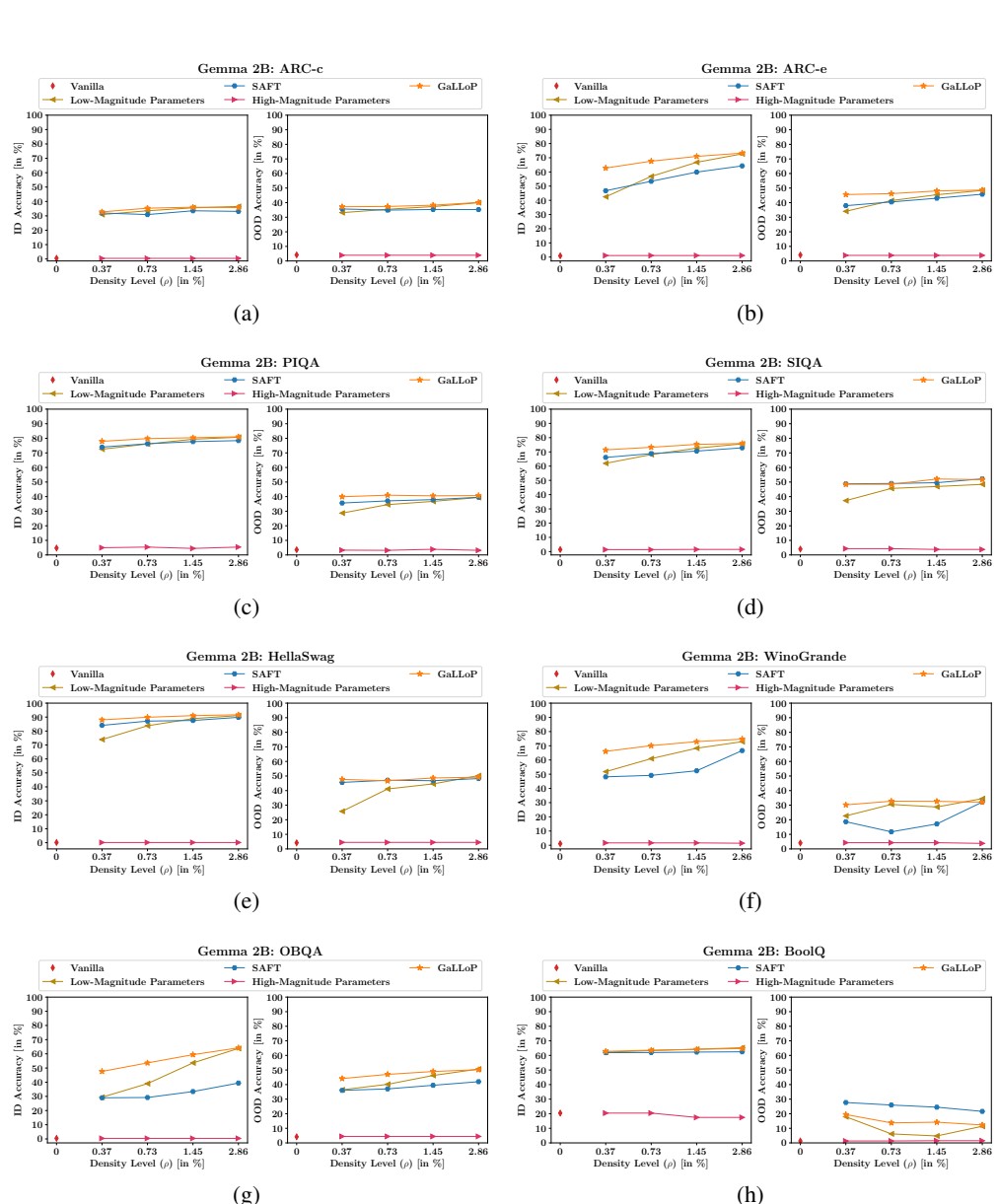

Figure 22: Gemma 2B models fine-tuned with GaLLoP form a dominant Pareto front for both ID and OOD performance over models fine-tuned by selecting parameters on the basis of either parameter or gradient magnitudes when fine-tuning is performed on a) ARC-c, b) ARC-e, c) PIQA, d) SIQA, e) HellaSwag, f) WinoGrande, g) OBQA, and h) BoolQ (except OOD).

# I   THEORY

In this section, we provide theoretical justifications for the dual parameter selection criteria of our SpFT algorithm, GaLLoP.

Consider a model with a parameter vector $\boldsymbol{\theta} \in \mathbb{R}^D$ which has been pre-trained on a dataset $\mathcal{D}^p = \{(\boldsymbol{x}_n^p, \boldsymbol{y}_n^p)\}_{n=1}^N$, where, $\boldsymbol{x}_n^p \in \mathbb{R}^P$ and $\boldsymbol{y}_n^p \in \mathbb{R}^Q$. This model needs to be fine-tuned on a dataset $\mathcal{D}^f = \{(\boldsymbol{x}_n^f, \boldsymbol{y}_n^f)\}_{n=1}^N$, where, $\boldsymbol{x}_n^f \in \mathbb{R}^P$ and $\boldsymbol{y}_n^f \in \mathbb{R}^Q$. Given a density level $\rho$ indicating the fraction of model parameters to be fine-tuned, the aim of our SpFT algorithm is to enhance the ID as well as the OOD generalizability of our model.

Accordingly, fine-tuning a model with our SpFT algorithm must not only minimize the model's loss on the downstream task (for enhancing ID generalizability) but at the same time, also preserve and enhance the pre-trained knowledge stored in the model parameters (for enhancing OOD generalizability).

Given that the model was pre-trained with a loss function $\mathcal{L}^p(\boldsymbol{x}_n^p, \boldsymbol{y}_n^p; \boldsymbol{\theta})$ and will now be fine-tuned with a loss function $\mathcal{L}^f(\boldsymbol{x}_n^f, \boldsymbol{y}_n^f; \boldsymbol{\theta})$, fine-tuning the model with GaLLoP must therefore lead to the maximal change in $\mathcal{L}^f(\boldsymbol{x}_n^f, \boldsymbol{y}_n^f; \boldsymbol{\theta})$ so as to achieve the fastest convergence to the downstream task optimum while at the same time resulting in a minimal change in $\mathcal{L}^p(\boldsymbol{x}_n^p, \boldsymbol{y}_n^p; \boldsymbol{\theta})$ so as to preserve the pre-trained knowledge possessed by the model.

Accordingly, our multi-objective optimization problem is as follows:

$$\min_{\boldsymbol{\theta}} \left( -\Delta\mathcal{L}^f(\boldsymbol{x}_n^f, \boldsymbol{y}_n^f; \boldsymbol{\theta}), \Delta\mathcal{L}^p(\boldsymbol{x}_n^p, \boldsymbol{y}_n^p; \boldsymbol{\theta}) \right)$$
$$\text{such that } \|\Delta\boldsymbol{\theta}\|_0 = \lfloor \rho D \rfloor, \tag{14}$$

where, $\|.\|_0$ denotes the $\mathbb{L}_0$ norm.

A direct analytical solution of the above problem is difficult due to the non-convexity of the loss functions but it is possible to obtain an approximate solution by analyzing the influence of model parameters and/or their gradients on the two loss functions individually.

## I.1   CRITERION 1: ENHANCEMENT OF ID GENERALIZABILITY

Following the principle of gradient descent (Cauchy, 1847), parameters with the largest gradient magnitudes yield the fastest convergence on the downstream task and hence, maximize $\Delta\mathcal{L}^f(\boldsymbol{x}_n^f, \boldsymbol{y}_n^f; \boldsymbol{\theta})$, as is desired.

Nevertheless, it is interesting to examine this rationale behind our first selection criterion from another viewpoint: one that concerns the Fisher information matrix.

For a conditional probability distribution $p_{\boldsymbol{\theta}}(\boldsymbol{y}^f|\boldsymbol{x}^f)$ generated by the model over the output $\boldsymbol{y}^f$, given an input $\boldsymbol{x}^f$, one way to estimate the importance of each of its parameters is to determine the amount by which the model's prediction would change upon perturbing its parameters by a small amount ($\Delta\boldsymbol{\theta}$), i.e., via the Kullback-Leibler (KL) divergence $KL(p_{\boldsymbol{\theta}}(\boldsymbol{y}^f|\boldsymbol{x}^f) \,||\, p_{\boldsymbol{\theta}+\Delta\boldsymbol{\theta}}(\boldsymbol{y}^f|\boldsymbol{x}^f))$.

For an infinitesimally small perturbation $\Delta\boldsymbol{\theta} \to 0$, using the second-order Taylor series expansion, the expectation of the said KL divergence over the inputs $\boldsymbol{x}$ can be given by (Pascanu & Bengio, 2014):

$$\mathbb{E}_{\boldsymbol{x}^f \sim p(\boldsymbol{x}^f)}[KL(p_{\boldsymbol{\theta}}(\boldsymbol{y}^f|\boldsymbol{x}^f) \,||\, p_{\boldsymbol{\theta}+\Delta\boldsymbol{\theta}}(\boldsymbol{y}^f|\boldsymbol{x}^f))] = \Delta\boldsymbol{\theta}^T \boldsymbol{F}_{\boldsymbol{\theta}} \Delta\boldsymbol{\theta} + \mathcal{O}(\Delta\boldsymbol{\theta}^3), \tag{15}$$

where, $\boldsymbol{F}_{\boldsymbol{\theta}} \in \mathbb{R}^{D \times D}$ is the (theoretical) Fisher information matrix and is given by (Martens, 2020):

$$\boldsymbol{F}_{\boldsymbol{\theta}} = \mathbb{E}_{\boldsymbol{x}^f \sim p(\boldsymbol{x}^f)}[\mathbb{E}_{\boldsymbol{y}^f \sim p_{\boldsymbol{\theta}}(\boldsymbol{y}^f|\boldsymbol{x}^f)}[\nabla_{\boldsymbol{\theta}} \log p_{\boldsymbol{\theta}}(\boldsymbol{y}^f|\boldsymbol{x}^f) \nabla_{\boldsymbol{\theta}} \log p_{\boldsymbol{\theta}}(\boldsymbol{y}^f|\boldsymbol{x}^f)^T]]. \tag{16}$$

It is intractable to compute the full $D \times D$ Fisher information matrix for deep neural networks and hence, it is commonly approximated as a diagonal matrix, i.e., a $D$-dimensional vector, which is given by (Martens, 2020; Sung et al., 2021):

$$\hat{\boldsymbol{F}}_{\boldsymbol{\theta}} \approx \mathbb{E}_{\boldsymbol{x}^f \sim p(\boldsymbol{x}^f)}[\mathbb{E}_{\boldsymbol{y}^f \sim p_{\boldsymbol{\theta}}(\boldsymbol{y}^f|\boldsymbol{x}^f)}[\nabla_{\boldsymbol{\theta}} \log p_{\boldsymbol{\theta}}(\boldsymbol{y}^f|\boldsymbol{x}^f)]^2]. \tag{17}$$

In a practical scenario, we do not have access to the full input distribution $p(\boldsymbol{x}^f)$ and instead, only have access to a finite set of input samples $\{\boldsymbol{x}_n^f\}_{n=1}^N$ which can together serve as an approximation of $p(\boldsymbol{x}^f)$. Hence, Equation 17 becomes (Martens, 2020; Singh & Alistarh, 2020; Sung et al., 2021):

$$\hat{\boldsymbol{F}}_{\boldsymbol{\theta}} \approx \frac{1}{N} \sum_{n=1}^N [\mathbb{E}_{\boldsymbol{y}^f \sim p_{\boldsymbol{\theta}}(\boldsymbol{y}^f|\boldsymbol{x}_n^f)}[\nabla_{\boldsymbol{\theta}} \log p_{\boldsymbol{\theta}}(\boldsymbol{y}^f|\boldsymbol{x}_n^f)]^2]. \tag{18}$$

Furthermore, in supervised learning (as is the case here), we already have access to the ground truths $\{\boldsymbol{y}_n^f\}_{n=1}^N$ corresponding to all the inputs $\{\boldsymbol{x}_n^f\}_{n=1}^N$. This allows us to replace $\mathbb{E}_{\boldsymbol{y}^f \sim p_{\boldsymbol{\theta}}(\boldsymbol{y}^f|\boldsymbol{x}_n^f)}[\nabla_{\boldsymbol{\theta}} \log p_{\boldsymbol{\theta}}(\boldsymbol{y}^f|\boldsymbol{x}_n^f)]^2$ by $[\nabla_{\boldsymbol{\theta}} \log p_{\boldsymbol{\theta}}(\boldsymbol{y}_n^f|\boldsymbol{x}_n^f)]^2$. Finally, since $\mathcal{L}^f$ is generally the cross-entropy loss, $\mathcal{L}^f(\boldsymbol{x}_n^f, \boldsymbol{y}_n^f; \boldsymbol{\theta}) = -\log p_{\boldsymbol{\theta}}(\boldsymbol{y}_n^f|\boldsymbol{x}_n^f)$. Hence, Equation 18 becomes (Singh & Alistarh, 2020; Sung et al., 2021):

$$\hat{\boldsymbol{F}}_{\boldsymbol{\theta}} = \frac{1}{N} \sum_{n=1}^N [\nabla_{\boldsymbol{\theta}} \mathcal{L}^f(\boldsymbol{x}_n^f, \boldsymbol{y}_n^f; \boldsymbol{\theta})]^2, \tag{19}$$

where, $\hat{\boldsymbol{F}}_{\boldsymbol{\theta}}$ is referred to as the empirical Fisher.

The usage of the empirical Fisher is preferred in practice since it is more tractable and efficient to compute than the theoretical Fisher information matrix (Singh & Alistarh, 2020; Sung et al., 2021).

Next, from Equation 19, we can see that the importance of a model parameter, as measured by the empirical Fisher, is directly proportional to the average of the square of the gradient of the loss w.r.t. that model parameter. Let $\boldsymbol{g} \in \mathbb{R}^D$ denote the gradient vector. This implies that:

$$\hat{\boldsymbol{F}}_{\boldsymbol{\theta}} \propto (\boldsymbol{g})^2. \tag{20}$$

Returning back to our initial objective, we can now see that in order to enhance our model's ID generalizability, we need to select the top-$\rho$% of the parameters which possess the highest (empirical) Fisher information for the given downstream task. Using Equation 20, this translates to selecting the top-$\rho$% of the parameters with the highest $(\boldsymbol{g})^2$ values. Now, mathematically, it is evident that $\text{abs}(\boldsymbol{g}) \propto (\boldsymbol{g})^2$ (where, abs(.) computes the element-wise absolute value of a vector) and hence, selecting the top-$\rho$% of the parameters with the highest $\text{abs}(\boldsymbol{g})$ values efficiently (since the computational cost associated with computing the square of $\boldsymbol{g}$ is avoided) satisfies our first criterion.

### I.2 CRITERION 2: ENHANCEMENT OF OOD GENERALIZABILITY

In order to enhance the OOD generalizabilty of a model, it is first and foremost necessary to preserve the model's pre-trained knowledge. In this regard, we first need to establish a connection between the pre-training loss function and the magnitude of the pre-trained model's parameters.

For the pre-training loss function $\mathcal{L}^p(\boldsymbol{x}_n^p, \boldsymbol{y}_n^p; \boldsymbol{\theta})$, using the Taylor series expansion, we can state that (LeCun et al., 1989):

$$\Delta\mathcal{L}^p(\boldsymbol{x}_n^p, \boldsymbol{y}_n^p; \boldsymbol{\theta}) = \sum_i \frac{\partial\mathcal{L}^p(\boldsymbol{x}_n^p, \boldsymbol{y}_n^p; \boldsymbol{\theta})}{\partial\boldsymbol{\theta}_i} \Delta\boldsymbol{\theta}_i + \mathcal{O}(\|\Delta\boldsymbol{\theta}\|^2). \tag{21}$$

where, $\mathcal{O}(\|\Delta\boldsymbol{\theta}\|^2)$ denotes the second- (containing the Hessian) and higher-order terms. These are intractable to compute for deep neural networks (Lee et al., 2019) and hence, we focus solely on a first-order approximation. Hence, the influence of a parameter on the pre-training loss is given by:

$$\Delta\mathcal{L}_i^p(\boldsymbol{x}_n^p, \boldsymbol{y}_n^p; \boldsymbol{\theta}) \approx \left|\frac{\partial\mathcal{L}^p(\boldsymbol{x}_n^p, \boldsymbol{y}_n^p; \boldsymbol{\theta})}{\partial\boldsymbol{\theta}_i} \Delta\boldsymbol{\theta}_i\right|. \tag{22}$$

Now, the model parameters which would have a minimal impact on the pre-training loss are in fact, those parameters which can potentially be pruned in the pre-trained model. Accordingly, we can study the influence of a parameter $\boldsymbol{\theta}_i$ on the pre-training loss by computing the change affected by it in the value of the loss function when it gets pruned, i.e., for the case when $\Delta\boldsymbol{\theta}_i = \boldsymbol{\theta}_i - 0 = \boldsymbol{\theta}_i$.

Although we cannot possibly have access to the full pre-training dataset $\mathcal{D}^p$ for the model, a reasonable assumption is that the model's (pre-) training would have converged and hence, its pre-training

gradients are bound to be quite small. On the other hand, the model's (pre-trained) parameters can be either large or small. Mathematically, since:

$$|\boldsymbol{\theta}_i| \gtrless \left| \frac{\partial \mathcal{L}^p(\boldsymbol{x}_n^p, \boldsymbol{y}_n^p; \boldsymbol{\theta})}{\partial \boldsymbol{\theta}_i} \right|, \tag{23}$$

from Equation 22, we get that:

$$\Delta \mathcal{L}_i^p(\boldsymbol{x}_n^p, \boldsymbol{y}_n^p; \boldsymbol{\theta}) \propto |\boldsymbol{\theta}_i|. \tag{24}$$

Therefore, parameters with the smallest pre-trained magnitudes are the least important for the pre-trained model since pruning them leads to the least impact on the pre-training loss. In other words, these parameters are minimally disruptive to the pre-trained knowledge and potentially represent the under-utilized capacity of the pre-trained model.

Therefore, fine-tuning these low-magnitude parameters can *potentially* be a way to preserve the pre-trained knowledge stored in the model. We emphasize on the word 'potentially' because it is theoretically difficult to prove, with mathematical rigor (since any assumptions on the loss curvature of models with billions of parameters are likely to be too restrictive), that the parameters which can be pruned and hence, preserve the model's pre-trained knowledge, would also minimally disrupt the model's pre-trained knowledge upon fine-tuning them. However, this can still be demonstrated empirically and hence, on performing our experiments (as discussed in Section 3), we find that fine-tuning the top-$\rho\%$ of the model parameters with the smallest pre-trained magnitudes preserves and in fact, enhances the pre-trained knowledge since it leads to gains in the generalizability of the pre-trained model. Enhancement of the OOD generalizability, i.e., the model's pre-trained knowledge, (as is observed by us) is likely brought about by fine-tuning these low-magnitude parameters on the related yet distributionally-different downstream task which assists the model in developing a shared understanding of these tasks.

Our experimental results and findings are also in agreement with recent studies (Zhou et al., 2025; Ramesh et al., 2024) (which perform experiments on different datasets and models) and support our hypothesis that fine-tuning low-magnitude parameters enhances pre-trained knowledge, which is crucial for enhancing OOD generalizability.

Therefore, in accordance with the conclusions of our discussions in the aforementioned sections: Section I.1 and Section I.2, we can finally arrive at one way of (approximately) solving our multi-objective optimization problem outlined in Equation 14. Formulating our dual parameter selection criteria as detailed in Equation 2 in Section 4, wherein we fine-tune the top-$\rho\%$ of the model parameters with the largest task gradient magnitudes and the smallest pre-trained magnitudes, can therefore allow us to enhance both the ID and the OOD generalizability of our model.

