# OpenReview forum: "GaLLoP: Gradient-based Sparse Learning on Low-Magnitude Parameters"
_ICLR.cc/2026/Conference — Submitted to ICLR 2026_

### Official Review · Reviewer_RmwJ · 2025-10-23

**Soundness:** 2
**Presentation:** 2
**Contribution:** 2
**Rating:** 4
**Confidence:** 4

**Summary:**

GaLLoP is a sparse fine-tuning method that updates only parameters that are simultaneously highly task-relevant (large gradients) and minimally disruptive to pre-trained knowledge (small magnitudes), boosting both in- and out-of-distribution generalization.
It runs in two phases—score-and-select a small subset using gradient accumulation over a data slice, then fine-tune only those while masking the rest—and is model-agnostic. Across LLaMA-3 8B and Gemma-2B on eight commonsense datasets, it typically forms a dominant ID/OOD Pareto front and shows strong stability across seeds compared to LoRA, DoRA, SAFT, SpIEL, WiSE-FT, LiNeS, and FFT.
The paper also introduces diagnostics for catastrophic forgetting and memorization, finds near-zero values for GaLLoP, and notes a limitation: its unstructured sparsity is less hardware-efficient, motivating future work on structured densification.

**Strengths:**

1. The paper’s core idea—selecting a sparse set of parameters that are simultaneously high-gradient and low-magnitude—creatively fuses two complementary signals to target updates that help in-domain learning without erasing pre-trained knowledge, and it is implemented in a clean two-phase procedure.  It also contributes two evaluation diagnostics, forget ratio and collapse rate, that formalize catastrophic forgetting and severe memorization in a way that is easy to compute and compare across methods.

2. The empirical study is broad and careful: eight commonsense datasets, two model families, and strong baselines spanning full fine-tuning, parameter-efficient methods, and editing techniques.  Results consistently show a dominant ID/OOD Pareto front for GaLLoP, with particularly strong averages on LLaMA-3 8B and balance on Gemma-2B. Reproducibility is helped by explicit algorithmic pseudocode and detailed hyperparameters.

3. The exposition is easy to follow

4. Addressing the ever-present tension between adaptation and retention, the method delivers strong OOD gains while curbing overfitting that plagues several popular baselines—an outcome that matters for reliable deployment of fine-tuned LLMs.  Because GaLLoP is simple, model-agnostic, and efficient, it lowers the barrier to adopting sparse fine-tuning in practice.

**Weaknesses:**

1. The core scoring rule combines gradient size with (inverse) weight magnitude, but closely adjacent ideas already exist: SAFT selects by large gradients alone; Pruning and finetuning selects the smallest pre-trained weights; SNIP scores parameters using a gradient–weight product at (re)initialization; Movement Pruning selects by the magnitude of weight updates during fine-tuning; and dynamic sparse training like RigL periodically regrows weights based on gradient signals. A strong way to sharpen the contribution would be to add head-to-head ablations against these concrete criteria on exactly the same LLM setup and to explain theoretically why the ratio used here should dominate the product used in SNIP under typical LLM curvature/noise regimes.

2. OOD is instantiated in an eight-dataset round-robin within the same commonsense QA family. That’s a convenient stress test but not a true distribution shift across modalities or task formats. To strengthen claims about generalization, evaluate on qualitatively different targets (instruction following, long-form QA/summarization, math/code, safety and refusal behavior) and on established OOD suites (e.g., CLIP-style ImageNet variants or robustness benchmarks) to mirror SAFT’s cross-domain evaluations.

3. The paper argues that layer-wise score sampling cuts sort/memory costs and claims the method is “highly efficient,” but provides no end-to-end wall-clock, peak-memory, or throughput numbers (selection phase and training phase) versus LoRA/DoRA/SAFT.

4. Sensitivity and design choices need deeper ablations - Important parameters (global vs layer-wise thresholds, effect of ϵ, optimizer interactions like weight decay, gradient-accumulation window length) are fixed or only lightly discussed.

5. LoRA/DoRA/SAFT/SpIEL/WiSE-FT/LiNeS are useful, but widely used PEFT baselines such as IA³ and BitFit are absent; including them (and tuning LoRA ranks under an iso-FLOPs or iso-trainable-parameter protocol) would make the comparison more complete. Also disclose and equalize the number of backpropagated layers and gradient-checkpointing settings, which materially affect wall-clock and stability.

Papers:
[1]: https://arxiv.org/abs/2407.03036? "SAFT: Towards Out-of-Distribution Generalization in Fine-Tuning"
[2]: https://proceedings.mlr.press/v119/evci20a/evci20a.pdf "Rigging the Lottery: Making All Tickets Winners"
[3]: https://arxiv.org/abs/1810.02340 "SNIP: Single-shot Network Pruning based on Connection Sensitivity"

**Questions:**

See Weaknesses above.

---

> ### Author Response · Authors · 2025-11-19
> **Response to Reviewer RmwJ's Concerns [Part 1]**
>
> Thank you for your constructive feedback! We are happy to hear that you found our paper's *core idea a creative fusion of two complementary signals*, our *implementation a clean two-phase procedure*, *its contribution of two evaluation diagnostics, forget ratio and collapse rate, that formalize catastrophic forgetting and severe memorization, as a way to easily compute and compare them across methods*, *our empirical study as broad with strong baselines*, *our results showing strong averages on LLaMA-3 8B and balance on Gemma-2B*, *our implementation reproducible due to explicit algorithmic pseudocode and detailed hyperparameters*, *our exposition easy to follow*, the fact that *GaLLoP delivers strong OOD gains while curbing overfitting that plagues several popular baselines—an outcome that matters for reliable deployment of fine-tuned LLMs*, and the fact that *because GaLLoP is simple, model-agnostic, and efficient, it lowers the barrier to adopting sparse fine-tuning in practice*. We now address your concerns below:
>
> > **W1: Ablations on Parameter Selection Criteria**
>
> Firstly, we would like to point out that we have already performed ablations and experiments with GaLLoP against SAFT (fine-tuning with high-gradient parameters [1]), PaFi (Pruning and Fine-Tuning; fine-tuning of low-magnitude parameters [2]), and SpIEL (dynamic sparse fine-tuning of high-gradient parameters [3]), and have already shown and discussed the same in Sections 3 and 6 of our main text, and have provided additional details in Appendices E, F, G, and H of our paper. In fact, as stated by its authors [3], SpIEL (or, in particular, SpIEL-AG, the most performant variant of SpIEL that we compare GaLLoP against) is actually just RigL but with a slight improvement in that the parameters to be grown are selected from gradients estimated over some selection steps rather than from a single minibatch (as in RigL) since high variance in gradient magnitudes from a minibatch can harm the parameter selection process in the growth phase (see Section 3.2 of their paper): hence, comparisons with RigL are already covered in our paper.
>
> Secondly, SNIP [4] and Movement Pruning [5] are pruning techniques and their selection criteria is used to prune parameters and not fine-tune them (as is done by us in GaLLoP and by other sparse fine-tuning techniques). Hence, they are not relevant for comparison.
>
> Thirdly, on theory, following the suggestions put forward by reviewer `xcfq` and reviewer `dMY7`, we have now added a theory section in the appendix (Appendix I in the updated version of our paper) wherein we give theoretical justifications behind our dual parameter selection criteria. We encourage you to check out the new theory section!

---

> > ### Author Response · Authors · 2025-11-19
> > **Response to Reviewer RmwJ's Concerns [Part 2]**
> >
> > > **W2: OOD Evaluation**
> >
> > In the context of fine-tuning, we would like to clarify that an Out-Of-Distribution (OOD) task refers to a related task which possesses a distribution different from the fine-tuning task and OOD is *not solely* limited to distribution shifts and in fact, also refers to detecting new classes and cross-dataset transfer [1]. In fact, a task which just possesses a distribution shift is a weak OOD task since it resembles the fine-tuning (ID) task response format and contains the entire set/a subset of classes present in the fine-tuning (ID) task dataset. This is also the reason why the WiSE-FT paper [6] which performs experiments on distribution shifts of the ImageNet dataset (as also used in SAFT) states that it avoids referring to them as OOD datasets and simply labels them as distribution shifts.
> >
> > In contrast, our experimental setup spans 8 reasoning and question answering datasets which are not simply distributional shifts but possess different classes and different numbers of classes (answer options), different response/task formats, and incorporate a wide variety of different tasks which test models on grade-school level science questions, physical properties of objects, day-to-day activities of humans, general knowledge facts, social and emotional intelligence, and pronoun resolution (further details have already been provided in Appendix B of our paper). A similar choice of ID/OOD datasets has also been used in a recent paper on overtraining (see Fig. 2 on page 2 of the paper) [7]. Furthermore, in contrast to the cross-dataset transfer evaluation in the SAFT paper [1] wherein just a single possible ID/OOD combination of its datasets is considered with different density levels for its competing algorithms, we consider all possible ID and OOD combinations of our datasets and perform experiments across 4 density levels which are the same for all our competing algorithms. In this fair comparison and extensive experimentation, GaLLoP consistently improves over (/matches) the ID and OOD performance of the competing algorithms and is moreover, the only algorithm which allows for the attainment of 0\% forget ratios and 0\% collapse rates across all density levels and experimental runs -- all these results only strengthen our claims on generalizability.
> >
> > Therefore, we respectfully disagree that ours is not a true OOD evaluation. Reviewer `dMY7` has also appreciated that *our strategy is well-motivated by related experiments* and supported by *exhaustive experimental results which demonstrate its efficacy*.
> >
> > We would also like to highlight that while we are currently working on experiments on additional architectures and modalities, our computational resources are currently limited. Nevertheless, we believe that our current experiments are extensive enough and hope to add new experiments in the final version of our paper.

---

> > > ### Author Response · Authors · 2025-11-19
> > > **Response to Reviewer RmwJ's Concerns [Part 3]**
> > >
> > > > **W3: Efficiency**
> > >
> > > Firstly, we would like to clarify that the claim that GaLLoP is highly efficient (in Appendix A of our paper) has not been made in isolation and rather, we have clearly stated that it is the layer-wise uniform random sampling of only a small percentage of the total scores that makes our algorithm highly efficient as compared to the case when it is implemented without this practical optimization: ''*This* drastically reduces the compute and memory requirements of GaLLoP and hence, makes it highly efficient''. In fact, in Appendix A, we have already given a detailed explanation of how this optimization imparts a high efficiency to GaLLoP: for the case of a model with $D$ number of parameters in total, post the sampling of scores, an element-wise comparison of scores (for mask computation) with the computed score threshold $s_t$ (from the sampled scores) takes only $\mathcal{O}(D)$ time in contrast to the earlier case, without any sampling, wherein sorting of all the $D$ scores (to compute $s_t$) takes $\mathcal{O}(D\log D)$ time and furthermore, saving only $s\\%$ of the scores (with sampling) allows us to save on $(100 - s)\\%$ of the total memory required to save all the $D$ scores (as is the case without sampling). Given that we deal with fine-tuning models with billions of parameters, i.e., $D \sim 10^9$, $\mathcal{O}(D) << \mathcal{O}(D\log D)$, and if storing a single score value is assumed to consume $m$ bits, $s\\%$ of $mD << mD$, since s is very small (just 10\%; as we describe in Appendix A and Fig. 7). The consequent high efficiency of GaLLoP is thus clearly justified.
> > >
> > > Secondly, as far as the efficiency comparison with LoRA and DoRA (RFT techniques) and SAFT (SpFT technique) is concerned, as we have already mentioned in our conclusion section, just like other SpFT algorithms such as SAFT, GaLLoP also leads to unstructured sparsity which leads to slight increase in their latency as compared to RFT techniques such as LoRA and DoRA (total memory requirements are almost the same for all: RFT as well as SpFT). In the course of experimentation, we found that all SpFT algorithms take about 1.05-1.5X longer (fine-tuning dataset size dependent) for fine-tuning models than RFT techniques owing to the additional parameter selection phase. These latency figures are based on our experiments on NVIDIA RTX A6000 and RTX 6000 Ada GPUs, which are not optimized for unstructured sparsity-based computations. We would also like to highlight that with the recent advancements in hardware such as Cerebras WS-3 [8], natively accelerating unstructured sparsity has become a reality, and this could potentially allow SpFT to be as efficient as/even more efficient than RFT. Nevertheless, as we also already note, making the sparsity (obtained via the usage of SpFT) structured for dense computations remains an interesting direction for future work and we believe that the strong generalizability gains of GaLLoP over RFT techniques (as already acknowledged by you) outweighs this marginal latency increase.

---

> > > > ### Author Response · Authors · 2025-11-19
> > > > **Response to Reviewer RmwJ's Concerns [Part 4]**
> > > >
> > > > > **W4: Ablations/Discussions on Sensitivity and Design Choices w.r.t. GaLLoP**
> > > >
> > > > 1. **Global vs layer-wise thresholds**: We would like to clarify that, as already explained in Appendix A of our paper, the score threshold is a global threshold, i.e., it is just a single value which is computed from either all the scores or a small sample of them. In other words, it is only the sampling of the scores which is done layer-wise and to this end, we have already provided the results of representative ablation studies (see Fig. 7a and 7b in Appendix A) wherein it can clearly be observed that layer-wise sampling of only 10\% of total scores allows us to obtain an effective density level that matches the required density level.
> > > > 2. **Effect of $\epsilon$**: Thank you for asking this question! We set $\epsilon$ to $10^{-8}$ as it is just an order of magnitude smaller than the smallest non-zero pre-trained parameter magnitude in both the LLMs used in our experiments and hence, does not artificially inflate any pre-trained parameter's magnitude. We have clarified this in Appendix A of the updated version of our paper.
> > > > 3. **Optimizer interactions like weight decay**: Thank you for asking this question! We did not use weight decay while fine-tuning models using GaLLoP since as we show across all our experiments, infusing sparsity via SpFT already acts as an effective regularizer by ensuring that only a small fraction of the model parameters receive updates. Analyzing the effect of introducing weight decay along with unstructured sparsity remains an interesting direction for future work.
> > > > 4. **Gradient-accumulation window length**: Thank you for asking this question! The number of gradient accumulation steps, i.e., the gradient-accumulation window length, was always chosen such that the effective batch size (= Per-GPU batch size $\times$ gradient accumulation steps $\times$ no. of GPUs) remains 16 for each fine-tuning experiment for a fair comparison. The effective batch size was limited to 16 because unfortunately, SpIEL does not support distributed fine-tuning and its authors did not use gradient accumulation in their own experiments [3]. Consequently, for our experiments with SpIEL, 16 was the largest batch size which could fit in our NVIDIA RTX A6000/RTX 6000 Ada GPU's memory. We have clarified this in Appendix C in the updated version of our paper.

---

> > > > > ### Author Response · Authors · 2025-11-19
> > > > > **Response to Reviewer RmwJ's Concerns [Part 5]**
> > > > >
> > > > > > **W5: Comparisons**
> > > > >
> > > > > Firstly, we would like to clarify why (IA)$^3$ and BitFit were not included as baselines:
> > > > >
> > > > > 1. **(IA)$^3$**: This is a few-shot PEFT technique [9] in contrast to our SpFT technique (GaLLoP) and the other competing algorithms (that we have compared GaLLoP against) which are all zero-shot PEFT techniques. For our work, we only consider the zero-shot setting wherein models are fine-tuned and evaluated without providing any few-shot examples along with a dataset's own input, for a robust evaluation of generalizability. Hence, it is not relevant for comparison. Moreover, in our paper, for a fair comparison, for a given experiment on fine-tuning a model, we ensure that all PEFT algorithms introduce/fine-tune the same number of parameters, i.e., the density level is fixed at the same value. In contrast, (IA)$^3$ does not allow for any modifications in the number of parameters it introduces for fine-tuning [9], which biases evaluation.
> > > > > 2. **BitFit**: This PEFT technique fine-tunes only the bias terms of a model [10]. Given that LLaMA3 8B (https://huggingface.co/meta-llama/Meta-Llama-3-8B/blob/main/model.safetensors.index.json) and Gemma 2B (https://huggingface.co/google/gemma-2b/blob/main/model.safetensors.index.json) do not possess any bias terms (they only have weights) and the official BitFit code (https://github.com/benzakenelad/BitFit/blob/main/glue_evaluator.py#L73) specifically looks for *designated* bias terms in a model for fine-tuning, BitFit cannot be used. In fact, a majority of LLMs do not possess bias terms, for e.g., PaLM [11]: "No biases were used in any of the dense kernels or layer norms. We found this to result in increased training stability for large models." (see Section 2, page 8 of their paper), OLMo [12]: "Following LLaMA, PaLM, and others, we exclude all bias terms from our architecture in order to improve training stability." (see Section 2, page 1 of their paper): hence, BitFit is majorly not applicable for fine-tuning models and cannot be utilized for comparison.
> > > > >
> > > > > Secondly, we would also like to clarify that LoRA ranks have already been tuned under an iso-trainable parameter setting. In fact, as already stated in Section 5.5 of our paper, in all our experiments on fine-tuning models with PEFT algorithms, we ensure that the density levels remain fixed and the same for all PEFT algorithms: "We explore four density levels: {0.24\%, 0.47\%, 0.93\%, 1.85\%} for LLaMA3 8B, and {0.37\%, 0.73\%, 1.45\%, 2.86\%} for Gemma 2B which correspond to ranks {8, 16, 32, 64} used for reparametrized fine-tuning." This ensures that all our comparisons are fair and complete (an iso-FLOPs setting is not possible since the parameter selection phase in all SpFT techniques is bound to consume more FLOPs (on hardware optimized for dense computations) than the RFT techniques; this setting maybe achieved with recent advancements in hardware such as Cerebras WS-3 [8] which can natively accelerate unstructured sparsity and remains an interesting direction for future work).
> > > > >
> > > > > Thirdly, we have already disclosed and equalized the number of backpropagated layers for all our PEFT experiments: as stated in Section 5.5, to ensure a fair comparison, we apply each fine-tuning algorithm across all the same transformer layers (we accidentally missed adding the word `same' and have added it now in that sentence in our paper's updated version; thanks for catching this!) and as specified in Tables 5 and 6 in Appendix C of our paper, the target modules are q_proj, k_proj, v_proj, up_proj, and down_proj (following the recommendations of the DoRA paper [13]) in LLaMA3 8B and Gemma 2B. We accidentally forgot to mention these target modules in Tables 2 and 3 in Appendix C and have now explicitly mentioned them in the updated version of our paper. Thank you for asking this!
> > > > >
> > > > > Finally, we would like to clarify that the activation (/gradient) checkpointing settings are also already the same for all fine-tuning algorithms and we apply activation checkpointing to only the attention layers of our models (we have explicitly specified this in the updated version of our paper) since they account for a majority of the activation memory and utilizing activation checkpointing can help in reducing the memory footprint of fine-tuning. Application of activation checkpointing to only the attention layers is a default in torchtune recipes [14]. Thank you for asking this!
> > > > >
> > > > > **We hope that our response clarifies your concerns and we kindly request you to reconsider and increase your score.**

---

> > > > > > ### Author Response · Authors · 2025-11-19
> > > > > > **References**
> > > > > >
> > > > > > [1] Bac Nguyen *et al*. SAFT: Towards out-of-distribution generalization in fine-tuning. In Computer Vision – ECCV 2024: 18th European Conference, Milan, Italy, September 29–October 4, 2024, Proceedings, Part LXIX, pp. 138–154. URL: https://doi.org/10.1007/978-3-031-72890-7_9.
> > > > > >
> > > > > > [2] Baohao Liao *et al*. Parameter-efficient fine-tuning without introducing new latency. In Anna Rogers, Jordan Boyd-Graber, and Naoaki Okazaki (eds.), Proceedings
> > > > > > of the 61st Annual Meeting of the Association for Computational Linguistics (Volume 1: Long Papers), pp. 4242–4260, Toronto, Canada, July 2023. Association for Computational Linguistics. URL: https://aclanthology.org/2023.acl-long.233/.
> > > > > >
> > > > > > [3] Alan Ansell *et al*. Scaling sparse fine-tuning to large language models, 2024. URL: https://arxiv.org/abs/2401.16405.
> > > > > >
> > > > > > [4] Namhoon Lee *et al*. SNIP: Single-shot network pruning based on connection sensitivity. In International Conference on Learning Representations, 2019. URL: https://openreview.net/forum?id=B1VZqjAcYX.
> > > > > >
> > > > > > [5] Victor Sanh *et al*. Movement pruning: adaptive sparsity by fine-tuning. In Proceedings of the 34th International Conference on Neural Information Processing Systems, NIPS ’20, Red Hook, NY, USA, 2020. Curran Associates Inc. URL: https://proceedings.neurips.cc/paper/2020/file/eae15aabaa768ae4a5993a8a4f4fa6e4-Paper.pdf.
> > > > > >
> > > > > > [6] Mitchell Wortsman *et al*. Robust fine-tuning of zero-shot models. In Proceedings of the IEEE/CVF Conference on Computer Vision and Pattern Recognition (CVPR), pp. 7959–7971, June 2022. URL: https://openaccess.thecvf.com/content/CVPR2022/papers/Wortsman_Robust_Fine-Tuning_of_Zero-Shot_Models_CVPR_2022_paper.pdf.
> > > > > >
> > > > > > [7] Jacob Mitchell Springer *et al*. Overtrained language models are harder to fine-tune. In Forty-second International Conference on Machine Learning, 2025. URL: https://openreview.net/forum?id=YW6edSufht.
> > > > > >
> > > > > > [8] Sean Lie. Cerebras architecture deep dive: First look inside the hardware/software co-design for deep learning. IEEE Micro, 43(3):18–30, May 2023. URL: https://doi.org/10.1109/MM.2023.3256384.
> > > > > >
> > > > > > [9] Haokun Liu *et al*. Few-shot parameter-efficient fine-tuning is better and cheaper than in-context learning. In S. Koyejo, S. Mohamed, A. Agarwal, D. Belgrave, K. Cho, and A. Oh (eds.), Advances in Neural Information Processing Systems, volume 35, pp. 1950–1965. Curran Associates, Inc., 2022. URL: https://proceedings.neurips.cc/paper_files/paper/2022/file/0cde695b83bd186c1fd456302888454c-Paper-Conference.pdf.
> > > > > >
> > > > > > [10] Elad Ben Zaken *et al*. BitFit: Simple parameter-efficient fine-tuning for transformer-based masked language-models. In Smaranda Muresan, Preslav Nakov, and Aline Villavicencio (eds.), Proceedings of the 60th Annual Meeting of the Association for Computational Linguistics (Volume 2: Short Papers), pp. 1–9, Dublin, Ireland, May 2022. Association for Computational Linguistics. URL: https://aclanthology.org/2022.acl-short.1/.
> > > > > >
> > > > > > [11] Aakanksha Chowdhery *et al*. PaLM: scaling language modeling with pathways. J. Mach. Learn. Res., 24(1), January 2023. URL: https://dl.acm.org/doi/pdf/10.5555/3648699.3648939
> > > > > >
> > > > > > [12] Dirk Groeneveld *et al*. OLMo: Accelerating the science of language models. In Lun-Wei Ku, Andre Martins, and Vivek Srikumar (eds.), Proceedings of the 62nd Annual Meeting of the Association for Computational Linguistics (Volume 1: Long Papers), pp. 15789–15809, Bangkok, Thailand, August 2024. Association for Computational Linguistics. URL: https://aclanthology.org/2024.acl-long.841/.
> > > > > >
> > > > > > [13] Shih yang Liu *et al*. DoRA: Weight-decomposed low-rank adaptation. In Forty-first International Conference on Machine Learning, 2024. URL: https://openreview.net/forum?id=3d5CIRG1n2.
> > > > > >
> > > > > > [14] torchtune maintainers and contributors. torchtune: PyTorch’s post-training library, April 2024. URL: https://github.com/pytorch/torchtune.

---

### Official Review · Reviewer_dMY7 · 2025-10-31

**Soundness:** 3
**Presentation:** 3
**Contribution:** 3
**Rating:** 6
**Confidence:** 3

**Summary:**

The paper presents a method for sparse fine tuning that attempts to avoid catastrophic forgetting while avoiding over fitting on newer tasks. Both in-distribution and out-of-distribution performance is shown to improve. The technique chooses a small subset of parameters that are fine tuned based on two criteria - (i) magnitude of the parameter in the pretrained model, (ii) and gradient of the parameter while fine tuning. A small data set sample is used for fine tuning to estimate the later value. The idea is simple but is well motivated by experimental results. Empirical results demonstrate the superiority of the proposed technique over related SpFT techniques.

**Strengths:**

1) The paper proposes a simple, intuitive, but effective strategy for sparse fine tuning.
2) The strategy is well motivated by related experiments.
3) Exhaustive experimental results are presented to demonstrate the efficacy of the proposed technique.
4) An improvement in out-of-distribution tasks is shown.

**Weaknesses:**

1) The idea is only experimentally demonstrated to work well. Theoretical justifications are missing. There are a number of view points and observations in literature about the choice of parameters while fine tuning. The role of magnitude of a parameter in determining its importance is not unanimously agreed upon. Similarly, gradient value while backpropagating on a small batch of data might be too noisy as a signal to score parameter importance. LoRA and related low-dimension projection techniques techniques looks at spectral properties of the parameter matrix rather than the magnitude and gradients. Other approaches like the ones based on expander graphs consider network connectivity rather than parameter weights. Authors may discuss/support the proposed algorithm in more "theoretical" term.
2) The efficacy of the proposed technique in a continual learning scenario is not discussed. Magnitude and gradient based parameter selection may either be erroneous or yield a low sparsification ratio when performed in sequence of rounds on multiple tasks. Experiments need to be presented to counter this claim. Such a scenario was mentioned as a motivation in the introduction.
3) Because of the rather dynamic and unstructured nature of the proposed parameter selection technique the method may not align well with hardware acceleration thus defeating the benefits of sparse fine tuning.

**Questions:**

1) How does the method perform on already sparse LLM models? Such models are increasingly becoming more popular. A small experiment on applying the proposed algorithm to some sparse LLM may provide valuable insight.
 2) What is the effect of running the proposed method over multiple tasks in a sequence. The effect of overfitting and catastrophic forgetting would be more evident.
3) Authors may delve deeper into studying the set of parameters chosen for fine-tuning using the first and the second criteria (namely, low magnitude, and high gradient) respectively. What is the percentage intersection of these sets of parameters?
4) A summary table comparing the performance of the proposed approach with SOTA in terms of accuracy/forgetfullness/density may be reported in the main paper itself.

---

> ### Author Response · Authors · 2025-11-19
> **Response to Reviewer dMY7's Concerns [Part 1]**
>
> Thank you for your constructive feedback! We are happy to hear that you found our strategy *simple, intuitive, effective*, *well-motivated by related experiments*, and supported by *exhaustive experimental results* which *demonstrate the superiority of the proposed technique over related SpFT techniques*! We now address your concerns below:
>
> > **W1: Theory**
>
> We thank you for your suggestions to incorporate theoretical justifications for our parameter selection criteria! We have now added a **new section on theory as Appendix I of the updated version of our paper** wherein as suggested by you and reviewer `xcfq`, we now discuss the connections between gradient magnitude and parameter sensitivity and show how parameter magnitude relates to the pre-trained knowledge stored in a model. Empirical justifications showing how fine-tuning the top-$\rho$\% of the low-magnitude parameters enhances a model's generalizability have already been shown and discussed in Section 3 of our paper and we have also already provided a full ablation study in Appendix H of our paper wherein we demonstrate that fine-tuning the top-$\rho$\% of the model parameters which possess low-magnitudes and high task gradients leads to a dominant Pareto front for both ID as well as OOD performance, across all density levels and experimental runs, over fine-tuning the top-$\rho$\% model parameters based on whether they have either a) low-magnitudes or b) high task gradients only.
>
> To summarize our theoretical justifications, we are also discussing two key ideas below (though we encourage you to check out Appendix I which contains a more complete version):

---

> > ### Author Response · Authors · 2025-11-19
> > **Response to Reviewer dMY7's Concerns [Part 2]**
> >
> > Consider a model with a parameter vector $\boldsymbol{\theta} \in \mathbb{R}^D$ which has been pre-trained on a dataset $\mathcal{D}^p = {\\{(\boldsymbol{x}^p\_n, \boldsymbol{y}^p\_n)\\}}\_{n = 1}^{N}$, where, $\boldsymbol{x}^p_n \in \mathbb{R}^{P}$ and $\boldsymbol{y}^p_n \in \mathbb{R}^{Q}$, with a pre-training loss function $\mathcal{L}^p(\boldsymbol{x}^p_n, \boldsymbol{y}^p_n; \boldsymbol{\theta})$. This model will now be fine-tuned on a dataset $\mathcal{D}^f = {\\{(\boldsymbol{x}^f_n, \boldsymbol{y}^f_n)\\}}_{n = 1}^{N}$, where, $\boldsymbol{x}^f_n \in \mathbb{R}^{P}$ and $\boldsymbol{y}^f_n \in \mathbb{R}^{Q}$, with a fine-tuning loss function $\mathcal{L}^f(\boldsymbol{x}^f_n, \boldsymbol{y}^f_n; \boldsymbol{\theta})$.
> >
> > **Criterion 1: Connection between Gradient Magnitude and Parameter Sensitivity**
> >
> > For a conditional probability distribution $p_{\boldsymbol{\theta}}(\boldsymbol{y}^f|\boldsymbol{x}^f)$ generated by the model over the output $\boldsymbol{y}^f \in \mathbb{R}^{Q}$, given an input $\boldsymbol{x}^f \in \mathbb{R}^{P}$, one way to estimate the importance of each of its parameters is to determine the amount by which the model's prediction would change upon perturbing its parameters by a small amount ($\Delta\boldsymbol{\theta}$), i.e., via the Kullback-Leibler (KL) divergence $KL (p_{\boldsymbol{\theta}}(\boldsymbol{y}^f|\boldsymbol{x}^f) || p_{\boldsymbol{\theta}+\Delta\boldsymbol{\theta}}(\boldsymbol{y}^f|\boldsymbol{x}^f))$.
> >
> > For an infinitesimally small perturbation $\Delta\boldsymbol{\theta} \rightarrow 0$, using the second-order Taylor series expansion, the expectation of the said KL divergence over the inputs $\boldsymbol{x}$ can be given by [1]:
> >
> > $$\mathbb{E}\_{\boldsymbol{x}^f \sim p(\boldsymbol{x}^f)}[KL (p\_{\boldsymbol{\theta}}(\boldsymbol{y}^f|\boldsymbol{x}^f) || p_{\boldsymbol{\theta}+\Delta\boldsymbol{\theta}}(\boldsymbol{y}^f|\boldsymbol{x}^f))] = \Delta\boldsymbol{\theta}^T \boldsymbol{F}_{\boldsymbol{\theta}} \Delta\boldsymbol{\theta} + \mathcal{O}(\Delta\boldsymbol{\theta}^3),$$
> >
> > where, $\boldsymbol{F}\_{\boldsymbol{\theta}}$ is the (theoretical) Fisher information matrix and is given by [2]:
> > $$\boldsymbol{F}\_{\boldsymbol{\theta}} = \mathbb{E}\_{\boldsymbol{x}^f\sim p(\boldsymbol{x}^f)}[\mathbb{E}\_{\boldsymbol{y}^f\sim p\_{\boldsymbol{\theta}}(\boldsymbol{y}^f|\boldsymbol{x}^f)}[\nabla_{\boldsymbol{\theta}}\log p_{\boldsymbol{\theta}}(\boldsymbol{y}^f|\boldsymbol{x}^f)\nabla_{\boldsymbol{\theta}}\log p_{\boldsymbol{\theta}}(\boldsymbol{y}^f|\boldsymbol{x}^f)^T]].$$
> >
> > It is intractable to compute the full $D \times D$ Fisher information matrix for deep neural networks and hence, it is commonly approximated as a diagonal matrix, i.e., a $D$-dimensional vector, which is given by [2, 4]:
> > $$\hat{\boldsymbol{F}\_{\boldsymbol{\theta}}} \approx \mathbb{E}\_{\boldsymbol{x}^f\sim p(\boldsymbol{x}^f)}[\mathbb{E}\_{\boldsymbol{y}^f\sim p\_{\boldsymbol{\theta}}(\boldsymbol{y}^f|\boldsymbol{x}^f)}[\nabla_{\boldsymbol{\theta}}\log p_{\boldsymbol{\theta}}(\boldsymbol{y}^f|\boldsymbol{x}^f)]^2].$$
> >
> > In a practical scenario, we do not have access to the full input distribution $p(\boldsymbol{x}^f)$ and instead, only have access to a finite set of input samples $\\{ \boldsymbol{x}^f_n \\}\_{n = 1}^N$ which can together serve as an approximation of $p(\boldsymbol{x}^f)$. Hence, we now get that [2, 3, 4]:
> > $$\hat{\boldsymbol{F}\_{\boldsymbol{\theta}}} \approx \dfrac{1}{N}\sum_{n = 1}^N[\mathbb{E}\_{\boldsymbol{y}^f\sim p\_{\boldsymbol{\theta}}(\boldsymbol{y}^f|\boldsymbol{x}^f_n)}[\nabla\_{\boldsymbol{\theta}}\log p\_{\boldsymbol{\theta}}(\boldsymbol{y}^f|\boldsymbol{x}^f_n)]^2].$$
> >
> > Furthermore, in supervised learning (as is the case here), we already have access to the ground truths $\\{\boldsymbol{y}^f\_n\\}\_{n = 1}^N$ corresponding to all the inputs $\\{\boldsymbol{x}^f\_n\\}\_{n = 1}^N$. This allows us to replace $\mathbb{E}\_{\boldsymbol{y}^f\sim p\_{\boldsymbol{\theta}}(\boldsymbol{y}^f|\boldsymbol{x}^f_n)}[\nabla_{\boldsymbol{\theta}}\log p_{\boldsymbol{\theta}}(\boldsymbol{y}^f|\boldsymbol{x}^f_n)]^2$ by $[\nabla_{\boldsymbol{\theta}}\log p_{\boldsymbol{\theta}}(\boldsymbol{y}^f_n|\boldsymbol{x}^f_n)]^2$. Finally, since $\mathcal{L}^f$ is generally the cross-entropy loss, $\mathcal{L}^f(\boldsymbol{x}^f_n, \boldsymbol{y}^f_n; \boldsymbol{\theta}) = -\log p_{\boldsymbol{\theta}}(\boldsymbol{y}^f_n|\boldsymbol{x}^f_n)$. Hence, we now get that [3, 4]:$$\hat{\boldsymbol{F}\_{\boldsymbol{\theta}}} = \dfrac{1}{N}\sum_{n = 1}^N [\nabla_{\boldsymbol{\theta}}\mathcal{L}^f(\boldsymbol{x}^f_n, \boldsymbol{y}^f_n; \boldsymbol{\theta})]^2,$$
> > where, $\hat{\boldsymbol{F}\_{\boldsymbol{\theta}}}$ is referred to as the empirical Fisher.
> >
> > The usage of the empirical Fisher is preferred in practice since it is more tractable and efficient to compute than the theoretical Fisher information matrix [3, 4].

---

> > > ### Author Response · Authors · 2025-11-19
> > > **Response to Reviewer dMY7's Concerns [Part 3]**
> > >
> > > Next, from the expression for the empirical Fisher, we can see that the importance of a model parameter, as measured by the empirical Fisher, is directly proportional to the average of the square of the gradient of the loss w.r.t. that model parameter. Let $\boldsymbol{g} \in \mathbb{R}^D$ denote the gradient vector. This implies that:$$\hat{\boldsymbol{F}_{\boldsymbol{\theta}}} \propto (\boldsymbol{g})^2.$$
> > >
> > > Returning back to our initial objective, we can now see that in order to enhance our model's ID generalizability, we need to select the top-$\rho$\% of the parameters which possess the highest (empirical) Fisher information for the given downstream task. Using the aforementioned proportionality, this translates to selecting the top-$\rho$\% of the parameters with the highest $(\boldsymbol{g})^2$ values. Now, mathematically, it is evident that $\text{abs}(\boldsymbol{g}) \propto (\boldsymbol{g})^2$ (where, $\text{abs}(.)$ computes the element-wise absolute value of a vector) and hence, selecting the top-$\rho$\% of the parameters with the highest $\text{abs}(\boldsymbol{g})$ values efficiently (since the computational cost associated with computing the square of $\boldsymbol{g}$ is avoided) satisfies our first criterion.
> > >
> > > **Criterion 2: Connection between Parameter Magnitude and Pre-Trained Knowledge**
> > >
> > > In order to draw connections between parameter magnitude and pre-trained knowledge, we can consider the problem of figuring out which model parameters when pruned would result in a minimal impact on the pre-trained loss.
> > >
> > > For the pre-training loss function $\mathcal{L}^p(\boldsymbol{x}^p_n, \boldsymbol{y}^p_n; \boldsymbol{\theta})$, using the Taylor series expansion, we can state that:$$\Delta\mathcal{L}^p(\boldsymbol{x}^p\_n, \boldsymbol{y}^p\_n; \boldsymbol{\theta}) = \sum\_{i}\frac{\partial \mathcal{L}^p(\boldsymbol{x}^p\_n, \boldsymbol{y}^p\_n; \boldsymbol{\theta})}{\partial \boldsymbol{\theta}\_{i}} \Delta \boldsymbol{\theta}\_{i} + \mathcal{O}(\lVert \Delta\boldsymbol{\theta} \rVert^2).$$
> > >
> > > where, $\mathcal{O}(\lVert \Delta\boldsymbol{\theta} \rVert^2)$ denotes the second- (containing the Hessian) and higher-order terms. These are intractable to compute for deep neural networks [6] and hence, we focus solely on a first-order approximation. Hence, the influence of a parameter on the pre-training loss is given by:$$\Delta\mathcal{L}\_i^p(\boldsymbol{x}^p\_n, \boldsymbol{y}^p\_n; \boldsymbol{\theta}) \approx \left|\frac{\partial \mathcal{L}^p(\boldsymbol{x}^p\_n, \boldsymbol{y}^p\_n; \boldsymbol{\theta})}{\partial \boldsymbol{\theta}\_{i}} \Delta\boldsymbol{\theta}\_{i}\right|.$$
> > >
> > > Now, the model parameters which would have a minimal impact on the pre-training loss are in fact, those parameters which can potentially be pruned in the pre-trained model. Accordingly, we can study the influence of a parameter $\boldsymbol{\theta}\_{i}$ on the pre-training loss by computing the change affected by it in the value of the loss function when it gets pruned, i.e., for the case when $\Delta \boldsymbol{\theta}\_{i} = \boldsymbol{\theta}\_{i} - 0 = \boldsymbol{\theta}\_{i}$.
> > >
> > > Although we cannot possibly have access to the full pre-training dataset $\mathcal{D}^p$ for the model, a reasonable assumption is that the model's (pre-) training would have converged and hence, its pre-training gradients are bound to be quite small. On the other hand, the model's (pre-trained) parameters can be either large or small. Mathematically, since: $$\left|\boldsymbol{\theta}\_{i}\right| \gtrapprox \left|\frac{\partial \mathcal{L}^p(\boldsymbol{x}^p\_n, \boldsymbol{y}^p\_n; \boldsymbol{\theta})}{\partial \boldsymbol{\theta}\_{i}}\right|,$$
> > > we get that:
> > > $$\Delta\mathcal{L}\_i^p(\boldsymbol{x}^p\_n, \boldsymbol{y}^p\_n; \boldsymbol{\theta}) \propto \left|\boldsymbol{\theta}\_{i}\right|.$$
> > >
> > > Therefore, parameters with the smallest pre-trained magnitudes are the least important for the pre-trained model since pruning them leads to the least impact on the pre-training loss. In other words, these parameters are minimally disruptive to the pre-trained knowledge and potentially represent the under-utilized capacity of the pre-trained model.
> > >
> > > Therefore, fine-tuning these low-magnitude parameters can potentially be a way to preserve the pre-trained knowledge stored in the model.

---

> > > > ### Author Response · Authors · 2025-11-19
> > > > **Response to Reviewer dMY7's Concerns [Part 4]**
> > > >
> > > > > **W2 and Q2: Continual Learning**
> > > >
> > > > We thank you for suggesting the exploration of this field. However, we would like to mention that such a continual learning scenario has not been mentioned as a motivation in the introduction or anywhere else in our paper. For our work, we focussed on experiments in a static setting following a round-robin approach wherein we fine-tuned models on a single dataset and evaluated their performance on the others to examine their ID and OOD accuracy, forget ratios (quantification of catastrophic forgetting), and collapse rates (quantification of severe memorization) across all possible task combinations and across all density levels. Exploring the utility of GaLLoP in a continual learning scenario is beyond the current scope of our paper but represents an interesting direction for future work. We also appreciate that you found that *our strategy is well-motivated by related experiments* and supported by *exhaustive experimental results which demonstrate its efficacy*.
> > > >
> > > > > **W3: Unstructured Nature of Sparsity**
> > > >
> > > > We respectfully disagree that the fact that GaLLoP may not align well with hardware acceleration defeats the benefits of sparse fine-tuning. On the contrary, as already noted by us in the conclusion of our paper (Section 7), previously developed sparse fine-tuning (SpFT) techniques such as SAFT [7] and SpIEL [8] also lead to unstructured sparsity as is the case with GaLLoP. Hence, in this regard, GaLLoP does not defeat the benefits of SpFT. In fact, we would like to highlight that with the recent advancements in hardware such as Cerebras WS-3 [9], natively accelerating unstructured sparsity has become a reality. Nevertheless, as we already acknowledge in our paper, structured densification of this sparsity remains an interesting direction for future work and we believe that the strong generalizability gains of GaLLoP over competing algorithms (as already acknowledged by you as *demonstrating its superiority over related SpFT techniques*) makes it a strong candidate for adopting it for fine-tuning models in practice.
> > > >
> > > > > **Q1: Sparse LLMs**
> > > >
> > > > We thank you for suggesting this area for exploration. The focus of our work was on the sparse fine-tuning of the currently ubiquitous dense LLMs but we are working on experiments in this area and hope to add them in the final version of our paper, as our computational resources are currently limited.
> > > >
> > > > > **Q3: Intersection between the Parameter Sets selected by the First and Second Criteria**
> > > >
> > > > We thank you for asking this interesting question! While the percentage intersection of the sets of parameters selected by the first and second criteria varies depending upon the fine-tuning dataset owing to the differing task gradient magnitudes, as we observed from our experiments, an important noticeable trend across all datasets is that this percentage intersection tends to increase with the increase in the density level due to low-magnitude and high-gradient dilution, as we already show and discuss in detail in Appendix H of our paper.

---

> > > > > ### Author Response · Authors · 2025-11-19
> > > > > **Response to Reviewer dMY7's Concerns [Part 5]**
> > > > >
> > > > > > **Q4: Summary of Comparisons with SOTA**
> > > > >
> > > > > We thank you for this suggestion! While the figures and their captions in the results section of our paper's main text already provide a summary of our results, please find the summary table below (this has now been added as a part of our conclusion since 10 main text pages are allowed during the rebuttal/discussion period as per ICLR guidelines):
> > > > >
> > > > > | Performance Metric | GaLLoP | Competing Algorithms |
> > > > > | :-------------------------: | :------------: | :----------------------------: |
> > > > > | **ID and OOD Accuracy** | Fine-tuned models form a **dominant Pareto front** and/or **show high performance** (LLaMA3 8B) or **consistently show high and balanced performance** (Gemma 2B). | Fine-tuned models either breakdown in the overtrained regime (LLaMA3 8B) or increasingly overfit the distribution of the fine-tuning dataset with the increase in the density level and/or fine-tuning dataset size (Gemma 2B). |
> > > > > | **Forget Ratio (Catastrophic Forgetting)** | Fine-tuned models show **0\% forget ratios** across all density levels and experimental runs. | Fine-tuned models show increasingly high forget ratios with the increase in the density level because they overfit the distribution of the fine-tuning dataset. |
> > > > > | **Collapse Rate (Severe Memorization)** | Fine-tuned models show **0\% collapse rates** across all density levels and experimental runs. | Fine-tuned models (except SAFT-based) show increasingly high collapse rates with the increase in the density level because they overfit the distribution of the fine-tuning dataset. |
> > > > > | **Stability** | Fine-tuned models show the **highest performance stability** by attaining the highest median ID and OOD accuracies with the least interquartile ranges and showing 0\% variability in their forget ratios and collapse rates. | Fine-tuned models exhibit unstable performance by attaining lower median ID and OOD accuracies with high interquartile ranges and showing high values of and/or high variability in their forget ratios and collapse rates. |
> > > > >
> > > > > **We hope that our response clarifies your concerns and we kindly request you to reconsider and increase your score.**

---

> > > > > > ### Author Response · Authors · 2025-11-19
> > > > > > **References**
> > > > > >
> > > > > > [1] Razvan Pascanu *et al*. Revisiting natural gradient for deep networks, 2014. URL: https://arxiv.org/abs/1301.3584.
> > > > > >
> > > > > > [2] James Martens *et al*. New insights and perspectives on the natural gradient method. Journal of Machine Learning Research, 21(146):1–76, 2020. URL: http://jmlr.org/papers/v21/17-678.html.
> > > > > >
> > > > > > [3] Sidak Pal Singh *et al*. Woodfisher: Efficient second-order approximation for neural network compression. In H. Larochelle, M. Ranzato, R. Hadsell, M.F. Balcan, and H. Lin (eds.), Advances in Neural Information Processing Systems, volume 33, pp. 18098–18109. Curran Associates, Inc., 2020. URL: https://proceedings.neurips.cc/paper_files/paper/2020/file/d1ff1ec86b62cd5f3903ff19c3a326b2-Paper.pdf.
> > > > > >
> > > > > > [4] Yi-Lin Sung *et al*. Training neural networks with fixed sparse masks. In A. Beygelzimer, Y. Dauphin, P. Liang, and J. Wortman Vaughan (eds.), Advances in Neural Information Processing Systems, 2021. URL: https://openreview.net/forum?id=Uwh-v1HSw-x.
> > > > > >
> > > > > > [5] Yann LeCun *et al*. Optimal brain damage. In D. Touretzky (ed.), Advances in Neural Information Processing Systems, volume 2. Morgan-Kaufmann, 1989. URL: https://proceedings.neurips.cc/paper_files/paper/1989/file/6c9882bbac1c7093bd25041881277658-Paper.pdf.
> > > > > >
> > > > > > [6] Namhoon Lee *et al*. SNIP: Single-shot network pruning based on connection sensitivity. In International Conference on Learning Representations, 2019. URL: https://openreview.net/forum?id=B1VZqjAcYX.
> > > > > >
> > > > > > [7] Bac Nguyen *et al*. SAFT: Towards out-of-distribution generalization in fine-tuning. In Computer Vision – ECCV 2024: 18th European Conference, Milan, Italy, September 29–October 4, 2024, Proceedings, Part LXIX, pp. 138–154. URL: https://doi.org/10.1007/978-3-031-72890-7_9.
> > > > > >
> > > > > > [8] Alan Ansell *et al*. Scaling sparse fine-tuning to large language models, 2024. URL: https://arxiv.org/abs/2401.16405.
> > > > > >
> > > > > > [9] Sean Lie. Cerebras architecture deep dive: First look inside the hardware/software co-design for deep learning. IEEE Micro, 43(3):18–30, May 2023. URL: https://doi.org/10.1109/MM.2023.3256384.

---

> > ### Comment · Reviewer_dMY7 · 2025-11-20
> > **Acknowledging theoretical justifications and ablation studies**
> >
> > We appreciate the theoretical justifications of the relation between parameter values and pretrained knowledge. I am in the process of reading the justification in details and will revert back soon. The new ablation studies are valuable. They clarify some of the query points.

---

> ### Author Response · Authors · 2025-11-24
> **Gentle Reminder to Reviewer dMY7**
>
> Thank you for your prompt acknowledgment! We hope that you have gone through our justification in detail and we are looking forward to a positive response and a reconsideration for an increased score!

---

> > ### Comment · Reviewer_dMY7 · 2025-11-26
> >
> > The theoretical justifications are interesting. The ablations are also valuable. We would like to see more adaptation to sparse implementation. Although the current arguments for it are accepted.

---

> > > ### Author Response · Authors · 2025-11-26
> > >
> > > Thanks a lot for appreciating and finding our theoretical justifications interesting and our ablations studies valuable! We are very happy to see that you have increased our score to an 8 and support the acceptance of our work!

---

### Official Review · Reviewer_xcfq · 2025-10-31

**Soundness:** 2
**Presentation:** 2
**Contribution:** 2
**Rating:** 2
**Confidence:** 4

**Summary:**

This paper proposes GaLLoP, a sparse fine-tuning method that selects a subset of parameters to update based on the ratio between their gradient magnitude and pretrained weight magnitude. The key idea is that parameters with large gradients (task-relevant) and small pretrained magnitudes (less tied to prior knowledge) can be fine-tuned to improve in-domain (ID) performance while mitigating catastrophic forgetting on out-of-domain (OOD) tasks. Experiments on eight commonsense reasoning datasets show that GaLLoP achieves comparable or better ID/OOD accuracy compared to existing sparse fine-tuning baselines such as SAFT and SpIEL.

**Strengths:**

1. Simple and straightforward design : The proposed scoring criterion is conceptually clear and easy to implement without architectural modification or additional modules.

2. Parameter-efficient fine-tuning : Only a small fraction of parameters (around <2%) are updated, yielding memory-efficient adaptation with no extra trainable layers.

3. Empirical stability : The method shows zero collapse or forgetting ratios across multiple seeds and densities, indicating robustness to optimization noise.

**Weaknesses:**

1. Lack of theoretical justification : The claim that gradient magnitude directly represents parameter sensitivity is overstated. Without a connection to curvature-based measures such as the Fisher Information Matrix or Hessian spectrum, it is unclear whether $\|\|g\|\|$ accurately captures parameter importance.

2. Unverified assumption about parameter magnitude : The interpretation that low-magnitude parameters encode less critical or “more adjustable” knowledge is speculative and not theoretically demonstrated. The connection between parameter norm and the nature of stored knowledge seems vague.

3. Non-typical ID/OOD setup : The so-called OOD evaluation is defined only as cross-dataset testing within a single commonsense reasoning benchmark family. This does not constitute a true domain-shift scenario and thus limits claims about generalization.

4. Limited experimental scope : Despite involving eight datasets, the experiments are confined to the commonsense reasoning domain, and only two model scales (Gemma-2B, LLaMA3-8B) are tested. Broader evaluation on diverse task types (e.g., reasoning, factual QA, math) or architectures would strengthen the argument.

5. Marginal gains over baselines : On Gemma-2B, GaLLoP performs nearly identically to SAFT; even on LLaMA3-8B, the reported improvement is modest (see Fig. 8 in Appendix E). The performance advantage appears minor and inconsistent, suggesting that the proposed selection rule may not offer substantial benefits over existing sparse fine-tuning methods.

**Questions:**

1. The paper assumes that gradient magnitude represents parameter sensitivity and that small-magnitude parameters correspond to more “adjustable” knowledge, but both claims lack theoretical or empirical justification. Could the authors provide supporting evidence such as theoretical relation between $\|\|g\|\|$, Fisher Information, or Hessian-based parameter sensitivity and analyses showing that parameter magnitude indeed aligns with knowledge stability or adaptability in pre-trained models?
2. The ID/OOD setup in this paper is based on cross-dataset evaluation within commonsense reasoning tasks, rather than across distinct domains. Could the authors clarify why this configuration should be regarded as out-of-distribution generalization rather than merely cross-task transfer? Additionally, have the authors examined whether GaLLoP’s advantages persist under more conventional domain-shift OOD settings?
3. Experimental coverage is limited to two baselines of different scales and a single task. Do the authors plan to test on other architectures or modalities to demonstrate general applicability of the proposed criterion?

---

> ### Author Response · Authors · 2025-11-19
> **Response to Reviewer xcfq's Concerns [Part 1]**
>
> Thank you for your constructive feedback! We are happy to hear that you found our scoring criterion *conceptually clear* and *easy to implement* with a *simple and straightforward design*, our adaptation method *memory-efficient* and *empirically stable*, indicating its *robustness to optimization noise*! We now address your concerns below:
>
> > **W1, W2, Q1: Theory**
>
> We thank you for your suggestions to incorporate theoretical justifications for our parameter selection criteria! We have now added a **new section on theory as Appendix I of the updated version of our paper** wherein as suggested by you and reviewer `dMY7`, we now discuss the connections between gradient magnitude and parameter sensitivity and show how parameter magnitude relates to the pre-trained knowledge stored in a model. Empirical justifications showing how fine-tuning the top-$\rho$\% of the low-magnitude parameters enhances a model's generalizability have already been shown and discussed in Section 3 of our paper and we have also already provided a full ablation study in Appendix H of our paper wherein we demonstrate that fine-tuning the top-$\rho$\% of the model parameters which possess low-magnitudes and high task gradients leads to a dominant Pareto front for both ID as well as OOD performance, across all density levels and experimental runs, over fine-tuning the top-$\rho$\% model parameters based on whether they have either a) low-magnitudes or b) high task gradients only.
>
> To summarize our theoretical justifications, we are also discussing two key ideas below (though we encourage you to check out Appendix I which contains a more complete version):

---

> > ### Author Response · Authors · 2025-11-19
> > **Response to Reviewer xcfq's Concerns [Part 2]**
> >
> > Consider a model with a parameter vector $\boldsymbol{\theta} \in \mathbb{R}^D$ which has been pre-trained on a dataset $\mathcal{D}^p = {\\{(\boldsymbol{x}^p\_n, \boldsymbol{y}^p\_n)\\}}\_{n = 1}^{N}$, where, $\boldsymbol{x}^p_n \in \mathbb{R}^{P}$ and $\boldsymbol{y}^p_n \in \mathbb{R}^{Q}$, with a pre-training loss function $\mathcal{L}^p(\boldsymbol{x}^p_n, \boldsymbol{y}^p_n; \boldsymbol{\theta})$. This model will now be fine-tuned on a dataset $\mathcal{D}^f = {\\{(\boldsymbol{x}^f_n, \boldsymbol{y}^f_n)\\}}_{n = 1}^{N}$, where, $\boldsymbol{x}^f_n \in \mathbb{R}^{P}$ and $\boldsymbol{y}^f_n \in \mathbb{R}^{Q}$, with a fine-tuning loss function $\mathcal{L}^f(\boldsymbol{x}^f_n, \boldsymbol{y}^f_n; \boldsymbol{\theta})$.
> >
> > **Criterion 1: Connection between Gradient Magnitude and Parameter Sensitivity**
> >
> > For a conditional probability distribution $p_{\boldsymbol{\theta}}(\boldsymbol{y}^f|\boldsymbol{x}^f)$ generated by the model over the output $\boldsymbol{y}^f \in \mathbb{R}^{Q}$, given an input $\boldsymbol{x}^f \in \mathbb{R}^{P}$, one way to estimate the importance of each of its parameters is to determine the amount by which the model's prediction would change upon perturbing its parameters by a small amount ($\Delta\boldsymbol{\theta}$), i.e., via the Kullback-Leibler (KL) divergence $KL (p_{\boldsymbol{\theta}}(\boldsymbol{y}^f|\boldsymbol{x}^f) || p_{\boldsymbol{\theta}+\Delta\boldsymbol{\theta}}(\boldsymbol{y}^f|\boldsymbol{x}^f))$.
> >
> > For an infinitesimally small perturbation $\Delta\boldsymbol{\theta} \rightarrow 0$, using the second-order Taylor series expansion, the expectation of the said KL divergence over the inputs $\boldsymbol{x}$ can be given by [1]:
> >
> > $$\mathbb{E}\_{\boldsymbol{x}^f \sim p(\boldsymbol{x}^f)}[KL (p\_{\boldsymbol{\theta}}(\boldsymbol{y}^f|\boldsymbol{x}^f) || p_{\boldsymbol{\theta}+\Delta\boldsymbol{\theta}}(\boldsymbol{y}^f|\boldsymbol{x}^f))] = \Delta\boldsymbol{\theta}^T \boldsymbol{F}_{\boldsymbol{\theta}} \Delta\boldsymbol{\theta} + \mathcal{O}(\Delta\boldsymbol{\theta}^3),$$
> >
> > where, $\boldsymbol{F}\_{\boldsymbol{\theta}}$ is the (theoretical) Fisher information matrix and is given by [2]:
> > $$\boldsymbol{F}\_{\boldsymbol{\theta}} = \mathbb{E}\_{\boldsymbol{x}^f\sim p(\boldsymbol{x}^f)}[\mathbb{E}\_{\boldsymbol{y}^f\sim p\_{\boldsymbol{\theta}}(\boldsymbol{y}^f|\boldsymbol{x}^f)}[\nabla_{\boldsymbol{\theta}}\log p_{\boldsymbol{\theta}}(\boldsymbol{y}^f|\boldsymbol{x}^f)\nabla_{\boldsymbol{\theta}}\log p_{\boldsymbol{\theta}}(\boldsymbol{y}^f|\boldsymbol{x}^f)^T]].$$
> >
> > It is intractable to compute the full $D \times D$ Fisher information matrix for deep neural networks and hence, it is commonly approximated as a diagonal matrix, i.e., a $D$-dimensional vector, which is given by [2, 4]:
> > $$\hat{\boldsymbol{F}\_{\boldsymbol{\theta}}} \approx \mathbb{E}\_{\boldsymbol{x}^f\sim p(\boldsymbol{x}^f)}[\mathbb{E}\_{\boldsymbol{y}^f\sim p\_{\boldsymbol{\theta}}(\boldsymbol{y}^f|\boldsymbol{x}^f)}[\nabla_{\boldsymbol{\theta}}\log p_{\boldsymbol{\theta}}(\boldsymbol{y}^f|\boldsymbol{x}^f)]^2].$$
> >
> > In a practical scenario, we do not have access to the full input distribution $p(\boldsymbol{x}^f)$ and instead, only have access to a finite set of input samples $\\{ \boldsymbol{x}^f_n \\}\_{n = 1}^N$ which can together serve as an approximation of $p(\boldsymbol{x}^f)$. Hence, we now get that [2, 3, 4]:
> > $$\hat{\boldsymbol{F}\_{\boldsymbol{\theta}}} \approx \dfrac{1}{N}\sum_{n = 1}^N[\mathbb{E}\_{\boldsymbol{y}^f\sim p\_{\boldsymbol{\theta}}(\boldsymbol{y}^f|\boldsymbol{x}^f_n)}[\nabla\_{\boldsymbol{\theta}}\log p\_{\boldsymbol{\theta}}(\boldsymbol{y}^f|\boldsymbol{x}^f_n)]^2].$$
> >
> > Furthermore, in supervised learning (as is the case here), we already have access to the ground truths $\\{\boldsymbol{y}^f\_n\\}\_{n = 1}^N$ corresponding to all the inputs $\\{\boldsymbol{x}^f\_n\\}\_{n = 1}^N$. This allows us to replace $\mathbb{E}\_{\boldsymbol{y}^f\sim p\_{\boldsymbol{\theta}}(\boldsymbol{y}^f|\boldsymbol{x}^f_n)}[\nabla_{\boldsymbol{\theta}}\log p_{\boldsymbol{\theta}}(\boldsymbol{y}^f|\boldsymbol{x}^f_n)]^2$ by $[\nabla_{\boldsymbol{\theta}}\log p_{\boldsymbol{\theta}}(\boldsymbol{y}^f_n|\boldsymbol{x}^f_n)]^2$. Finally, since $\mathcal{L}^f$ is generally the cross-entropy loss, $\mathcal{L}^f(\boldsymbol{x}^f_n, \boldsymbol{y}^f_n; \boldsymbol{\theta}) = -\log p_{\boldsymbol{\theta}}(\boldsymbol{y}^f_n|\boldsymbol{x}^f_n)$. Hence, we now get that [3, 4]:$$\hat{\boldsymbol{F}\_{\boldsymbol{\theta}}} = \dfrac{1}{N}\sum_{n = 1}^N [\nabla_{\boldsymbol{\theta}}\mathcal{L}^f(\boldsymbol{x}^f_n, \boldsymbol{y}^f_n; \boldsymbol{\theta})]^2,$$
> > where, $\hat{\boldsymbol{F}\_{\boldsymbol{\theta}}}$ is referred to as the empirical Fisher.
> >
> > The usage of the empirical Fisher is preferred in practice since it is more tractable and efficient to compute than the theoretical Fisher information matrix [3, 4].

---

> > > ### Author Response · Authors · 2025-11-19
> > > **Response to Reviewer xcfq's Concerns [Part 3]**
> > >
> > > Next, from the expression for the empirical Fisher, we can see that the importance of a model parameter, as measured by the empirical Fisher, is directly proportional to the average of the square of the gradient of the loss w.r.t. that model parameter. Let $\boldsymbol{g} \in \mathbb{R}^D$ denote the gradient vector. This implies that:$$\hat{\boldsymbol{F}_{\boldsymbol{\theta}}} \propto (\boldsymbol{g})^2.$$
> > >
> > > Returning back to our initial objective, we can now see that in order to enhance our model's ID generalizability, we need to select the top-$\rho$\% of the parameters which possess the highest (empirical) Fisher information for the given downstream task. Using the aforementioned proportionality, this translates to selecting the top-$\rho$\% of the parameters with the highest $(\boldsymbol{g})^2$ values. Now, mathematically, it is evident that $\text{abs}(\boldsymbol{g}) \propto (\boldsymbol{g})^2$ (where, $\text{abs}(.)$ computes the element-wise absolute value of a vector) and hence, selecting the top-$\rho$\% of the parameters with the highest $\text{abs}(\boldsymbol{g})$ values efficiently (since the computational cost associated with computing the square of $\boldsymbol{g}$ is avoided) satisfies our first criterion.
> > >
> > > **Criterion 2: Connection between Parameter Magnitude and Pre-Trained Knowledge**
> > >
> > > In order to draw connections between parameter magnitude and pre-trained knowledge, we can consider the problem of figuring out which model parameters when pruned would result in a minimal impact on the pre-trained loss.
> > >
> > > For the pre-training loss function $\mathcal{L}^p(\boldsymbol{x}^p_n, \boldsymbol{y}^p_n; \boldsymbol{\theta})$, using the Taylor series expansion, we can state that:$$\Delta\mathcal{L}^p(\boldsymbol{x}^p\_n, \boldsymbol{y}^p\_n; \boldsymbol{\theta}) = \sum\_{i}\frac{\partial \mathcal{L}^p(\boldsymbol{x}^p\_n, \boldsymbol{y}^p\_n; \boldsymbol{\theta})}{\partial \boldsymbol{\theta}\_{i}} \Delta \boldsymbol{\theta}\_{i} + \mathcal{O}(\lVert \Delta\boldsymbol{\theta} \rVert^2).$$
> > >
> > > where, $\mathcal{O}(\lVert \Delta\boldsymbol{\theta} \rVert^2)$ denotes the second- (containing the Hessian) and higher-order terms. These are intractable to compute for deep neural networks [6] and hence, we focus solely on a first-order approximation. Hence, the influence of a parameter on the pre-training loss is given by:$$\Delta\mathcal{L}\_i^p(\boldsymbol{x}^p\_n, \boldsymbol{y}^p\_n; \boldsymbol{\theta}) \approx \left|\frac{\partial \mathcal{L}^p(\boldsymbol{x}^p\_n, \boldsymbol{y}^p\_n; \boldsymbol{\theta})}{\partial \boldsymbol{\theta}\_{i}} \Delta\boldsymbol{\theta}\_{i}\right|.$$
> > >
> > > Now, the model parameters which would have a minimal impact on the pre-training loss are in fact, those parameters which can potentially be pruned in the pre-trained model. Accordingly, we can study the influence of a parameter $\boldsymbol{\theta}\_{i}$ on the pre-training loss by computing the change affected by it in the value of the loss function when it gets pruned, i.e., for the case when $\Delta \boldsymbol{\theta}\_{i} = \boldsymbol{\theta}\_{i} - 0 = \boldsymbol{\theta}\_{i}$.
> > >
> > > Although we cannot possibly have access to the full pre-training dataset $\mathcal{D}^p$ for the model, a reasonable assumption is that the model's (pre-) training would have converged and hence, its pre-training gradients are bound to be quite small. On the other hand, the model's (pre-trained) parameters can be either large or small. Mathematically, since: $$\left|\boldsymbol{\theta}\_{i}\right| \gtrapprox \left|\frac{\partial \mathcal{L}^p(\boldsymbol{x}^p\_n, \boldsymbol{y}^p\_n; \boldsymbol{\theta})}{\partial \boldsymbol{\theta}\_{i}}\right|,$$
> > > we get that:
> > > $$\Delta\mathcal{L}\_i^p(\boldsymbol{x}^p\_n, \boldsymbol{y}^p\_n; \boldsymbol{\theta}) \propto \left|\boldsymbol{\theta}\_{i}\right|.$$
> > >
> > > Therefore, parameters with the smallest pre-trained magnitudes are the least important for the pre-trained model since pruning them leads to the least impact on the pre-training loss. In other words, these parameters are minimally disruptive to the pre-trained knowledge and potentially represent the under-utilized capacity of the pre-trained model.
> > >
> > > Therefore, fine-tuning these low-magnitude parameters can potentially be a way to preserve the pre-trained knowledge stored in the model.

---

> > > > ### Author Response · Authors · 2025-11-19
> > > > **Response to Reviewer xcfq's Concerns [Part 4]**
> > > >
> > > > > **W3, W4, Q2, Q3: Experimentation**
> > > >
> > > > In the context of fine-tuning, we would like to clarify that an Out-Of-Distribution (OOD) task refers to a related task which possesses a distribution different from the fine-tuning task and OOD is *not solely* limited to domain-shifts and in fact, also refers to detecting new classes and cross-dataset transfer [7]. In fact, a task which just possesses a domain/distribution-shift is a weak OOD task since it resembles the fine-tuning (ID) task response format and contains the entire set/a subset of classes present in the fine-tuning (ID) task dataset. This is also the reason why the WiSE-FT paper [8] which performs experiments on distribution/domain-shifts of the ImageNet dataset states that it avoids referring to them as OOD datasets and simply labels them as distribution shifts.
> > > >
> > > > In contrast, our experimental setup spans 8 reasoning and question answering datasets which are not simply distributional shifts but possess different classes and different numbers of classes (answer options), different response formats, and incorporate a wide variety of different tasks which test models on grade-school level science questions, physical properties of objects, day-to-day activities of humans, general knowledge facts, social and emotional intelligence, and pronoun resolution (further details have already been provided in Appendix B of our paper). A similar choice of ID/OOD datasets has also been used in a recent paper on overtraining (see Fig. 2 on page 2 of the paper) [9]. Furthermore, in contrast to previous studies which perform experiments considering only 1-2 density levels and/or two models from a single model architecture family and with similar sizes [7, 10, 11], we perform experiments on two models with vastly different model sizes (8B and 2B), different model architectures (LLaMA and Gemma), and cover all the 8 possible ID and OOD combinations of these datasets for all our 4 density levels. We show that GaLLoP consistently improves over (/matches) the ID and OOD performance of the competing algorithms and is moreover, the only algorithm which allows for the attainment of 0\% forget ratios and 0\% collapse rates across all density levels and experimental runs. All these experimental results only strengthen our claims on generalizability.
> > > >
> > > > Therefore, we respectfully disagree with the fact that our experimentation has a limited scope. Reviewer `dMY7` has also appreciated that *our strategy is well-motivated by related experiments* and supported by *exhaustive experimental results which demonstrate its efficacy*.
> > > >
> > > > We would also like to highlight that while we are currently working on experiments on additional architectures and modalities, our computational resources are currently limited. Nevertheless, we believe that our current experiments are extensive enough and hope to add new experiments in the final version of our paper.

---

> > > > > ### Author Response · Authors · 2025-11-19
> > > > > **Response to Reviewer xcfq's Concerns [Part 5]**
> > > > >
> > > > > > **W5: Gains over Baselines**
> > > > >
> > > > > We respectfully disagree that the performance gains of GaLLoP over SAFT are minor, inconsistent, and/or modest.
> > > > >
> > > > > For LLaMA3 8B, GaLLoP demonstrates strong performance gains and forms a dominant Pareto front for both ID and OOD accuracy (on average) over all the models fine-tuned and/or edited with the competing algorithms across all density levels. Models fine-tuned with GaLLoP consistently surpass those fine-tuned with SAFT with a high average margin of roughly 10\% (for both ID and OOD accuracy) which only narrows down for the highest density level. These accuracy gains over SAFT are also reflected across a majority of individual experimental runs with even higher gains ranging from 20 - 50\% for some datasets and density levels (see Fig. 8 of Appendix E) and as we already mentioned in Appendix E, it is only the SIQA dataset, for which the performance yielded by SAFT is comparable to that yielded by GaLLoP and only the BoolQ dataset, on which models fine-tuned with SAFT outperform those fine-tuned with GaLLoP for the highest density levels.
> > > > >
> > > > > Next, for Gemma 2B, models fine-tuned with GaLLoP are the only ones which consistently attain a high and balanced ID and OOD performance (on average) across all density levels. In fact, Gemma 2B models fine-tuned with GaLLoP form a dominant Pareto front for both ID and OOD accuracy (on average as well as across individual experimental runs) over those fine-tuned with SAFT, across all density levels and datasets (except only BoolQ) with high average margins of roughly 5-10\% which only narrow down for the highest density level.
> > > > >
> > > > > Note that the case of BoolQ is an interesting one and we have already analyzed in Appendix E.1 how it is easy to take advantage of its skewed response distribution to gain on performance (as with SAFT) and how all the models fine-tuned with GaLLoP desirably do not memorize the response distribution and instead try to understand and correctly answer even those questions whose responses fall in the tail of the response distribution for BoolQ. Additionally, as we have already discussed in Appendix H, the narrowing down of the performance gap for the highest density level is expected as all static sparse fine-tuning (SpFT) algorithms such as GaLLoP and SAFT must converge to FFT performance in the asymptotic limit ($\rho \rightarrow 100\\%$). Nevertheless, our experiments show that by combining the strengths of high task gradients and low-magnitudes, fine-tuning with GaLLoP is expected to continue forming a performance upper bound over other static SpFT algorithms even in the asymptotic limit ($\rho \rightarrow 100\\%$).
> > > > >
> > > > > Furthermore, **accuracy is not the only barometer of performance for an algorithm**. This is indeed the reason why we compute and examine the forget ratios and collapse rates of models fine-tuned with GaLLoP and those fine-tuned and/or edited with the competing algorithms and find that GaLLoP is the only algorithm which consistently yields 0\% forget ratios and 0\% collapse rates -- this observation holds true for both LLaMA3 8B and Gemma 2B models, across all density levels and experimental runs. We appreciate your acknowledgement of the fact that this *empirical stability* of GaLLoP indicates its *robustness to optimization noise*.
> > > > >
> > > > > In fact, GaLLoP's substantial benefits over competing SpFT techniques are also supported by reviewer `dMY7` and reviewer `RmwJ`. Reviewer `dMY7` appreciates that *our empirical results demonstrate the superiority of our proposed technique over related SpFT techniques*. Further, reviewer `RmwJ` appreciates that our *results consistently show a dominant ID/OOD Pareto front for GaLLoP, with particularly strong averages on LLaMA-3 8B and balance on Gemma-2B* and also states that `*addressing the ever-present tension between adaptation and retention, the method delivers strong OOD gains while curbing overfitting that plagues several popular baselines—an outcome that matters for reliable deployment of fine-tuned LLMs. Because GaLLoP is simple, model-agnostic, and efficient, it lowers the barrier to adopting sparse fine-tuning in practice*'.
> > > > >
> > > > > **We hope that our response clarifies your concerns and we kindly request you to reconsider and increase your score.**

---

> > > > > > ### Author Response · Authors · 2025-11-19
> > > > > > **References**
> > > > > >
> > > > > > [1] Razvan Pascanu *et al.*. Revisiting natural gradient for deep networks, 2014. URL: https://arxiv.org/abs/1301.3584.
> > > > > >
> > > > > > [2] James Martens. New insights and perspectives on the natural gradient method. Journal of Machine Learning Research, 21(146):1–76, 2020. URL: http://jmlr.org/papers/v21/17-678.html.
> > > > > >
> > > > > > [3] Sidak Pal Singh *et al.*. Woodfisher: Efficient second-order approximation for neural network compression. In H. Larochelle, M. Ranzato, R. Hadsell, M.F. Balcan, and H. Lin (eds.), Advances in Neural Information Processing Systems, volume 33, pp. 18098–18109. Curran Associates, Inc., 2020. URL: https://proceedings.neurips.cc/paper_files/paper/2020/file/d1ff1ec86b62cd5f3903ff19c3a326b2-Paper.pdf.
> > > > > >
> > > > > > [4] Yi-Lin Sung *et al.*. Training neural networks with fixed sparse masks. In A. Beygelzimer, Y. Dauphin, P. Liang, and J. Wortman Vaughan (eds.), Advances in Neural Information Processing Systems, 2021. URL: https://openreview.net/forum?id=Uwh-v1HSw-x.
> > > > > >
> > > > > > [5] Yann LeCun *et al.*. Optimal brain damage. In D. Touretzky (ed.), Advances in Neural Information Processing Systems, volume 2. Morgan-Kaufmann, 1989. URL: https://proceedings.neurips.cc/paper_files/paper/1989/file/6c9882bbac1c7093bd25041881277658-Paper.pdf.
> > > > > >
> > > > > > [6] Namhoon Lee *et al.*. SNIP: Single-shot network pruning based on connection sensitivity. In International Conference on Learning Representations, 2019. URL: https://openreview.net/forum?id=B1VZqjAcYX.
> > > > > >
> > > > > > [7] Bac Nguyen *et al.*. SAFT: Towards out-of-distribution generalization in fine-tuning. In Computer Vision – ECCV 2024: 18th European Conference, Milan, Italy, September 29–October 4, 2024, Proceedings, Part LXIX, pp. 138–154. URL: https://doi.org/10.1007/978-3-031-72890-7_9.
> > > > > >
> > > > > > [8] Mitchell Wortsman *et al.*. Robust fine-tuning of zero-shot models. In Proceedings of the IEEE/CVF Conference on Computer Vision and Pattern Recognition (CVPR), pp. 7959–7971, June 2022. URL: https://openaccess.thecvf.com/content/CVPR2022/papers/Wortsman_Robust_Fine-Tuning_of_Zero-Shot_Models_CVPR_2022_paper.pdf.
> > > > > >
> > > > > > [9] Jacob Mitchell Springer *et al.*. Overtrained language models are harder to fine-tune. In Forty-second International Conference on Machine Learning, 2025. URL: https://openreview.net/forum?id=YW6edSufht.
> > > > > >
> > > > > > [10] Alan Ansell *et al.*. Scaling sparse fine-tuning to large language models, 2024. URL: https://arxiv.org/abs/2401.16405.
> > > > > >
> > > > > > [11] Shih yang Liu *et al.*. DoRA: Weight-decomposed low-rank adaptation. In Forty-first International Conference on Machine Learning, 2024. URL: https://openreview.net/forum?id=3d5CIRG1n2.

---

> > > > > > ### Comment · Reviewer_xcfq · 2025-11-25
> > > > > >
> > > > > > **Comment to W5: Gains over Baselines**
> > > > > >
> > > > > > **1. Limited Baselines**
> > > > > >
> > > > > > Your choice of baselines appears overly restrictive. Although GaLLoP is a sparse fine-tuning (SpFT) method, it should not be benchmarked only against methods that use similar gradient-based parameter selection rules. What matters for evaluation is effective parameter budget, not the internal mechanism used to select parameters.
> > > > > >
> > > > > > There exist other strong and widely-used sparse fine-tuning baselines—such as S²FT [1]—as well as non-sparse PEFT methods like LoRA, QLoRA, and parallel adapter variants, which often achieve competitive or superior performance at comparable or smaller trainable-parameter ratios. Unless GaLLoP’s active parameter set matches SAFT’s sparsity ratio exactly, excluding these baselines makes it difficult to assess the practical competitiveness of the method.
> > > > > >
> > > > > > To fairly position GaLLoP among existing techniques, the evaluation should compare against all methods with similar numbers of trainable parameters, regardless of whether they use gradient-based selection or adapter-based updates. Without such comparisons, it is unclear whether GaLLoP’s gains reflect advantages of the method itself or simply differences in sparsity and parameter budgets.
> > > > > >
> > > > > > **2. Absence of detailed numerics and statistical validation for highly overlapping behavior in forgetting and memorization**
> > > > > >
> > > > > > The paper does not report any numerical forget or collapse values nor any statistical measures (standard deviations, confidence intervals, significance tests). All plots show both GA LLoP and SAFT at exactly 0% forget ratio and 0% collapse rate across datasets, density levels, and random seeds, making them statistically indistinguishable. Without numerical metrics, the claim of superiority over SAFT in catastrophic forgetting or memorization cannot be substantiated.
> > > > > >
> > > > > >
> > > > > >
> > > > > > [1] Yang, Xinyu, et al. "S $^{2} $ FT: Efficient, scalable and generalizable LLM fine-tuning by structured sparsity." Advances in Neural Information Processing Systems 37 (2024): 59912-59947.

---

> > > > > ### Comment · Reviewer_xcfq · 2025-11-25
> > > > >
> > > > > **Comment on W3/W4 (OOD setup and scope)**
> > > > >
> > > > > While your clarification is noted, the current evaluation is still closer to cross-dataset testing within one domain than to widely accepted OOD setups. In the robustness literature, OOD typically refers to explicit distribution shifts, such as:
> > > > >
> > > > > - Natural distribution shifts with the same label space (e.g., ImageNet → ImageNet-V2/R/A/C/ObjectNet).
> > > > > - Domain/subpopulation shifts (e.g., WILDS benchmark).
> > > > > - Cross-domain evaluation in NLP (e.g., commonsense QA -> factual QA, biomedical QA, math QA).
> > > > >
> > > > > Your datasets all belong to the same commonsense reasoning family, with similar linguistic style and knowledge type. This is a valid cross-dataset generalization test, but it does not match standard OOD definitions.
> > > > > I suggest adding at least one established OOD scenario to align with standard robustness evaluation practice.

---

> > > > ### Comment · Reviewer_xcfq · 2025-11-25
> > > > **Comments on Theoretical Justification (Part 2: Parameter Magnitude & Pre-trained Knowledge)**
> > > >
> > > > **1. Unjustified Removal of Hessian and Higher-order Terms**
> > > >
> > > > Your argument relies on a first-order Taylor approximation of the change in pre-training loss when pruning a parameter:
> > > > $\Delta L^p(\theta_i) \approx \frac{\partial L^p}{\partial \theta_i}\,\Delta \theta_i$.
> > > >
> > > > where $\nabla \theta_i = -\theta$. However, pruning constitutes a non-infinitesimal perturbation, and the dropped terms—including the Hessian contribution $\frac{1}{2}\,\theta_i^2\,H_{ii}$, and the cross-terms—are of comparable or greater magnitude in deep nonlinear networks. Simply stating that these terms are “intractable” does not justify omitting them: intractable does not imply negligible. Without providing a proper theoretical bound, the arbitrary approximation cannot be used to reason about parameter sensitivity.
> > > >
> > > > **2. Unsupported Assumptions Regarding Pre-trained Gradients and Parameter Magnitude**
> > > >
> > > > The derivation further assumes:
> > > > - The pre-trained model has “converged,” so ${\partial L}/{\partial \theta_i}$ is “quite small.”
> > > > - Therefore, the influence of pruning is dominated by the magnitude $\|\theta\|$, implying that smaller-magnitude parameters encode less important pretraining knowledge.
> > > >
> > > > Both assumptions are informal, and inconsistent with modern large-scale pretraining. Recent large language models do not converge to stationary points where per-parameter gradients vanish; stochastic optimization, weight decay, and early stopping all prevent such behavior.
> > > >
> > > > These assumptions are conveniently introduced to obtain the desired conclusion—that small parameters correspond to low-impact knowledge—the resulting claim is not theoretically supported.

---

> ### Comment · Reviewer_xcfq · 2025-11-25
> **Comments on Theoretical Justification (Part 1: Connection between Gradient Magnitude and Parameter Sensitivity)**
>
> I do know that empirical Fisher can serve as the approximate of true Fisher. What I want to clarify is that the true measure of empirical Fisher is not dominated by the norm of the average gradient. Formally, $\hat F = \mathrm{Cov}(g_n) + \bar g \bar g^\top$, so the empirical Fisher includes both
> (i) the squared mean gradient and (ii) the covariance (variance and disagreement) of per-sample gradients.
>
> The empirical Fisher is deeply tied the "curvature of the loss landscape", not just the gradient magnitude.
> A parameter direction can have tiny gradient but very high curvature (sharp minimum); $\hat F$ will tend to capture this curvature, even though the mean gradient norm is negligible.
>
> Therefore he second term (ii) can dominate, particularly when gradients cancel on average, when the data distribution induces large gradient variability, or when curvature is high but the mean gradient is small.

---

> ### Author Response · Authors · 2025-11-25
> **Response to New Comments posted by Reviewer xcfq**
>
> We thank the reviewer for the comments and we now address all the reviewer's concerns below:
>
> > **1. Theory**:
>
> Further to our previous response, we would firstly like to clarify that the assumptions made by us in our theoretical justifications are in line with a long list of previous studies already published in the literature [1,2,3,4,5,6,7,9,10]. Moreover, as previous studies have also acknowledged, it is not possible to establish such theoretical connections without making such assumptions. Additionally, our experiments and so do those of other related previous works (gradient-based or magnitude-based; already cited in our paper) show that the theoretical connections established by us do indeed hold true in practice. We therefore respectfully disagree with the reviewer that our theoretical justifications are not well supported.
>
> > **2. Experimental Setup**:
>
> We appreciate that the reviewer has taken note of our clarification and has acknowledged that our experimental setup is a valid cross-dataset generalization test. Further to our clarifications in the previous response, we would like to clarify that there is no consensus amongst previous studies [7,8,9] which perform different kinds of experiments to demonstrate gains in OOD generalizability. Furthermore, in line with our previous response, we would again like to clarify that in contrast to previous studies which perform experiments considering only 1-2 density levels and/or two models from a single model architecture family and with similar sizes [7, 10, 11], we perform experiments on two models with vastly different model sizes (8B and 2B), different model architectures (LLaMA and Gemma), and cover all the 8 possible ID and OOD combinations of these datasets for all our 4 density levels while ensuring that all PEFT algorithms fine-tune the same number of model parameters (same density levels) and are applied across the same model layers. We thank the reviewer for the suggestion of adding experiments on another OOD scenario and are working towards adding the same in our final version, as our computational resources are currently limited.
>
> > **3. Gains over Baselines**:
>
> Firstly, we respectfully disagree with the reviewer that our choice of baselines is overly restrictive. We would like to clarify that as we have already discussed in our paper, we compare GaLLoP against LoRA, DoRA, SAFT, SpIEL, SpFT of only low-magnitude model parameters, SpFT of only high-magnitude model parameters, WiSE-FT, LiNeS, and FFT. Secondly, we would also like to clarify that we have already performed our experiments considering the same number of trainable parameters for all our PEFT algorithms. In our paper, we have already clarified this in section 5.5: ''To ensure a fair comparison, we apply each fine-tuning algorithm across all the same transformer layers and maintain the same density level for a given experiment'' and all our figures in the results section of our paper show that this is indeed the case since we provide values of all our performance metrics for all PEFT algorithms across all our density levels. Thirdly, we respectfully disagree with the reviewer on the fact that we do not report any numerical forget or collapse values nor any statistical measures. On the contrary, we explicitly show and discuss the numerical values of all forget ratios and collapse rates for all competing algorithms in Sections 6.2 and 6.3 of our paper. Furthermore, we show and explicitly state and discuss in Section 6.2 that it is only GaLLoP that attains 0\% forget ratios across all density levels; Fig. 4a) reveals that SAFT leads to a non-zero forget ratio for a 0.47\% density level and hence, SAFT does not attain 0\% forget ratios across all density levels. Moreover, Section 6.4 of our paper provides the mean, median, and interquartile range values for competing algorithms and GaLLoP and shows that GaLLoP indeed attains the highest stability.

---

### Official Review · Reviewer_EVgh · 2025-11-03

**Soundness:** 1
**Presentation:** 2
**Contribution:** 2
**Rating:** 2
**Confidence:** 4

**Summary:**

Summary: This paper proposes a method to mitigate forgetting, by choosing whether to update a particular parameter based on a score that is its ( ( gradient_magnitude ) / ( magnitude + epsilon) ) - the top "k" highest scoring ones are updated in that iteration and others are not.

Evaluation of forgetting is done by taking 8 datasets, fine-tuning on one ("IID") and measuring performance on the others ("OOD") - and averaging this over round-robin selection of IID/OOD.

**Strengths:**

The paper is clearly written.

**Weaknesses:**

(A) comparisons: Mitigating catastrophic forgetting is a rich field and this paper is missing comparisons against several state of the art methods, for example:
(1) https://arxiv.org/abs/1612.00796 (classic method, works very well on modern LLMs as well)
(2) https://arxiv.org/abs/2407.20999 (another method that selects individual parameters - based on a different logic - to update, instead of updating all parameters)
(3) https://icml.cc/virtual/2025/poster/46655 ( more recent paper and method, contains overview of preceeding)

Indeed this method is only compared against other methods that update fewer parameters, but if the main aim is mitigating catastrophic forgetting then a much more robust comparison against other approaches like the three above, and also others, is needed.

(B) reasoning: what is the intuition behind this method ? Eg Fig 1 shows that fine-tuning the lowest magnitude weights gives gains, but what about updating a randomly chosen subset of  weights ?

(C) Novelty: there are many methods that choose only a subset of parameters to update based on values, gradients or a combination thereof; this paper is yet another in that line (albeit with a slightly variant selection logic). The novelty from an ideas perspective is low.

**Questions:**

Fig 3: It is very surprising that OOD performance improves over Vanilla - this is the opposite of forgetting, as it means that fine-tuning on one dataset improves performance on 7 others when doing e.g. LoRA or other such. How is this possible ? It seems to go against what other papers in forgetting show - that fine-tuning on a dataset improves ID but degrades OOD.

---

> ### Author Response · Authors · 2025-11-19
> **Response to Reviewer Evgh's Concerns [Part 1]**
>
> Thank you for your constructive feedback! We are happy to hear that you found our paper '*clearly written*'! We now address your concerns below:
>
> > **W1: Comparisons**
>
> Firstly, we would like to clarify that the main aim of our sparse fine-tuning algorithm (GaLLoP) is to enhance the ID as well as OOD generalizability of a model. As shown in our paper, enhancing a model's generalizability does not *solely* come about by preventing catastrophic forgetting: boosting downstream task performance, preventing memorization, and ensuring stability, are all equally important and complementary factors to be considered. As a concrete example, note that by definition, the forget ratio (quantifies the extent of catastrophic forgetting; see (6) in Section 5.4 of our paper) measures the drop in OOD performance of a fine-tuned model relative to the performance of its vanilla counterpart, and hence, even though a model may attain a 0\% forget ratio, it can still undergo severe memorization of patterns present in the fine-tuning dataset, which leads to a non-zero collapse rate (for e.g., for the first three density levels, Gemma 2B models fine-tuned with SpIEL show 0\% forget ratios (Fig. 4b) but non-zero and increasingly high collapse rates (Fig. 5b)). In fact, our experiments show that GaLLoP is the only fine-tuning algorithm which not only consistently improves upon (/matches) the ID and OOD performance of several leading PEFT and model editing techniques but also attains 0\% forget ratios and 0\% collapse rates across all density levels and experimental runs.
>
> Secondly, two of the algorithms that you have suggested for comparison: EWC [1] and MoFo [2] have been developed with the aim of mitigating forgetting in a continual learning scenario. We thank you for suggesting the exploration of this field. However, for our work, we focussed on experiments in a static setting following a round-robin approach wherein we fine-tuned models on a single dataset and evaluated their performance on the others to examine their ID and OOD performance across all possible task combinations and across all density levels. Exploring the utility of GaLLoP in a continual learning scenario is beyond the current scope of our paper but represents an interesting direction for future work.
>
> Thirdly, the remaining algorithm which you have suggested for comparison, FLOW [3], operates in the sample space rather than the parameter space. Hence, it represents a completely orthogonal approach to ours and even other PEFT algorithms which operate in the parameter space (the current focus of our work), and is not relevant for comparison.
>
> We appreciate your acknowledgement of the fact that we indeed compare our algorithm (GaLLoP) with other algorithms which update fewer parameters. Reviewer `dMY7` has also appreciated that '*our strategy is well-motivated*' and supported by '*exhaustive experimental results which demonstrate its efficacy*' and '*superiority over related SpFT techniques*'.
>
> > **W2 : Reasoning**
>
> In Section 3 of our paper, we have already provided an intuitive explanation behind our algorithm. To achieve our dual aim, we fine-tune the model parameters with the largest gradient magnitudes on the downstream task, to minimize the fine-tuning loss and fasten convergence to the task optimum (to enhance ID generalizability), and yet at the same time, ensure that we fine-tune only those such high-gradient parameters which have the lowest pre-trained magnitudes (to enhance OOD generalizability), since low-magnitude parameters are the under-utilized capacity of the pre-trained model and fine-tuning them is generally considered to be minimally disruptive to pre-trained knowledge. While the former, gradient-based, criterion is agreed upon in theory [4, 5], we empirically demonstrate that the latter, magnitude-based, criterion holds true in practice by showing that fine-tuning low-magnitude parameters yields gains in generalizability in contrast to fine-tuning high-magnitude parameters which does not (see Fig. 1).
>
> Furthermore, in line with suggestions put forward by reviewer `xcfq` and reviewer `dMY7`, we have now added a theory section in the appendix (Appendix I in the updated version of our paper) wherein we discuss how gradient magnitudes can be considered as a measure of parameter importance w.r.t. the Fisher information matrix and further analyze how low-magnitude parameters are connected to pre-trained knowledge. We encourage you to check out the new theory section!
>
> Finally, as far as the question of updating a randomly selected subset of weights is concerned, several previous works [4, 5, 6] which either update only high-gradient or low-magnitude parameters have already demonstrated that it yields low performance gains in practice.

---

> > ### Author Response · Authors · 2025-11-19
> > **Response to Reviewer Evgh's Concerns [Part 2]**
> >
> > > **W3: Novelty**
> >
> > We respectfully disagree with the fact that our ideas are not novel. As already explained in Section 2.1 of our paper, previous sparse fine-tuning techniques [4, 5, 8, 9] either only select parameters based on whether they have the largest task gradient magnitudes or whether they have the smallest pre-trained magnitudes. To the best of our knowledge, ours is the first sparse fine-tuning technique which selects fine-tuning parameters based on a dual selection criteria: high task gradient magnitudes and low pre-trained magnitudes, which, as we show through an extensive set of experiments, consistently improves generalizability and ensures stability. We also introduce two new metrics, the forget ratio and the collapse rate, to quantify the extent of catastrophic forgetting and severe memorization, and show that GaLLoP is the only sparse fine-tuning algorithm which attains 0\% forget ratios and 0\% collapse rates across all density levels and experimental runs.
> >
> > The novelty of our paper has also been appreciated by reviewer `RmwJ` who finds our core idea of selecting a sparse set of parameters that are simultaneously high-gradient and low-magnitude a *creative fusion* of two complementary signals to target updates that help in-domain learning without erasing pre-trained knowledge, our implementation a *clean two-phase procedure*, our contribution of *two new evaluation diagnostics (forget ratio and collapse rate)* which formalize catastrophic forgetting and severe memorization as a way to *easily compute and compare them across methods*, the fact that *GaLLoP delivers strong OOD gains while curbing overfitting that plagues several popular baselines—an outcome that matters for reliable deployment of fine-tuned LLMs*, and the fact that *because GaLLoP is simple, model-agnostic, and efficient, it lowers the barrier to adopting sparse fine-tuning in practice*. Furthermore, reviewer `xcfq` finds our proposed scoring criterion *conceptually clear* and *easy to implement* with a *simple and straightforward design* and reviewer `dMY7` also finds our sparse fine-tuning (SpFT) algorithm *simple, intuitive, and effective* which is *well-motivated* and supported with *exhaustive experimental results* which *demonstrate the superiority of the proposed technique over related SpFT techniques*.
> >
> > > **Q1: Explanation of Fig.3**
> >
> > We would like to clarify that catastrophic forgetting does not always take place, i.e., OOD performance upon fine-tuning on a given downstream task dataset does not necessarily lead to the loss of pre-trained knowledge and instead, might actually even enhance the OOD performance of the base (vanilla) model. The knowledge gained upon fine-tuning a model on a downstream task dataset can assist a model in performing well on a given task which although possesses a distribution different from the fine-tuning dataset, is in fact, related to it in some way.
> >
> > We respectfully disagree with the fact that this observation of ours goes against what other papers on forgetting show. In fact, one of the papers suggested by you, the FLOW [3] paper, actually points out the same observation in its experiments (Section 6.1; page 6 and Table 2; page 7): 'FLOW helps preserve (*and even somewhat enhance*) the general capabilities of the pre-trained model.' Even Fig.2 (page 5) of the LiNeS paper [7] (incorporated for comparisons in our paper) shows that some of the fine-tuned models edited with LiNeS demonstrate higher OOD performance than their pre-trained counterparts on control tasks.
> >
> > In our paper, we indeed show that catastrophic forgetting does take place when the OOD performance of a fine-tuned model falls below the performance of its vanilla counterpart and have discussed this in detail in Section 6.2 of our paper. In fact, our experiments show that GaLLoP is the only fine-tuning algorithm which yields 0\% forget ratios across all density levels and experimental runs and hence, consistently prevents catastrophic forgetting.
> >
> > **We hope that our response clarifies your concerns and we kindly request you to reconsider and increase your score.**

---

> > > ### Author Response · Authors · 2025-11-19
> > > **References**
> > >
> > > [1] James Kirkpatrick *et al.*. Overcoming catastrophic forgetting in neural networks. Proceedings of the National Academy of Sciences, 114(13):3521–3526, 2017. URL: https://www.pnas.org/doi/abs/10.1073/pnas.1611835114.
> > >
> > > [2] Yupeng Chen *et al.*. MoFO: Momentum-filtered optimizer for mitigating forgetting in LLM fine-tuning. Transactions on Machine Learning Research, 2025. URL: https://openreview.net/forum?id=T1qXIDn9my.
> > >
> > > [3] Sunny Sanyal *et al.*. Upweighting easy samples in fine-tuning mitigates forgetting. In Forty-second International Conference on Machine Learning, 2025. URL: https://openreview.net/forum?id=13HPTmZKbM.
> > >
> > > [4] Bac Nguyen *et al.*. SAFT: Towards out-of-distribution generalization in fine-tuning. In Computer Vision – ECCV 2024: 18th European Conference, Milan, Italy, September 29–October 4, 2024, Proceedings, Part LXIX, pp. 138–154. URL: https://doi.org/10.1007/978-3-031-72890-7_9.
> > >
> > > [5] Yi-Lin Sung *et al.*. Training neural networks with fixed sparse masks. In A. Beygelzimer, Y. Dauphin, P. Liang, and J. Wortman Vaughan (eds.), Advances in Neural Information Processing Systems, 2021. URL: https://openreview.net/forum?id=Uwh-v1HSw-x.
> > >
> > > [6] Chao Zhou *et al.*. Pay attention to small weights, 2025. URL: https://arxiv.org/abs/2506.21374.
> > >
> > > [7] Ke Wang *et al.*. LiNeS: Post-training layer scaling prevents forgetting and enhances model merging. In The Thirteenth International Conference on Learning Representations, 2025. URL: https://openreview.net/forum?id=J5sUOvlLbQ.
> > >
> > > [8] Baohao Liao *et al.*. Parameter-efficient fine-tuning without introducing new latency. In Anna Rogers, Jordan Boyd-Graber, and Naoaki Okazaki (eds.), Proceedings
> > > of the 61st Annual Meeting of the Association for Computational Linguistics (Volume 1: Long Papers), pp. 4242–4260, Toronto, Canada, July 2023. Association for Computational Linguistics. URL: https://aclanthology.org/2023.acl-long.233/.
> > >
> > > [9] Alan Ansell *et al.*. Scaling sparse fine-tuning to large language models, 2024. URL: https://arxiv.org/abs/2401.16405.

---

### Meta-Review · Area_Chair_5FSx · 2025-12-23

**Summary:**

The paper proposes a parameter-efficient fine-tuning method, GaLLoP, which updates only model parameters with large gradient magnitudes on downstream tasks and small magnitudes in the pretrained model. The authors evaluate GaLLoP on LLaMA-3 and Gemma, reporting improvements in both in-distribution and out-of-distribution performance.

Reviewers raise concerns regarding the lack of theoretical justification, unclear motivation, limited novelty relative to existing methods, unverified assumptions, limited experimental scope, marginal gains over strong baselines, and insufficient discussion of efficiency.

Although the authors addressed some reviewer comments during the rebuttal, the AC finds that the core weaknesses remain. In particular, the method lacks convincing theoretical support, relies on unverified assumptions, and demonstrates only marginal improvements over baselines. Therefore, the AC recommends rejection.

**Reviewer Concerns:**

Reviewers raised concerns regarding the lack of theoretical justification, unclear motivation, limited novelty relative to existing methods, unverified assumptions, limited experimental scope, marginal gains over strong baselines, and insufficient discussion of efficiency. While the authors actively addressed some concerns, the core weaknesses remain: the paper lacks clear theoretical support for the method design, and the improvements over baselines are marginal.

**Reviewer Scores:**

The reviewer scores changed from 2, 2, 6, 4 to 2, 2, 8, 4, given the author–reviewer discussion context.

---

### Decision · Program_Chairs · 2026-01-26

Reject